# ENSEMBLING SPARSE AUTOENCODERS

## ABSTRACT

Sparse autoencoders (SAEs) are used to decompose neural network activations into human-interpretable features. Typically, features learned by a single SAE are used for downstream applications. However, it has recently been shown that SAEs trained with different initial weights can learn different features, demonstrating that a single SAE captures only a limited subset of features that can be extracted from the activation space. Motivated by this limitation, we introduce and formalize SAE ensembles. Furthermore, we propose to ensemble multiple SAEs through *naive bagging* and *boosting*. In naive bagging, SAEs trained with different weight initializations are ensembled, whereas in boosting SAEs sequentially trained to minimize the residual error are ensembled. Theoretically, naive bagging and boosting are justified as approaches to reduce reconstruction error. Empirically, we evaluate our ensemble approaches with three settings of language models and SAE architectures. Our empirical results demonstrate that, compared to the base SAE and an expanded SAE that matches the number of features in the ensemble, ensembling SAEs can improve the reconstruction of language model activations, diversity of features, and SAE stability. Additionally, on downstream tasks such as concept detection and spurious correlation removal, SAE ensembles achieve better performance, showing improved practical utility.

## 1 INTRODUCTION

Sparse autoencoders (SAEs) have been shown to decompose neural network activations[1] into a high-dimensional and sparse space of human-interpretable features (Cunningham et al., 2023; Gao et al., 2024; Lieberum et al., 2024; Rajamanoharan et al., 2024a). Recent work has focused on the application of SAEs to language models with interpretability use cases such as detecting concepts (Gao et al., 2024; Movva et al., 2025), identifying internal mechanisms of model behaviors (Marks et al., 2024), and steering model behaviors (Farrell et al., 2024; Marks et al., 2024; O'Brien et al., 2024). In practice, a single SAE is usually selected for downstream interpretability applications. However, it has recently been shown that SAEs trained on the same activations learn different features while differing only in their initial weights (Fel et al., 2025; Paulo and Belrose, 2025). This suggests that, even with the same architecture and hyperparameters, each SAE captures a different and yet limited subset of features that can be extracted from the activation space, demonstrating that using a single SAE may not suffice in practice.

Therefore, our main question is: *Can we leverage multiple SAEs to improve performance?* This perspective is motivated by ensemble methods in supervised learning that leverage model variability to improve predictive performance. Classical examples include bagging (bootstrap aggregating) that leverages variability due to randomness (Breiman, 1996; 2001) and boosting that leverages variability due to different optimization objectives (Chen and Guestrin, 2016; Friedman, 2001). We propose to ensemble multiple SAEs and formalize SAE ensembles. Conceptually, SAE ensembles are defined as methods for combining the outputs of SAEs in the activation space. Nonetheless, we show that ensembling the outputs of SAEs corresponds to concatenating the SAE features and feature coefficients. We instantiate two approaches for ensembling SAEs (Figure 1). In *naive bagging*, SAEs differing only in their weight initializations are ensembled. In *boosting*, the ensemble aggregates SAEs that are iteratively trained to reconstruct the residual from previous iterations. In three settings of language models and SAE architectures, our empirical results show that naive bagging and boosting can lead to better reconstruction of language model activations, more diverse features, and better

---

[1]Activations from neural networks are often also described as embeddings or representations.

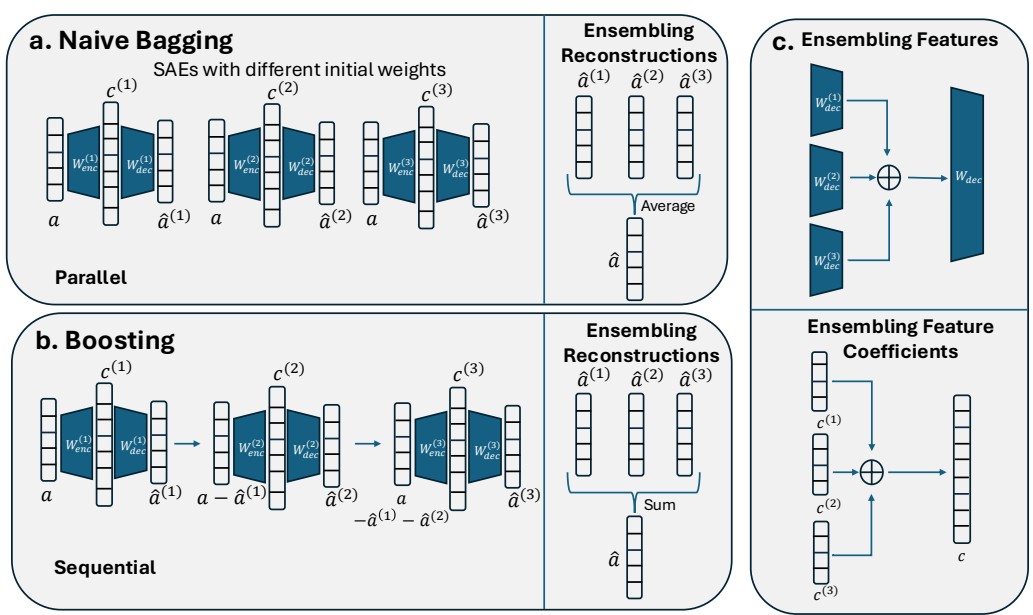

Figure 1: Overview of the proposed SAE ensembling strategies. **a.** *Naive Bagging* involves multiple SAEs with different weight initializations, which can be trained in parallel. The ensembled reconstruction is the average of reconstructions obtained from the individual SAEs. **b.** *Boosting* involves sequential training of SAEs on the residual error left from the previous iterations. The ensembled reconstruction is the sum of the reconstructions from the individual SAEs. **c.** For both approaches, ensembling the features and feature coefficients involves a concatenation.

stability. Finally, to demonstrate the practical utility of our ensemble methods, we apply them to the tasks of concept detection and spurious correlation removal, where ensembling multiple SAEs can outperform using only one SAE.

## 2    RELATED WORK

**SAEs.** SAEs have emerged as a scalable and unsupervised approach for extracting human-interpretable features from neural network activations (Cunningham et al., 2023; Fel et al., 2023), with recent work demonstrating their applications to language models (Gao et al., 2024; Lieberum et al., 2024). An SAE decomposes neural network activations into sparse linear combinations of features, which are vectors with the same dimensionality as the original activations. Overall, features learned by an SAE can often be annotated with semantic interpretations (Cunningham et al., 2023; Rao et al., 2024). Because the immediate goal of training an SAE is to decompose activations into sparse combinations of features, intrinsic metrics such as the explained variance of reconstructions and feature sparsity are used to evaluate SAEs (Gao et al., 2024; Rajamanoharan et al., 2024a;b). At the same time, SAEs are usually trained with the end goal of interpreting language model behaviors, with downstream use cases such as concept detection (Gao et al., 2024; Movva et al., 2025), mechanistic interpretability (Marks et al., 2024), and model steering (Farrell et al., 2024; Marks et al., 2024; O'Brien et al., 2024). Therefore, metrics specific to downstream applications such as concept detection accuracy and the SHIFT score have been proposed (Karvonen et al., 2025).

**Variability of SAEs.** In general, the variability between multiple SAEs can come from several sources. First, SAEs with different architecture designs can learn different features. For example, it has been shown that the choice of SAE activation function corresponds to assumptions about the separability structure of the features to be learned (Hindupur et al., 2025). The SAE size also has an impact on the types of features learned—a smaller SAE tends to learn high-level features, while a larger SAE tends to learn more specific features (Chanin et al., 2024). Second, given a fixed architecture, SAEs with different training hyperparameters can also learn different features. For example, it has been found that lower learning rates can help reduce the number of dead features that

rarely activate (Gao et al., 2024). Finally, it has been shown that SAEs can learn different features even with the same architecture and hyperparameters, for example due to different initial weights (Fel et al., 2025; Paulo and Belrose, 2025). For the scope of this paper, we focus on the variability and ensembling of SAEs with the same architecture and hyperparameters. In other words, our SAE ensemble approaches are considered meta-algorithms compatible with any SAE architecture and hyperparameter configuration.

**Model ensembling.** Ensemble methods have been applied to leverage model variability for improving performance, especially in supervised learning. In bagging (bootstrap aggregating), predictions from models trained with bootstrapped data subsets are aggregated (Breiman, 1996; 2001). Boosting algorithms train successive models by focusing on the errors made in the previous iterations (Chen and Guestrin, 2016; Friedman, 2001). Stacking is an alternative framework that combines predictions from models with different architectures and inductive biases (Wolpert, 1992). More recently, it has been shown that averaging weights of models can lead to improved accuracy without additional inference time (Wortsman et al., 2022a;b). For unsupervised learning, ensemble methods have mostly been applied to form consensus for clustering and anomaly detection (Aggarwal, 2013; Domeniconi and Al-Razgan, 2009; Fern and Brodley, 2004; Ghosh and Acharya, 2011; Zimek et al., 2014). Motivated by the principle of ensembling, here we propose that SAEs can also be ensembled with respect to their outputs in the activation space. We theoretically show that ensembling SAE reconstructions corresponds to combining SAE features. We also demonstrate that ensembling SAEs can lead to improved intrinsic performance and practical utility when applied to language models.

Our work makes the following contributions. (1) We propose ensembling SAEs as a formal framework, showing that ensembling SAE reconstructions is equivalent to ensembling SAE features. (2) We instantiate two practical ensemble approaches, naive bagging and boosting, with theoretical justifications in relation to reconstruction performance. (3) We empirically demonstrate that ensembling multiple SAEs can improve performance in intrinsic metrics and downstream applications.

## 3 FORMALIZING SAE ENSEMBLES

This section provides the notation used throughout this paper, the definition of an SAE ensemble, and a theoretical result showing that ensembling SAEs is equivalent to concatenating their features.

### 3.1 NOTATION

In general, we consider a neural network that maps from a sample space $\mathcal{X}$ to a $d$-dimensional activation space. An SAE is an autoencoder $g : \mathbb{R}^d \to \mathbb{R}^d$ that reconstructs neural network activations, with the following form:

$$g(\mathbf{a}; \mathbf{W}_{\text{enc}}, \mathbf{W}_{\text{dec}}, \mathbf{b}_{\text{enc}}, \mathbf{b}_{\text{dec}}) = \mathbf{W}_{\text{dec}} h(\mathbf{W}_{\text{enc}} \mathbf{a} + \mathbf{b}_{\text{enc}}) + \mathbf{b}_{\text{dec}}, \tag{1}$$

where $\mathbf{W}_{\text{enc}} \in \mathbb{R}^{k \times d}, \mathbf{b}_{\text{enc}} \in \mathbb{R}^k, \mathbf{W}_{\text{dec}} \in \mathbb{R}^{d \times k}, \mathbf{b}_{\text{dec}} \in \mathbb{R}^d$ are the SAE weights and biases, and $h : \mathbb{R}^k \to \mathbb{R}^k$ is an activation function such as the ReLU, JumpReLU, and TopK functions (Cunningham et al., 2023; Lieberum et al., 2024; Gao et al., 2024).[2] Unlike conventional autoencoders, in an SAE we have $k > d$. Notably, the columns of the decoder matrix $\mathbf{W}_{\text{dec}}$ are considered features learned by the SAE. Particularly, let $\mathbf{W}_{\text{dec}}[:, i]$ denote the $i$th column of the decoder matrix. Then $\mathbf{f}_i = \mathbf{W}_{\text{dec}}[:, i] \in \mathbb{R}^d$ is the $i$th feature of the SAE,[3] for $i \in [k]$. Furthermore, elements in $\mathbf{c} = h(\mathbf{W}_{\text{enc}} \mathbf{a} + \mathbf{b}_{\text{enc}}) \in \mathbb{R}^k$ are considered coefficients for the features. Overall, Equation (1) can be rewritten to highlight that an SAE decomposes an activation into features, as follows:

$$g(\mathbf{a}; \mathbf{W}_{\text{enc}}, \mathbf{W}_{\text{dec}}, \mathbf{b}_{\text{enc}}, \mathbf{b}_{\text{dec}}) = \sum_{i=1}^{k} c_i \mathbf{f}_i + \mathbf{b}_{\text{dec}}. \tag{2}$$

For conciseness, we let $\theta = (\mathbf{W}_{\text{enc}}, \mathbf{W}_{\text{dec}}, \mathbf{b}_{\text{enc}}, \mathbf{b}_{\text{dec}})$ denote all the SAE parameters. Finally, we use $\hat{\mathbf{a}} = g(\mathbf{a}; \theta)$ to denote the SAE reconstruction.

---

[2] Here, $k$ denotes the SAE dimension and does not denote the number of active features in TopK SAE.

[3] In the literature, the $\mathbf{f}_i$'s are associated with different terms such as feature directions and decoder vectors. Here, we follow Cunningham et al. (2023) and call them features for brevity.

To train an SAE, a training set of activations $\{\mathbf{a}^{(n)}\}_{n=1}^N$ are collected by passing a set of samples $\{\mathbf{x}^{(n)}\}_{n=1}^N$ through the neural network. Then the SAE parameters are trained to minimize the following empirical loss:

$$\mathcal{L}_{\text{SAE}}\left(\{\mathbf{a}^{(n)}\}_{n=1}^N; \theta\right) = \frac{1}{N}\sum_{n=1}^N\left[\underbrace{\left\|\mathbf{a}^{(n)} - g\left(\mathbf{a}^{(n)}; \theta\right)\right\|_2^2}_{\text{reconstruction loss}} + \lambda\underbrace{\left\|\mathbf{c}^{(n)}\right\|_p}_{\text{sparsity loss}}\right], \tag{3}$$

where $\mathbf{c}^{(n)} = h(\mathbf{W}_{\text{enc}}\mathbf{a}^{(n)} + \mathbf{b}_{\text{enc}})$ corresponds to the feature coefficients for the $n$th sample, and $\lambda \geq 0$ is the penalty coefficient for the sparsity loss.

### 3.2 SAE Ensembles

In this work we focus on ensembling SAEs with the same architecture. Specifically, given $J$ SAEs with model parameters $\theta^{(j)}$ for $j \in [J]$, an SAE ensemble has the form:

$$\sum_{j=i}^J \alpha^{(j)} g\left(\cdot; \theta^{(j)}\right) \tag{4}$$

where $\alpha^{(j)} \geq 0$ is the ensemble weight for the $j$th SAE, and for generality the notation $g(\cdot; \theta^{(j)})$ indicates that each SAE can take arbitrary inputs in $\mathbb{R}^d$. This weighted-sum formulation is similar to classical ensemble methods, where a weighted sum of outputs from base models is used to make a prediction (Breiman, 1996; Friedman, 2001). With an SAE ensemble, the base model is now an SAE.

Different from classical ensembles, Equation (4) by itself does not fully specify an SAE ensemble, since SAE features and their coefficients are also critical components for downstream analyses. Interestingly, because the output of each SAE is a linear combination of its features, ensembling SAEs is equivalent to concatenating their feature coefficients and their decoder matrices (feature vectors). More formally, we have the following proposition, with the proof in Appendix A.

**Proposition 1.** *Suppose there are $J$ SAEs $g(\cdot; \theta^{(1)}), ..., g(\cdot; \theta^{(J)})$, with decoder matrices $\mathbf{W}_{dec}^{(1)}, ..., \mathbf{W}_{dec}^{(J)} \in \mathbb{R}^{d \times k}$ and decoder biases $\mathbf{b}_{dec}^{(1)}, ..., \mathbf{b}_{dec}^{(J)} \in \mathbb{R}^d$. For a given neural network activation $\mathbf{a} \in \mathbb{R}^d$, let $\mathbf{c}^{(1)}, ..., \mathbf{c}^{(J)} \in \mathbb{R}^k$ denote the feature coefficients. Then ensembling the $J$ SAEs is equivalent to reconstructing $\mathbf{a}$ with:*

$$\hat{\mathbf{a}} = \mathbf{W}_{dec}\mathbf{c} + \mathbf{b}_{dec} = \sum_{i'=1}^{kJ} \mathbf{c}_{i'}\mathbf{f}_{i'} + \mathbf{b}_{dec}, \tag{5}$$

*where*

$$\mathbf{c} = \begin{bmatrix} \alpha^{(1)}\mathbf{c}^{(1)} \\ \vdots \\ \alpha^{(J)}\mathbf{c}^{(J)} \end{bmatrix}, \mathbf{W}_{dec} = \left[\mathbf{W}_{dec}^{(1)} \cdots \mathbf{W}_{dec}^{(J)}\right], \mathbf{b}_{dec} = \sum_{j=1}^J \alpha^{(j)}\mathbf{b}_{dec}^{(j)}, \tag{6}$$

*and $\mathbf{f}_{i'} = \mathbf{W}_{dec}[:, i']$, with $\mathbf{c} \in \mathbb{R}^{kJ}, \mathbf{W}_{dec} \in \mathbb{R}^{d \times kJ}, \mathbf{b}_{dec} \in \mathbb{R}^d$.*

**Remark 1.** The ensemble weights $\{\alpha^{(j)}\}_{j=1}^J$ can be folded into either $\mathbf{c}$ or $\mathbf{W}_{\text{dec}}$ for Proposition 1 to hold. Since the columns of $\mathbf{W}_{\text{dec}}$ are often constrained to have unit norms to interpret the features as direction vectors (Cunningham et al., 2023; Rajamanoharan et al., 2024a), the ensemble weights are folded into $\mathbf{c}$ to retain the feature norms.

## 4 Ensemble Methods for SAEs

In this section we describe *naive bagging* and *boosting* as two approaches for ensembling SAEs.

### 4.1 Naive Bagging

Variability of SAEs due to weight initialization is utilized in naive bagging, motivated by prior work showing that SAEs differing only in their initial weights can learn different features (Fel et al., 2025;

Paulo and Belrose, 2025). Note that we refer to this method as *naive* because, unlike classical bagging, bootstrapped data subsets are not used. This is to ensure that each SAE is trained on the same dataset and isolate the effect of different initializations. Also, as SAEs are often trained on million- or even billion-scale datasets (Gao et al., 2024; Lieberum et al., 2024), bootstrapping becomes impractical due to memory and storage overhead. Concretely, given $J$ SAEs with different initial weights, naive bagging gives the following ensembled SAE:

$$g_{\text{NB}}\left(\mathbf{a}^{(*)}; \{\theta^{(j)}\}_{j=1}^{J}\right) = \frac{1}{J} \sum_{j=1}^{J} g\left(\mathbf{a}^{(*)}; \theta^{(j)}\right) \tag{7}$$

Conceptually, the uniform ensemble weight $\alpha^{(j)} = 1/J$ is motivated by considering naive bagging as a way to reduce reconstruction variance in the bias-variance decomposition (see Proposition 2 in Appendix A for a formal justification).

## 4.2 BOOSTING

Since SAEs with different initial weights still learn some overlapping features (Paulo and Belrose, 2025), naive bagging can result in redundant features in the ensemble. To address this redundancy, we propose a boosting-based ensemble strategy to encourage SAEs to capture different components of a given activation through sequential training. Starting from an initial SAE, each subsequent SAE is trained to capture the residual left from the previous iteration. Concretely, the $j$th SAE is trained with the following loss:

$$\mathcal{L}_{\text{Boost}}\left(\{\mathbf{a}^{(n)}\}_{n=1}^{N}; \theta^{(j)}\right) = \frac{1}{N} \sum_{n=1}^{N} \left[\left\|\mathbf{a}^{(n,j)} - g\left(\mathbf{a}^{(n,j)}; \theta^{(j)}\right)\right\|_2^2 + \lambda \left\|\mathbf{c}^{(n,j)}\right\|_p\right], \tag{8}$$

where

$$\mathbf{a}^{(n,j)} = \begin{cases} \mathbf{a}^{(n)}, & \text{if } j = 1. \\ \mathbf{a}^{(n)} - \sum_{\ell=1}^{j-1} g\left(\mathbf{a}^{(n,\ell)}; \theta^{(\ell)}\right), & \text{otherwise.} \end{cases}$$

Here, the first iteration corresponds to training an initial SAE with the original activations. For $j > 1$, $\mathbf{a}^{(n,j)}$ is the residual left from the $(j-1)$th iteration that the $j$th SAE should learn to reconstruct. It is worth noting that the regularization parameters $\lambda$ and $p$ remain the same throughout the training iterations. Intuitively, each SAE in boosting should learn features different from the previous SAEs by capturing the residual. A recent work adapts the matching pursuit algorithm for SAEs, known as MP-SAEs, which are similar to boosting because residuals are used across multiple iterations (Costa et al., 2025). However, MP-SAE is not an ensemble method since the same set of features are shared across all iterations, while each boosting run learns a separate set of features.

As another motivation, boosting can also lead to good reconstruction performance by bounding the bias term in the bias-variance decomposition (see Proposition 3 in Appendix A for a formal justification). Overall, given $J$ SAEs trained with Equation (8), boosting gives the following ensembled SAE:

$$g_{\text{Boost}}\left(\mathbf{a}^{(*)}; \{\theta^{(j)}\}_{j=1}^{J}\right) = \sum_{j=1}^{J} g\left(\mathbf{a}^{(*,j)}; \theta^{(j)}\right). \tag{9}$$

Note that boosting is trained sequentially, and the forward pass also runs sequentially during inference.

## 5 EXPERIMENTS

In this section, we evaluate our ensemble approaches with intrinsic evaluation metrics (Section 5.2) and demonstrate the utility of ensembling SAEs with two use cases (Section 5.3 and Section 5.4).

## 5.1 BASELINES

As baselines for each experimental setting, we compare ensemble methods with the base SAE and with an expanded SAE trained to have the same number of features as the ensembled SAEs. Since sparsity can have an impact on SAE performance for a given SAE size (Gao et al., 2024), expanded

SAEs are trained to have sparsity comparable to the ensembled SAEs, enabling a fair comparison. More details about the expanded SAE baseline are provided in Appendix C.

## 5.2 Evaluating Ensembled SAEs with Intrinsic Metrics

### 5.2.1 Setup

We evaluate our ensemble approaches on SAEs trained with activations from three different language models: GELU-1L, Pythia-160M, and Gemma 2-2B, which represent a range of model sizes. Following prior work, ReLU, TopK, and JumpReLU SAEs are trained with the residual stream activations from GELU-1L (Bricken et al., 2023), layer 8 from Pythia-160M (Gao et al., 2024), and layer 12 from Gemma 2-2B (Lieberum et al., 2024), respectively. Per-token activations are obtained from the Pile (Gao et al., 2020) for each language model with the corresponding context size. For training the SAEs, we use 800 million tokens from a version of the Pile with copyrighted contents removed.[4] A held-out test set of 7 million tokens is used for evaluation. Hyperparameters are swept for the base SAE, and hyperparameters giving an explained variance closest to 90% are selected. This ensures that the SAEs being ensembled are practically usable to explain the activations. All SAEs are trained using the Adam optimizer (Kingma and Ba, 2014). Additional details about the language models along with training times, inference times, and hyperparameter selection are provided in Appendix I.

### 5.2.2 Metrics

We evaluate different aspects of the ensembled SAEs using six intrinsic metrics: Explained Variance, Mean Squared Error (MSE), Relative Sparsity, Diversity, Connectivity, and Stability. Details about each of the metrics are provided in Appendix B. Stability is computed across 5 runs and the confidence intervals are obtained by using each of the runs as the base run for computing stability with the others.

### 5.2.3 Results

Figure 2 illustrates how the number of SAEs in the ensemble affects intrinsic performance for both naive bagging and boosting on Gemma 2-2B. The first point in each plot represents the base SAE. Consistent with prior work (Lieberum et al., 2024; Paulo and Belrose, 2025), the similarity threshold is set to $\tau = 0.7$ for the diversity metric. For completeness, we provide the results with additional values for $\tau$ in Appendix J. Increasing the number of SAEs in the ensemble generally improves performance for most metrics and maintains the performance for the others. Comparing the two ensemble approaches, boosting outperforms naive bagging across all metrics except for stability. This is consistent with the theoretical justification that naive bagging reduces variance (Section 4.1). On the other hand, since boosting aims for bias reduction (Section 4.2), it can learn more specific and low-level features, impacting stability. Also, boosting has a lower relative sparsity for >2 SAEs in the ensemble, indicating that boosting requires fewer active features. The boosted SAE can discover a higher number of diverse features, in terms of both feature directions and coefficients, as measured in diversity and connectivity. Results for GELU-1L and Pythia-160M are provided in Appendix E, where similar trends hold.

Detailed results for ensembles of 8 SAEs across all three language models are summarized in Table 1. We ensemble with 8 SAEs as most of the metrics begin to plateau by then. Compared to the base SAE, ensembling performs better in all the intrinsic metrics. Compared to an expanded SAE, naive bagging (NB) performs better in stability while worse in the other intrinsic metrics such as the reconstruction metrics. This is expected due to the stability-reconstruction tradeoff (Fel et al., 2025). However, as naive bagging improves both reconstruction and stability compared to the base SAE, it is reasonable that naive bagging can also be applied to the expanded SAE as a way to gain both reconstruction performance and stability. Notably, the stability of the expanded SAE is typically less than half of the stability of ensembled SAEs, indicating that a larger SAE can result in unreliable features. More importantly, boosting outperforms an expanded SAE in the reconstruction metrics, diversity, and stability, while having similar connectivity scores. This comparison highlights that the gains from ensembling are not just because the ensembled SAEs have more features. This comparison also shows that boosting is a strong alternative to expanding SAE size, especially for its better stability in applications that require interpretability tools to be reliable (Fel et al., 2025; Paulo and Belrose,

---

[4]https://huggingface.co/datasets/monology/pile-uncopyrighted

2025). Overall, we find that ensembling performs better than the base SAE and an expanded SAE on the intrinsic metrics.

Table 1: Intrinsic evaluation metrics for the base SAE, an expanded SAE, naive bagging (NB), and boosting (ensembling 8 SAEs). Means along with 95% confidence intervals are reported across 5 runs. Each expanded SAE is tuned to have an L0 similar to that of boosting, resulting in similar relative sparsities.

| Ensembling Method | Explained Variance (↑) | Relative Sparsity (↓) | MSE (↓) | Diversity (↑) | Connectivity (↑) | Stability (↑) |
|---|---|---|---|---|---|---|
| **GELU-1L** | | | | | | |
| Base SAE | 0.875 (0.0020) | 0.023 (0.0002) | 41.694 (0.536) | 16276.7 (10.47) | 0.307 (0.0057) | 0.705 (0.0016) |
| Expanded SAE | 0.946 (0.0003) | 0.007 (0.0000) | 17.893 (0.137) | 130411.6 (21.18) | **0.959 (0.0003)** | 0.372 (0.0022) |
| Ensembling (NB) | 0.895 (0.0006) | 0.023 (0.0000) | 35.147 (0.210) | 53087.0 (179.24) | 0.307 (0.0009) | **0.745 (0.0002)** |
| Ensembling (Boosting) | **0.961 (0.0018)** | **0.006 (0.0000)** | **12.542 (0.589)** | **130913.0 (5.48)** | 0.945 (0.0004) | 0.707 (0.0014) |
| **Pythia-160M** | | | | | | |
| Base SAE | 0.906 (0.0003) | 0.008 (0.0000) | 32.965 (0.077) | 15804.5 (0.02) | 0.912 (0.0013) | 0.677 (0.0026) |
| Expanded SAE | 0.987 (0.0041) | 0.008 (0.0000) | 4.387 (1.486) | **127821.0 (113.7)** | 0.978 (0.0006) | 0.204 (0.0006) |
| Ensembling (NB) | 0.929 (0.0000) | 0.008 (0.0000) | 24.704 (0.019) | 50390.0 (0.05) | 0.912 (0.0006) | **0.731 (0.0017)** |
| Ensembling (Boosting) | **0.998 (0.0021)** | 0.008 (0.0000) | **0.845 (0.547)** | 117018.2 (0.09) | **0.986 (0.0004)** | 0.680 (0.0025) |
| **Gemma 2-2B** | | | | | | |
| Base SAE | 0.920 (0.0006) | 0.059 (0.0002) | 716.659 (5.875) | 16013.0 (5.88) | 0.768 (0.0016) | 0.581 (0.0006) |
| Expanded SAE | 0.948 (0.0012) | 0.021 (0.0001) | 472.330 (10.759) | 127779.0 (69.33) | **0.993 (0.0003)** | 0.268 (0.0021) |
| Ensembling (NB) | 0.974 (0.0006) | 0.059 (0.0000) | 234.128 (6.228) | 58859.6 (295.38) | 0.769 (0.0007) | **0.633 (0.0014)** |
| Ensembling (Boosting) | **0.995 (0.0003)** | 0.021 (0.0002) | **46.538 (2.923)** | **128415.6 (114.89)** | 0.989 (0.0003) | 0.583 (0.0009) |

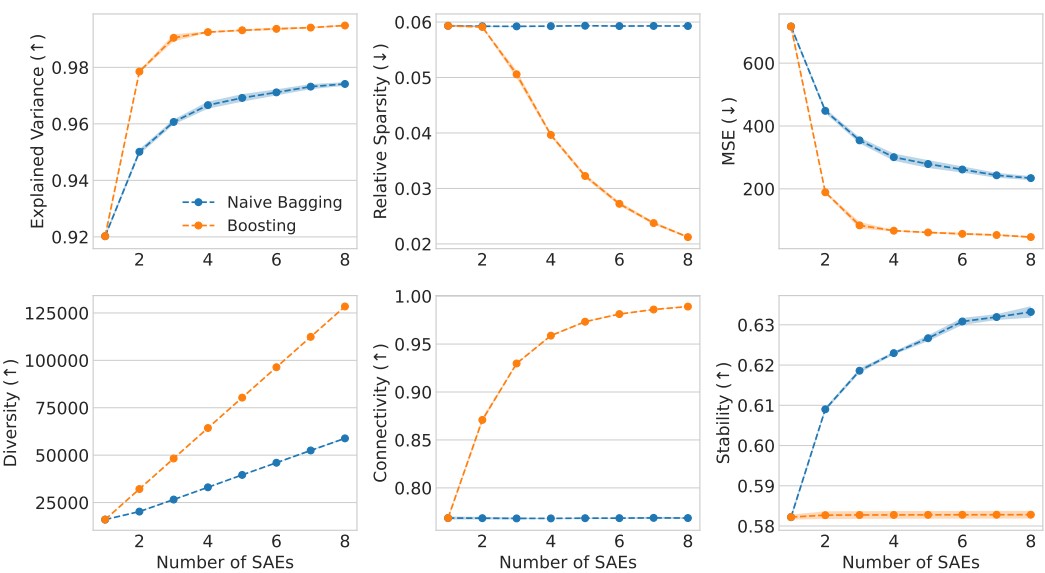

Figure 2: Ablation study to investigate the effect of the number of SAEs in the ensemble for naive bagging and boosting on the intrinsic evaluation metrics for Gemma 2-2B (Layer 12). The shaded regions indicate 95% confidence intervals across 5 different experiment runs. For naive bagging, the different experiment runs correspond to different sets of initial weights.

## 5.3 USE CASE 1: CONCEPT DETECTION

Interpretability use cases of SAEs such as debiasing, understanding sparse circuits, and hypothesis generation often require individual SAE features to correspond to semantic concepts (Cunningham et al., 2023; Marks et al., 2024; Movva et al., 2025). Therefore, here we apply our ensemble approaches to detect semantic concepts across a range of domains. Specifically, per-token activations are encoded using an ensembled SAE, and mean-pooling is applied to obtain a sequence-level

embedding. The SAE feature having the maximum mean difference between samples with and without the concept in the training set is selected to train a logistic regression classifier. Finally, accuracy on a held-out test set is used to evaluate the concept detection performance. We note that this evaluation procedure follows prior work (Gao et al., 2024; Karvonen et al., 2025).

**Setup.** We train a ReLU SAE as the base SAE on the residual stream activations from layer 4 of Pythia-70M, with 100 million tokens from the Pile (Gao et al., 2020). This setting is chosen since it has been used for concept-level tasks (Karvonen et al., 2024; Marks et al., 2024). Our concept detection use case encompasses four datasets: **(1) Amazon Review (Sentiment):** classifying the sentiment of the review (1 vs. 5 stars), **(2) GitHub Code:** identifying the coding language from source code, **(3) AG News:** classifying news articles by topics, and **(4) European Parliament:** detecting the language of a document.

**Results.** Table 2 (visualized in Supplementary Figure 8) illustrates the results of the concept detection task for our ensemble approaches (with 8 SAEs in each ensemble). Comparing the two ensemble approaches, naive bagging generally performs better than boosting in terms of the mean performance. One possible reason for the higher performance of naive bagging could be that it identifies features at a conceptual hierarchy which is suitable for this task, while boosting can potentially identify low-level features useful for the training set but do not generalize to the test set. However, multiple specific features can be combined to detect a more general concept. Indeed, boosting can perform better than naive bagging when the top 5 concept-associated features are considered instead of using only the top feature (Supplementary Table 1). Therefore, naive bagging should be used for applications where each concept is mapped to only one SAE feature, whereas boosting excels when each concept is mapped to multiple SAE features. Overall, considering the means along with the confidence intervals, we observe that ensembling performs slightly better than the base SAE and an expanded SAE across all the concept detection tasks (Table 2). For completeness, evaluation of SAEs trained for the other language models on the concept detection tasks is shown in Appendix G.

Table 2: Test accuracy of the logistic regression classifier for the top concept-associated feature across four concept detection tasks for ensembles with 8 SAEs. Means along with 95% confidence intervals are reported across 5 experiment runs.

|  | Amazon Review (Sentiment) | GitHub Code (Language) | AG News (Topic) | European Parliament (Language) |
|---|---|---|---|---|
| Base SAE | 0.618 (0.030) | 0.711 (0.020) | 0.733 (0.021) | 0.938 (0.016) |
| Expanded SAE | 0.600 (0.032) | 0.682 (0.025) | 0.746 (0.021) | 0.942 (0.009) |
| Ensembling (NB) | **0.631 (0.036)** | **0.715 (0.012)** | 0.742 (0.037) | **0.943 (0.016)** |
| Ensembling (Boosting) | 0.624 (0.037) | 0.682 (0.021) | **0.759 (0.021)** | 0.920 (0.015) |

## 5.4 Use Case 2: Spurious Correlation Removal

Neural networks have been previously shown to encode spurious correlations between non-essential input signals (e.g. image background) and the target label, which can negatively impact their generalization performance, robustness, and trustworthiness (DeGrave et al., 2021; Ye et al., 2024). Such biases can get exacerbated in more complex networks like large language models (Kotek et al., 2023; Navigli et al., 2023). Motivated by this, we consider the task of spurious correlation removal (SCR), as proposed in Karvonen et al. (2024). The evaluation procedure here follows Karvonen et al. (2025) and is an automated version of Sparse Human-Interpretable Feature Trimming (SHIFT) by Marks et al. (2024).

**Setup.** The goal of SCR is to identify specific SAE features for the spurious signal and debias a classifier by ablating those features. Here we use the Bias in Bios dataset (De-Arteaga et al., 2019), which maps professional biographies to profession and gender. First, the dataset is filtered for a pair of professions (e.g. professor and nurse) and then it is partitioned into two sets: one which is balanced in terms of profession and gender, and the other with biased gender association for a particular profession (e.g. male professors and female nurses). Then, a linear classifier $C_b$ is trained on the biased set using the activations from a language model. The goal is to debias this classifier using the features identified by the SAE to improve the accuracy on classifying profession in an unbiased held-out set. Complete details about the setup are provided in Appendix D.

**Results.** The (ensembled) SAEs from Section 5.3 are used here, with $L = 20$ features selected, following prior work (Karvonen et al., 2025). Table 3 (visualized in Supplementary Figure 9) shows the performance of our ensemble approaches for the SCR task across four pairs of profession, with the first profession biased towards males and the second towards females. Comparing the ensemble approaches, naive bagging does not perform as well as the baselines, which could be because in naive bagging there are more than $L$ similar features related to the spurious signal, and all of those features need to be ablated to observe an improved $A_{\text{abl}}$. In contrast, boosting outperforms naive bagging and the baselines, suggesting that it is more effective in isolating and removing gender-related features. Overall, these results show that ensembling can outperform the base SAE and an expanded SAE across all pairs of professions. Similar trends are observed as the number of top gender-related features $L$ is further increased (Supplementary Figure 7). For completeness, evaluation of additional SAEs on the SCR task is shown in Section G.

Table 3: $S_{\text{SHIFT}}$ scores for the spurious correlation removal task with the top 20 gender-related features identified across four pairs of professions for ensembles with 8 SAEs. Means along with 95% confidence intervals are reported across 5 experiment runs.

|  | **Professor vs. Nurse** | **Architect vs. Journalist** | **Surgeon vs. Psychologist** | **Attorney vs. Teacher** |
|---|---|---|---|---|
| Base SAE | 0.039 (0.008) | 0.004 (0.006) | 0.027 (0.006) | 0.017 (0.003) |
| Expanded SAE | 0.047 (0.014) | 0.006 (0.005) | 0.037 (0.009) | 0.021 (0.007) |
| Ensembling (NB) | 0.021 (0.003) | 0.004 (0.001) | 0.014 (0.002) | 0.003 (0.005) |
| Ensembling (Boosting) | **0.066 (0.016)** | **0.013 (0.011)** | **0.045 (0.014)** | **0.029 (0.003)** |

## 6 DISCUSSION

In this work, we propose and formalize ensembling SAEs as a way to improve performance by leveraging the feature variability of SAEs with the same architecture and hyperparameters. We instantiate two ensembling approaches, *naive bagging* and *boosting*. Theoretically, we justify both approaches as ways to improve reconstruction and show that ensembling in the output space of SAEs is equivalent to concatenation in the feature space. Empirically, we show that ensembling improves intrinsic performance, leading to better reconstruction of language model activations, more diverse features, and improved stability. We also demonstrate the practical utility of our ensembling approaches through quantitative validation on two downstream use cases, where ensembling can also lead to performance improvement.

Our ensemble approaches do come with some limitations. Both naive bagging and boosting are computationally more expensive than training the base SAE, since they require multiple SAEs to be trained. While this can be run in parallel for naive bagging, boosting has to be run sequentially. While ensembling performs better than the base SAE across all intrinsic metrics, this does not always translate to better downstream performance. For example, naive bagging could result in redundant features, causing a performance drop for tasks where multiple features are selected for ablation. On the other hand, boosting could learn features too specific, leading to lower performance for detecting high-level concepts with individual features. Thus, different ensemble approaches should be used based on the specific goals and procedures of downstream applications. Furthermore, similar to ensemble methods in supervised learning, ensemble approaches for SAE can incur higher model complexity compared to the base SAE. However, individual SAEs in an ensemble remain sparse, and the learned features are interpretable based on downstream evaluations.

As a framework, SAE ensembling can be considered a meta-algorithm, which can be extended to different settings. We scope this work to focus on SAEs with the same architecture, hyperparameters, and training data, but future directions can consider ensembling (stacking) different SAE architectures such as Matryoshka SAEs (Bussmann et al., 2025) and Switch SAEs (Mudide et al., 2024), or SAEs trained on data from different subdomains (Muhamed et al., 2025). Beyond language models, ensembling can also be used for SAEs trained on activations from models of other input domains (e.g. activations from vision models). Finally, future work can also explore ensembling from theoretical perspectives beyond reconstruction, such as feature identification.

## 7 REPRODUCIBILITY STATEMENT

The code to implement and evaluate the ensembling methods has been submitted as part of the supplementary material. It includes a README that describes the steps to obtain the datasets, train the base SAE, train the ensembling methods, and run the evaluations. Pseudocode for boosting is provided in Algorithm 1. Implementation details about the data and compute, along with hyperparameter selection curves are provided in Appendix I.

## 8 LLM USAGE

LLMs were used to identify and fix typos along with minor edits to improve presentation. No other aspects of writing the manuscript used LLMs.

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

# APPENDIX

## A  THEORETICAL RESULTS

Here we (re-)state and prove our results from Section 3.2, Section 4.1, and Section 4.2.

**Proposition 1.** *Suppose there are $J$ SAEs $g(\cdot; \theta^{(1)}), ..., g(\cdot; \theta^{(J)})$, with decoder matrices $\mathbf{W}_{dec}{}^{(1)}, ..., \mathbf{W}_{dec}{}^{(J)} \in \mathbb{R}^{d \times k}$ and decoder biases $\mathbf{b}_{dec}{}^{(1)}, ..., \mathbf{b}_{dec}{}^{(J)} \in \mathbb{R}^d$. For a given neural network activation $\mathbf{a} \in \mathbb{R}^d$, let $\mathbf{c}^{(1)}, ..., \mathbf{c}^{(J)} \in \mathbb{R}^k$ denote the feature coefficients. Then ensembling the $J$ SAEs is equivalent to reconstructing $\mathbf{a}$ with:*

$$\hat{\mathbf{a}} = \mathbf{W}_{dec}\mathbf{c} + \mathbf{b}_{dec} = \sum_{i'=1}^{kJ} \mathbf{c}_{i'}\mathbf{f}_{i'} + \mathbf{b}_{dec}, \tag{10}$$

*where*

$$\mathbf{c} = \begin{bmatrix} \alpha^{(1)}\mathbf{c}^{(1)} \\ \vdots \\ \alpha^{(J)}\mathbf{c}^{(J)} \end{bmatrix}, \mathbf{W}_{dec} = \begin{bmatrix} \mathbf{W}_{dec}{}^{(1)} \cdots \mathbf{W}_{dec}{}^{(J)} \end{bmatrix}, \mathbf{b}_{dec} = \sum_{j=1}^{J} \alpha^{(j)}\mathbf{b}_{dec}{}^{(j)}, \tag{11}$$

*and $\mathbf{f}_{i'} = \mathbf{W}_{dec}[:, i']$, with $\mathbf{c} \in \mathbb{R}^{kJ}, \mathbf{W}_{dec} \in \mathbb{R}^{d \times kJ}, \mathbf{b}_{dec} \in \mathbb{R}^d$.*

*Proof.* Based on the definition of an SAE ensemble in Equation (4) and the definition of feature coefficients, we have

$$\hat{\mathbf{a}} = \sum_{j=1}^{J} \alpha^{(j)} \left( \mathbf{W}_{\text{dec}}{}^{(j)}\mathbf{c}^{(j)} + \mathbf{b}_{\text{dec}}{}^{(j)} \right) \tag{12}$$

$$= \begin{bmatrix} \mathbf{W}_{\text{dec}}{}^{(1)} \cdots \mathbf{W}_{\text{dec}}{}^{(J)} \end{bmatrix} \begin{bmatrix} \alpha^{(1)}\mathbf{c}^{(1)} \\ \vdots \\ \alpha^{(J)}\mathbf{c}^{(J)} \end{bmatrix} + \sum_{j=1}^{J} \alpha^{(j)}\mathbf{b}_{\text{dec}}{}^{(j)} \tag{13}$$

$$= \mathbf{W}_{\text{dec}}\mathbf{c} + \mathbf{b}_{\text{dec}}, \tag{14}$$

where Equation (13) follows from observing that the sum of matrix-vector product is equivalent to the product of the concatenated matrix and vector. $\square$

Here, we provide a lemma showing the bias-variance decomposition for reconstructing a neural network activation with an ensembled SAE (Section 3.2).

**Lemma 1.** *Given a neural network activation $\mathbf{a}^{(*)}$, and the ensembled SAE $g_{Ens}(\cdot; \{\theta^{(j)}\}_{j=1}^{J})$ trained on activations $\{\mathbf{a}^{(n)}\}_{n=1}^{N}$, the expected reconstruction error can be decomposed into a bias term and a variance term. That is,*

$$\mathbb{E}_{\{\theta^{(j)}\}_{j=1}^{J}|\{\mathbf{a}^{(n)}\}_{n=1}^{N}} \left[ \left\| \mathbf{a}^{(*)} - g_{Ens}(\mathbf{a}^{(*)}; \{\theta^{(j)}\}_{j=1}^{J}) \right\|_2^2 \right] \tag{15}$$

$$= \underbrace{\left\| \mathbf{a}^{(*)} - \mathbb{E}_{\{\theta^{(j)}\}_{j=1}^{J}|\{\mathbf{a}^{(n)}\}_{n=1}^{N}}[g_{Ens}(\mathbf{a}^{(*)}; \{\theta^{(j)}\}_{j=1}^{J})] \right\|_2^2}_{\text{bias term}} \tag{16}$$

$$+ \underbrace{\mathbb{E}_{\{\theta^{(j)}\}_{j=1}^{J}|\{\mathbf{a}^{(n)}\}_{n=1}^{N}} \left[ \left\| \mathbb{E}_{\{\theta^{(j)}\}_{j=1}^{J}|\{\mathbf{a}^{(n)}\}_{n=1}^{N}}[g_{Ens}(\mathbf{a}^{(*)}; \{\theta^{(j)}\}_{j=1}^{J}] - g_{Ens}(\mathbf{a}^{(*)}; \{\theta^{(j)}\}_{j=1}^{J}) \right\|_2^2 \right]}_{\text{variance term}}. \tag{17}$$

*Proof.* Since all the expectations are taken with respect to the same randomness, their subscripts are dropped for notational ease. Also, let $\Theta^{(J)} = \{\theta^{(j)}\}_{j=1}^J$. We have

$$\mathbb{E}\left[\left\|\mathbf{a}^{(*)} - g_{\text{Ens}}(\mathbf{a}^{(*)};\Theta^{(J)})\right\|_2^2\right] \tag{18}$$

$$=\mathbb{E}\left[\left\|\mathbf{a}^{(*)} - \mathbb{E}[g_{\text{Ens}}(\mathbf{a}^{(*)};\Theta^{(J)})] + \mathbb{E}[g_{\text{Ens}}(\mathbf{a}^{(*)};\Theta^{(J)})] - g_{\text{Ens}}(\mathbf{a}^{(*)};\Theta^{(J)})\right\|_2^2\right] \tag{19}$$

$$=\mathbb{E}\left[\left\|\mathbf{a}^{(*)} - \mathbb{E}[g_{\text{Ens}}(\mathbf{a}^{(*)};\Theta^{(J)})]\right\|_2^2\right] \tag{20}$$

$$+ \mathbb{E}\left[\left\|\mathbb{E}[g_{\text{Ens}}(\mathbf{a}^{(*)};\Theta^{(J)})] - g_{\text{Ens}}(\mathbf{a}^{(*)};\Theta^{(J)})\right\|_2^2\right] \tag{21}$$

$$+ 2\mathbb{E}\left[\left(\mathbf{a}^{(*)} - \mathbb{E}[g_{\text{Ens}}(\mathbf{a}^{(*)};\Theta^{(J)})]\right)^\top \left(\mathbb{E}[g_{\text{Ens}}(\mathbf{a}^{(*)};\Theta^{(J)})] - g_{\text{Ens}}(\mathbf{a}^{(*)};\Theta^{(J)})\right)\right]. \tag{22}$$

Because $\mathbf{a}^{(*)}$ and $\mathbb{E}[g_{\text{Ens}}(\mathbf{a}^{(*)};\Theta^{(J)})]$ are constants with respect to the expectation, for (20) we have

$$\mathbb{E}\left[\left\|\mathbf{a}^{(*)} - \mathbb{E}[g_{\text{Ens}}(\mathbf{a}^{(*)};\Theta^{(J)})]\right\|_2^2\right] = \left\|\mathbf{a}^{(*)} - \mathbb{E}[g_{\text{Ens}}(\mathbf{a}^{(*)};\Theta^{(J)})]\right\|_2^2, \tag{23}$$

which is the stated bias term.

For the last term in (22), we have

$$\mathbb{E}\left[\left(\mathbf{a}^{(*)} - \mathbb{E}[g_{\text{Ens}}(\mathbf{a}^{(*)};\Theta^{(J)})]\right)^\top \left(\mathbb{E}[g_{\text{Ens}}(\mathbf{a}^{(*)};\Theta^{(J)})] - g_{\text{Ens}}(\mathbf{a}^{(*)};\Theta^{(J)})\right)\right] \tag{24}$$

$$= \left(\mathbf{a}^{(*)} - \mathbb{E}[g_{\text{Ens}}(\mathbf{a}^{(*)};\Theta^{(J)})]\right)^\top \left(\mathbb{E}[g_{\text{Ens}}(\mathbf{a}^{(*)};\Theta^{(J)})] - \mathbb{E}[g_{\text{Ens}}(\mathbf{a}^{(*)};\Theta^{(J)})]\right) = 0, \tag{25}$$

again because $\mathbf{a}^{(*)}$ and $\mathbb{E}[g_{\text{Ens}}(\mathbf{a}^{(*)};\Theta^{(J)})]$ are constants with respect to the expectation. Taken together, we have

$$\mathbb{E}\left[\left\|\mathbf{a}^{(*)} - g_{\text{Ens}}(\mathbf{a}^{(*)};\Theta^{(J)})\right\|_2^2\right] \tag{26}$$

$$= \left\|\mathbf{a}^{(*)} - \mathbb{E}[g_{\text{Ens}}(\mathbf{a}^{(*)};\Theta^{(J)})]\right\|_2^2 + \mathbb{E}\left[\left\|\mathbb{E}[g_{\text{Ens}}(\mathbf{a}^{(*)};\Theta^{(J)})] - g_{\text{Ens}}(\mathbf{a}^{(*)};\Theta^{(J)})\right\|_2^2\right], \tag{27}$$

where the first term is the stated bias term, and the second term is the stated variance term. $\quad\square$

We now show that naive bagging (Section 4.1) can reduce the reconstruction variance above. Formally, we have the following proposition.

**Proposition 2.** *Given a neural network activation $\mathbf{a}^{(*)}$ and the ensembled SAE $g_{NB}(\cdot;\{\theta^{(j)}\}_{j=1}^J)$ obtained through naive bagging trained on activations $\{\mathbf{a}^{(n)}\}_{n=1}^N$, the variance term in Lemma 1 goes to zero almost surely as $J \to \infty$.*

*Proof.* For notational ease, let $\mathbf{A} = \{\mathbf{a}^{(n)}\}_{n=1}^N$, and $\Theta^{(J)} = \{\theta^{(j)}\}_{j=1}^J$. By the definition of naive bagging, we have

$$g_{\text{NB}}(\mathbf{a}^{(*)};\Theta^{(J)}) = \frac{1}{J}\sum_{j=1}^J g(\mathbf{a}^{(*)};\theta^{(j)}). \tag{28}$$

It follows that the variance term in Lemma 1 can be written as

$$\mathbb{E}_{\Theta^{(J)}|\mathbf{A}}\left[\left\|\mathbb{E}_{\Theta^{(J)}|\mathbf{A}}\left[\frac{1}{J}\sum_{j=1}^J g(\mathbf{a}^{(*)};\theta^{(j)})\right] - \frac{1}{J}\sum_{j=1}^J g(\mathbf{a}^{(*)};\theta^{(j)})\right\|_2^2\right] \tag{29}$$

$$=\mathbb{E}_{\Theta^{(J)}|\mathbf{A}}\left[\left\|\mathbb{E}_{\theta|\mathbf{A}}[g(\mathbf{a}^{(*)};\theta)] - \frac{1}{J}\sum_{j=1}^J g(\mathbf{a}^{(*)};\theta^{(j)})\right\|_2\right], \tag{30}$$

where (30) follows from the linearity of expectation, and from the fact that $\theta^{(1)}, ..., \theta^{(J)}$ are identically distributed when conditioned on $\mathbf{A}$.

For practical neural networks and SAEs, we can assume that

$$\mathbb{E}_{\theta|\mathbf{A}} \left[ g_q(\mathbf{a}^{(*)}; \theta) \right] < \infty, \tag{31}$$

for each dimension $q \in [d]$. Furthermore, conditioned on $\mathbf{A}$, the trainings of $\theta^{(1)}, ..., \theta^{(J)}$ are identically and independently distributed. Therefore, we can apply the strong law of large numbers (Etemadi, 1981), obtaining

$$\frac{1}{J} \sum_{j=1}^{J} g_q(\mathbf{a}^{(*)}; \theta^{(j)}) = \mathbb{E}_{\theta|\mathbf{A}} \left[ g_q(\mathbf{a}^{(*)}; \theta) \right] \tag{32}$$

almost surely as $J \to \infty$. It then follows that

$$\frac{1}{J} \sum_{j=1}^{J} g(\mathbf{a}^{(*)}; \theta^{(j)}) = \mathbb{E}_{\theta|\mathbf{A}} \left[ g(\mathbf{a}^{(*)}; \theta) \right], \tag{33}$$

and for the variance term in Lemma 1:

$$\mathbb{E}_{\Theta^{(J)}|\mathbf{A}} \left[ \left\| \mathbb{E}_{\theta|\mathbf{A}}[g(\mathbf{a}^{(*)}; \theta)] - \frac{1}{J} \sum_{j=1}^{J} g(\mathbf{a}^{(*)}; \theta^{(j)}) \right\|_2^2 \right] \tag{34}$$

$$= \mathbb{E}_{\Theta^{(J)}|\mathbf{A}} \left[ \left\| \mathbb{E}_{\theta|\mathbf{A}}[g(\mathbf{a}^{(*)}; \theta)] - \mathbb{E}_{\theta|\mathbf{A}}[g(\mathbf{a}^{(*)}; \theta)] \right\|_2^2 \right] = 0, \tag{35}$$

almost surely as $J \to \infty$. $\qquad\square$

**Remark 2.** We note that all the expectations in the bias-variance decomposition in Lemma 1 are conditioned on the specific training set $\{\mathbf{a}^{(n)}\}_{n=1}^{N}$. This conditioning is needed for Proposition 2 to hold. Otherwise separate training runs of the SAE are dependent through the training set.

We now discuss the two assumptions needed for bounding the bias term in Lemma 1 for boosting (Section 4.2).

**Assumption 1.** *For a given neural network activation $\mathbf{a}^{(*)}$ and the ensembled SAE $g_{Boost}(\cdot; \{\theta^{(j)}\}_{j=1}^{J})$ obtained through boosting trained on the activations $\{\mathbf{a}^{(n)}\}_{n=1}^{N}$, we assume that*

$$\left\| \mathbf{a}^{(*)} - \mathbb{E}_{\{\theta^{(j)}\}_{j=1}^{J}|\{\mathbf{a}^{(n)}\}_{n=1}^{N}}[g_{Boost}(\mathbf{a}^{(*)}; \{\theta^{(j)}\}_{j=1}^{J})] \right\|_2^2 \tag{36}$$

$$\leq \frac{1}{N} \sum_{n=1}^{N} \left\| \mathbf{a}^{(n)} - \mathbb{E}_{\{\theta^{(j)}\}_{j=1}^{J}|\{\mathbf{a}^{(n)}\}_{n=1}^{N}}[g_{Boost}(\mathbf{a}^{(n)}; \{\theta^{(j)}\}_{j=1}^{J})] \right\|_2^2 + \varepsilon_G, \tag{37}$$

*for some constant $\varepsilon_G > 0$.*

**Remark 3.** Assumption 1 is essentially a generalization bound on the reconstruction performance for boosting. Intuitively, this assumption can hold because SAEs are regularized. However, note that this assumption can break down when $\mathbf{a}^{(*)}$ is much different from $\{\mathbf{a}^{(n)}\}_{n=1}^{N}$, which is a general pitfall for generalization bounds.

**Assumption 2.** *For the ensembled SAE $g_{Boost}(\cdot; \{\theta^{(j)}\}_{j=1}^{J})$ obtained through boosting trained on the activations $\{\mathbf{a}^{(n)}\}_{n=1}^{N}$, we assume that as $J \to \infty$,*

$$\frac{1}{N} \sum_{n=1}^{N} \left\| \mathbf{a}^{(n)} - \mathbb{E}_{\{\theta^{(j)}\}_{j=1}^{J}|\{\mathbf{a}^{(n)}\}_{n=1}^{N}}[g_{Boost}(\mathbf{a}^{(n)}; \{\theta^{(j)}\}_{j=1}^{J})] \right\|_2^2 \leq \varepsilon_I, \tag{38}$$

*for some constant $\varepsilon_I > 0$.*

**Remark 4.** Assumption 2 formalizes the intuition that boosting should be able to overfit almost perfectly to the training set. However, there is some irreducible error $\varepsilon_I$ because SAEs are simple and regularized models. This intuition is empirically verified in Supplementary Figure 1.

We now present the proposition showing that boosting with more iterations can lead to a bounded bias term in Lemma 1.

**Proposition 3.** *For a given neural network activation $\mathbf{a}^{(*)}$ and the ensembled SAE $g_{Boost}(\cdot; \{\theta^{(j)}\}_{j=1}^{J})$ obtained through boosting trained on the activations $\{\mathbf{a}^{(n)}\}_{n=1}^{N}$, under Assumption 1 and Assumption 2 we have, as $J \to \infty$,*

$$\left\| \mathbf{a}^{(*)} - \mathbb{E}_{\{\theta^{(j)}\}_{j=1}^{J} | \{\mathbf{a}^{(n)}\}_{n=1}^{N}} [g_{Boost}(\mathbf{a}^{(*)}; \{\theta^{(j)}\}_{j=1}^{J})] \right\|_2^2 \leq \varepsilon, \tag{39}$$

*for some constant $\varepsilon > 0$.*

*Proof.* The proof follows immediately under the assumptions. We have

$$\left\| \mathbf{a}^{(*)} - \mathbb{E}_{\{\theta^{(j)}\}_{j=1}^{J} | \{\mathbf{a}^{(n)}\}_{n=1}^{N}} [g_{\text{Boost}}(\mathbf{a}^{(*)}; \{\theta^{(j)}\}_{j=1}^{J})] \right\|_2^2 \tag{40}$$

$$\leq \frac{1}{N} \sum_{n=1}^{N} \left\| \mathbf{a}^{(n)} - \mathbb{E}_{\{\theta^{(j)}\}_{j=1}^{J} | \{\mathbf{a}^{(n)}\}_{n=1}^{N}} [g_{\text{Boost}}(\mathbf{a}^{(n)}; \{\theta^{(j)}\}_{j=1}^{J})] \right\|_2^2 + \varepsilon_G \tag{41}$$

$$\leq \varepsilon_I + \varepsilon_G, \tag{42}$$

where (41) uses Assumption 1, and (42) uses Assumption 2. Setting $\varepsilon = \varepsilon_I + \varepsilon_G$ completes the proof. $\square$

**Remark 5.** Proposition 3 is not surprising given Assumption 1 and Assumption 2. However, this formalization gives us insights about reasons why boosting may fail to reduce the bias term in the generalization region. That is, Assumption 1 or Assumption 2 may not hold (e.g. due to distribution shift or having too many constraints on the SAE, respectively).

**Remark 6.** Finally, we note that Proposition 2 and Proposition 3 are both asymptotic results with respect to the number of SAEs in the ensemble, primarily serving to motivate naive bagging and boosting from the perspective of the reconstruction error. Future work that relates reconstruction to the identifiability of human-interpretable features would be more directly useful for downstream interpretability tasks.

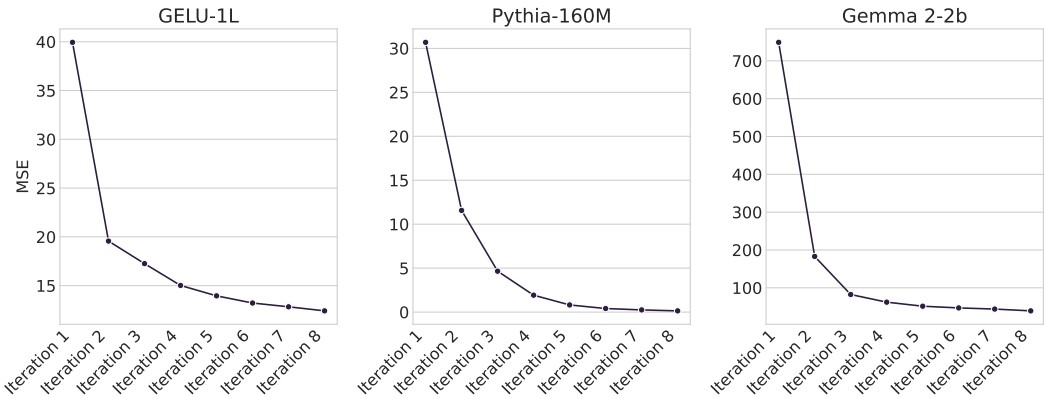

Supplementary Figure 1: MSE loss at the last training step for each iteration of a boosting ensemble with 8 SAEs. Reconstruction performance improves with each boosting iteration.

## B  EVALUATION METRICS

We evaluate our ensembling methods across six different evaluation metrics as described below. Here $N$ refers to the total number of per-token activations used for evaluation, and $m$ the total number of SAE features (e.g. $m = kJ$ for ensembled SAEs and $m = k$ for the base SAE).

**Reconstruction performance.** We use two standard metrics, mean squared error (MSE) and explained variance, to evaluate the reconstruction of activations:

$$\text{MSE} = \frac{1}{N} \sum_{n=1}^{N} \left\| \mathbf{a}^{(n)} - \hat{\mathbf{a}}^{(n)} \right\|_2^2, \text{ and}$$

$$\text{Explained Variance} = \frac{1}{d} \sum_{q=1}^{d} \left[ 1 - \frac{\sum_{n=1}^{N} (\mathbf{a}_q^{(n)} - \hat{\mathbf{a}}_q^{(n)})^2}{\sum_{n=1}^{N} (\mathbf{a}_q^{(n)} - \bar{\mathbf{a}}_q)^2} \right],$$

where $d$ is the activation dimensionality, and $\bar{\mathbf{a}}_q$ is the mean activation for the $q$th dimension.

**Relative sparsity.** Since SAEs with different number of features are compared (e.g., across different ensemble sizes), we use a measure of sparsity relative to the total number of features:

$$\text{Relative Sparsity} = \frac{1}{N} \sum_{n=1}^{N} \frac{\left\| \mathbf{c}^{(n)} \right\|_0}{m}.$$

**Diversity.** This metric counts the number of dissimilar features in an SAE in terms of the maximum cosine similarity:

$$\text{Diversity} = \sum_{i=1}^{m} \mathbb{1} \left[ \max_{i \neq j} |\langle \mathbf{f}_i, \mathbf{f}_j \rangle| \leq \tau \right],$$

where $\tau > 0$ is a threshold. Note that this metric does not depend on the evaluation tokens.

**Connectivity.** This metric, proposed in (Fel et al., 2025), measures the number of distinct pairs of SAE feature coefficients that are activated together across samples. It quantifies the diversity of the feature coefficients, with a high score indicating that a broad range of activations can be combined.

$$\text{Connectivity} = 1 - \left( \frac{1}{m^2} \left\| \mathbf{C}^\top \mathbf{C} \right\|_0 \right),$$

where $\mathbf{C} \in \mathbb{R}^{N \times m}$ is the matrix of feature coefficients across all samples, and here $\|\cdot\|_0$ counts the number of non-zero elements in a matrix.

**Stability.** This metric, adapted from (Paulo and Belrose, 2025), measures the maximum cosine similarity of the features that can be obtained across multiple runs of SAE training (with or without ensembling). Higher stability corresponds to the discovery of features that are similar across different runs. Note that this metric does not depend on the evaluation tokens. Given a total of $S$ training runs, the stability for the $s$th run is:

$$\text{Stability} = \frac{1}{m} \sum_{i=1}^{m} \max_{s' \in [S] \setminus s, j \in [m]} \langle \mathbf{f}_i^{(s)}, \mathbf{f}_j^{(s')} \rangle.$$

## C  DETAILS ON THE EXPANDED SAE BASELINE

For the expanded TopK SAE with Pythia-160M, we set K to the L0 norm of the ensembled SAEs. For SAE architectures without direct control over sparsity, we choose the sparsity achieved by boosting as the target L0. The sparsity of boosting instead of naive bagging is chosen for two reasons. Conceptually, naive bagging results in some redundant features, which contribute to higher L0 but do not reflect more diverse feature directions. Therefore, comparing with the L0 of boosting provides a more representative baseline when assessing whether the expanded SAE can match or exceed the performance of an ensemble. Empirically, we observe that it is impractical to obtain an L0 comparable to naive bagging. For JumpReLU SAEs with Gemma 2-2B, even a very small sparsity coefficient (1e-7) gives a lower L0 for the expanded SAE (around 3124) compared to the L0 of naive bagging (around 7648).

## D  ADDITIONAL DETAILS FOR THE SETUP OF THE SPURIOUS CORRELATION REMOVAL USE CASE

To achieve debiasing on a linear classifier $C_b$ that is trained on the biased set using the activations from a language model, a set of top $L$ SAE features is identified based on their probe attribution scores for a probe trained to predict the spurious signal (i.e. gender) (Karvonen et al., 2025). We use the same base SAE setup as the one used in the concept detection task – a ReLU SAE trained using Pythia-70M activations with 100 million tokens. Then, a modified classifier $C_m$ is trained after removing the spurious signal by zero-ablating the $L$ SAE features. The predictive performance of the modified classifier $C_m$ on profession for the held-out, balanced dataset indicates the SAE quality. Following Karvonen et al. (2025), the normalized evaluation score $S_{\text{SHIFT}}$ is defined as:

$$S_{\text{SHIFT}} = \frac{A_{\text{abl}} - A_{\text{base}}}{A_{\text{oracle}} - A_{\text{base}}},$$

where $A_{\text{abl}}$ is the accuracy for $C_m$, $A_{\text{base}}$ is the accuracy for $C_b$, and $A_{\text{oracle}}$ is the oracle accuracy with a classifier trained on a balanced dataset. It is worth noting that $A_{\text{base}}$ and $A_{\text{oracle}}$ do not depend on SAEs.

## E  RESULTS FOR GELU-1L AND PYTHIA-160M

Here we show the results for the intrinsic evaluations of GELU-1L (Supplementary Figure 2) and Pythia-160M (Supplementary Figure 3), with $\tau = 0.7$ for the diversity metric. Overall, the trend is similar to that of Gemma 2-2B; performance on most of the metrics improves as more SAEs are added to the ensemble, although it saturates for some of them around 8 SAEs. Also, boosting outperforms naive bagging in all metrics except for stability.

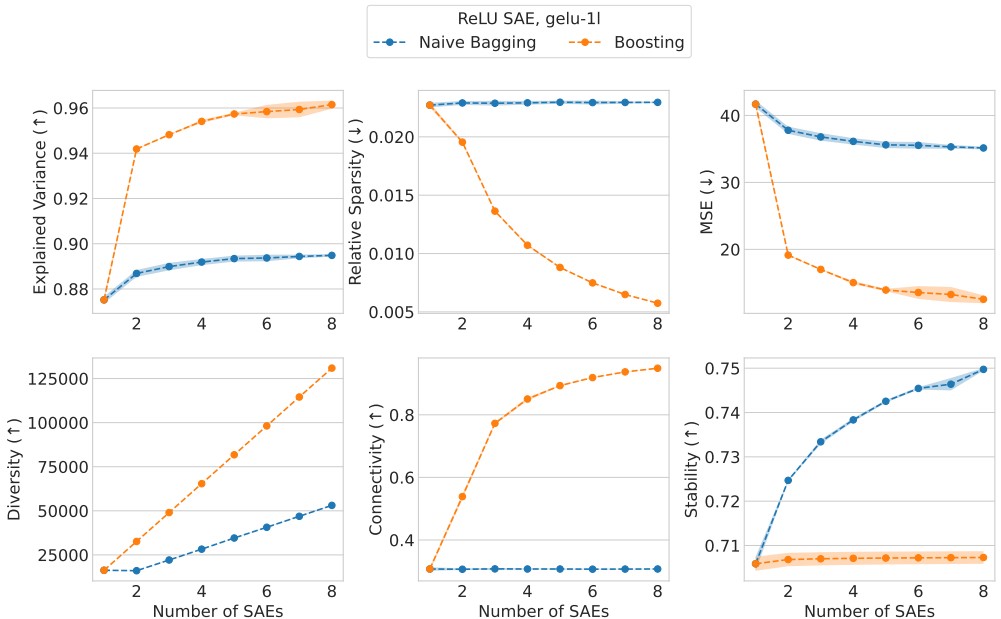

Supplementary Figure 2: Intrinsic evaluation of the ensembling approaches for GELU-1L. The shaded regions indicate 95% confidence intervals across 5 experiment runs.

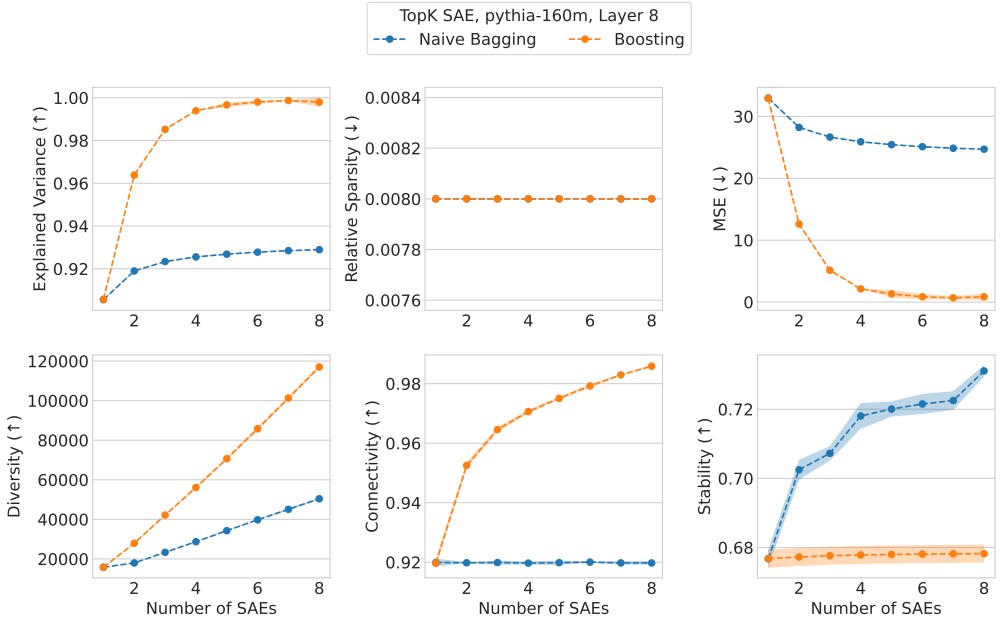

Supplementary Figure 3: Intrinsic evaluation of the ensembling approaches for Pythia-160M. The shaded regions indicate 95% confidence intervals across 5 experiment runs.

# F    VISUALIZING INTRINSIC EVALUATIONS

Here we provide bar plots for better visualization of the intrinsic evaluation. Supplementary Figures 4, 5, and 6 show the results for GeLU-1L, Pythia-160M, and Gemma 2-2B, respectively.

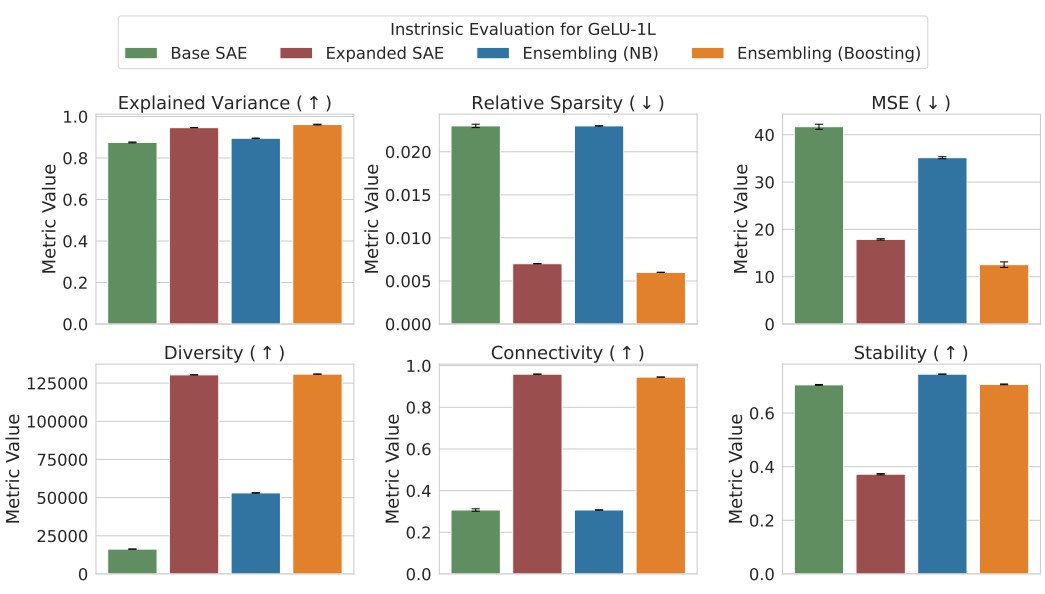

Supplementary Figure 4: Visualizing intrisinc evaluation for GeLU-1L. SAE ensembles consist of 8 SAEs. Means along with 95% confidence intervals are reported across 5 experiment runs.

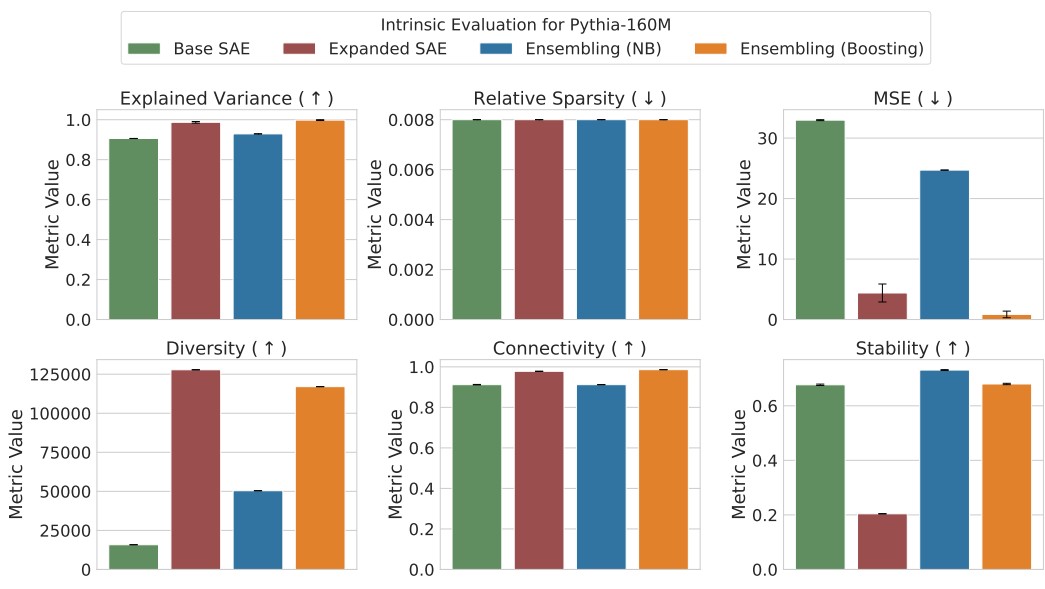

Supplementary Figure 5: Visualizing intrisinc evaluation for Pythia-160M. SAE ensembles consist of 8 SAEs. Means along with 95% confidence intervals are reported across 5 experiment runs.

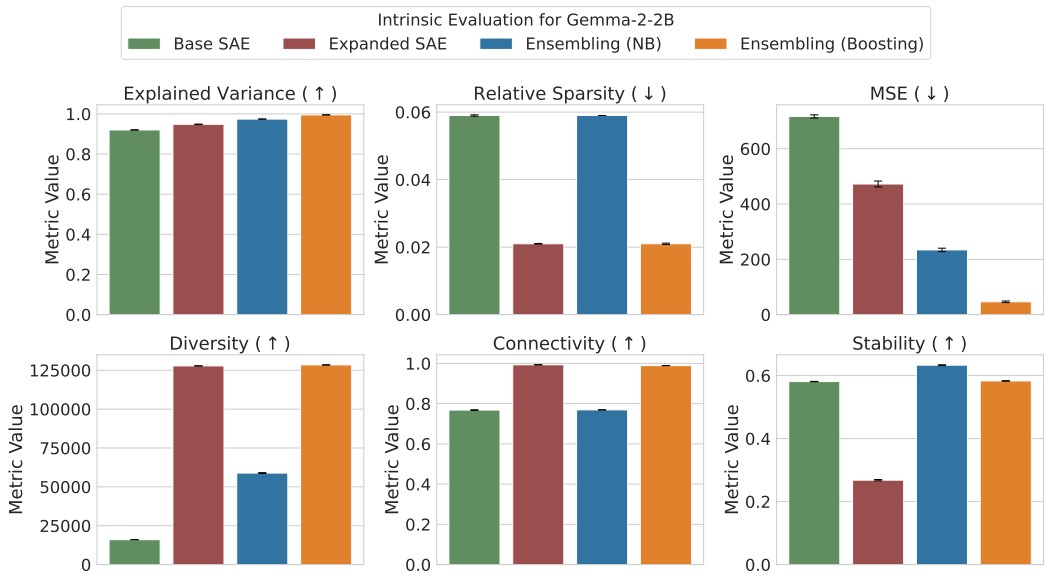

Supplementary Figure 6: Visualizing intrisinc evaluation for Gemma 2-2B. SAE ensembles consist of 8 SAEs. Means along with 95% confidence intervals are reported across 5 experiment runs.

# G  ADDITIONAL RESULTS FOR DOWNSTREAM USE CASES

Here we provide additional results for the downstream use cases (Section 5.3 and Section 5.4).

Supplementary Table 1 shows the test accuracy of a classifier trained using the top-5 concept-associated features identified by the ensembling methods across four tasks. The results are slightly different from those in Section 5.3, with boosting outperforming naive bagging and the base SAE for four out of the five tasks. This suggests that, while boosting does not identify the top feature, additional features from boosting can be selected to improve concept detection.

Supplementary Table 1: Test accuracy of the logistic regression classifier for the top-5 concept-associated feature across five concept detection tasks for Pythia-70M. SAE Ensembles consist of 8 SAEs. Means along with 95% confidence intervals are reported across 5 experiment runs.

|  | Amazon Review (Sentiment) | GitHub Code (Language) | AG News (Topic) | European Parliament (Language) |
| --- | --- | --- | --- | --- |
| Base SAE | 0.702 (0.015) | **0.805 (0.004)** | 0.851 (0.005) | 0.981 (0.003) |
| Expanded SAE | 0.703 (0.005) | 0.786 (0.012) | 0.862 (0.011) | 0.986 (0.001) |
| Ensembling (NB) | 0.689 (0.015) | 0.728 (0.005) | 0.783 (0.023) | 0.952 (0.004) |
| Ensembling (Boosting) | **0.708 (0.016)** | 0.795 (0.016) | **0.863 (0.008)** | **0.988 (0.000)** |

Supplementary Figure 7 shows the $S_{\mathrm{SHIFT}}$ scores for the spurious correlation removal task as the number of top gender-related features is varied. The trend is similar to what is observed in Section 5.4, with boosting outperforming naive bagging and the baselines for different numbers of ablated gender-related SAE features. The performance generally increases as the number of ablated features increases, indicating that there are multiple gender related features which are correctly identified by all the methods. This is especially worth noting for naive bagging, as increasing the number of ablated features might lead to all the redundant features related to the spurious signal getting ablated.

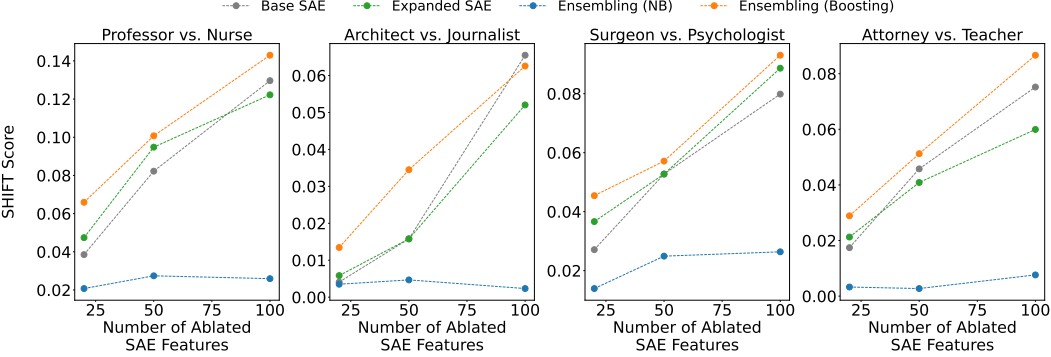

Supplementary Figure 7: $S_{\mathrm{SHIFT}}$ scores for the spurious correlation removal task vs. various numbers of top gender-related features identified across four pairs of professions for Pythia-70M. SAE ensembles consist of 8 SAEs. Means across 5 experiment runs are shown.

We further evaluate the downstream use cases for two of the language models used in the intrinsic evaluation, Pythia-160M and Gemma 2-2B. Supplementary Table 2 and Supplementary Figure 10 show the results of the concept detection task for the Pythia-160M model. Supplementary Table 3 and Supplementary Figure 11 show the results of the concept detection task for Gemma 2-2B. The results are consistent with those from the Pythia-70M model, with naive bagging outperforming the other methods on three out the four datasets. For Pythia-160M, expanded SAE performs better on the AG News (Topic) dataset while for Gemma 2-2B boosting performs better on the GitHub Code (Language) dataset.

Supplementary Table 4 and Supplementary Figure 12 show the $S_{\mathrm{SHIFT}}$ scores of the ensembling methods for the SCR task across four profession pairs for the Pythia-160M model. Supplementary

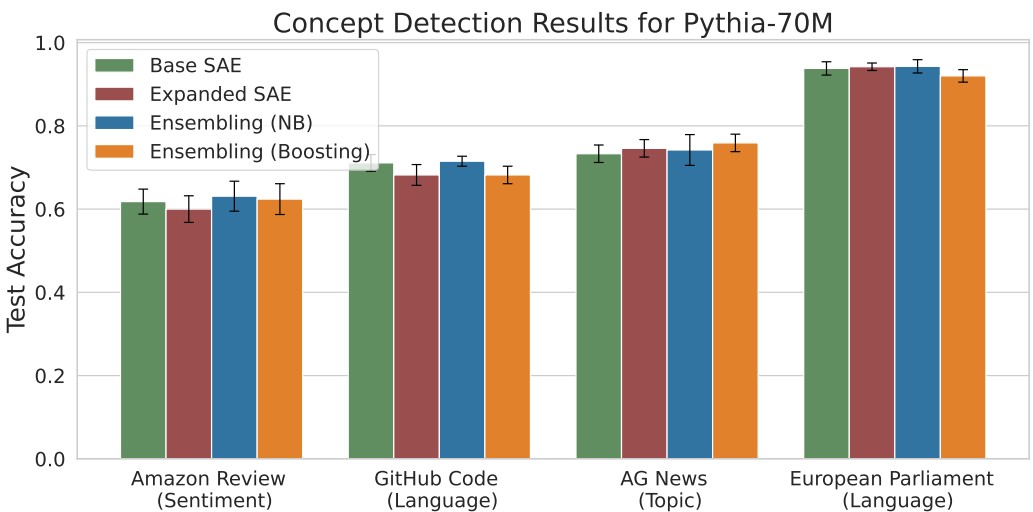

**Supplementary Figure 8:** Visualizing concept detection results for Pythia-70M. SAE Ensembles consist of 8 SAEs. Means along with 95% confidence intervals are reported across 5 experiment runs.

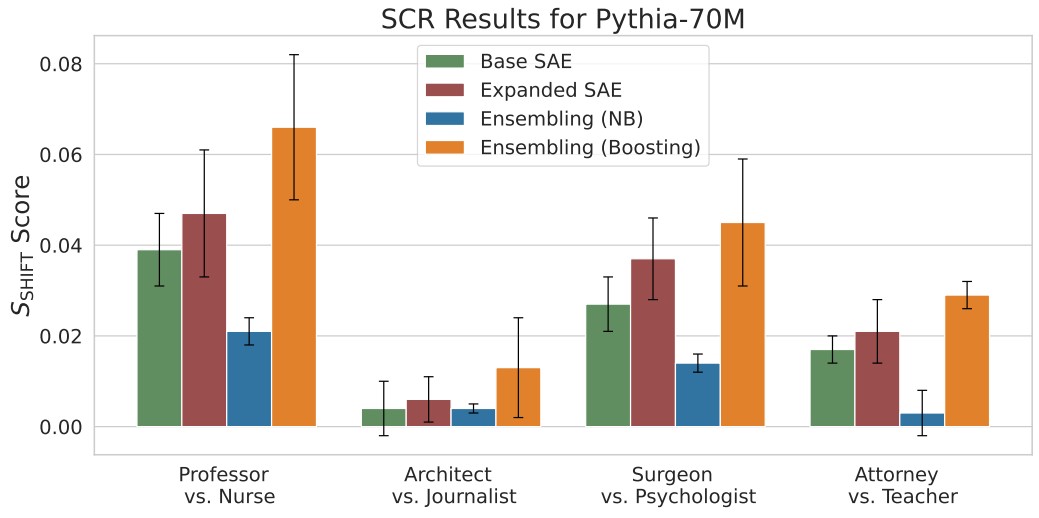

**Supplementary Figure 9:** Visualizing SCR results for Pythia-70M. SAE Ensembles consist of 8 SAEs. Means along with 95% confidence intervals are reported across 5 experiment runs.

Table 5 and Supplementary Figure 13 show the $S_{\text{SHIFT}}$ scores for Gemma 2-2B. Overall, the results are consistent with those obtained for the Pythia-70M model, with the boosted ensemble outperforming the other methods for three out of the four profession pairs. For Pythia-160M, naive bagging performs better on the Attorney vs. Teacher pair while for Gemma 2-2B naive bagging performs better on the Architect vs. Journalist pair. It is worth noting that some of the scores for Gemma 2-2B are negative which could happen when the probe accuracy after ablation is less than the baseline accuracy of the spurious probe before ablation.

Supplementary Table 2: Test accuracy of the logistic regression classifier for the top concept-associated feature across four concept detection tasks for Pythia-160M. SAE ensembles consist of 8 SAEs. Means along with 95% confidence intervals are reported across 5 experiment runs.

| | Amazon Review (Sentiment) | GitHub Code (Language) | AG News (Topic) | European Parliament (Language) |
|---|---|---|---|---|
| Base SAE | 0.663 (0.009) | 0.777 (0.032) | 0.810 (0.009) | 0.93 8 (0.012) |
| Expanded SAE | 0.636 (0.007) | 0.740 (0.059) | **0.826 (0.003)** | 0.942 (0.005) |
| Ensembling (NB) | **0.702 (0.027)** | **0.813 (0.002)** | 0.819 (0.012) | **0.945 (0.002)** |
| Ensembling (Boosting) | 0.663 (0.013) | 0.762 (0.029) | 0.798 (0.032) | 0.939 (0.008) |

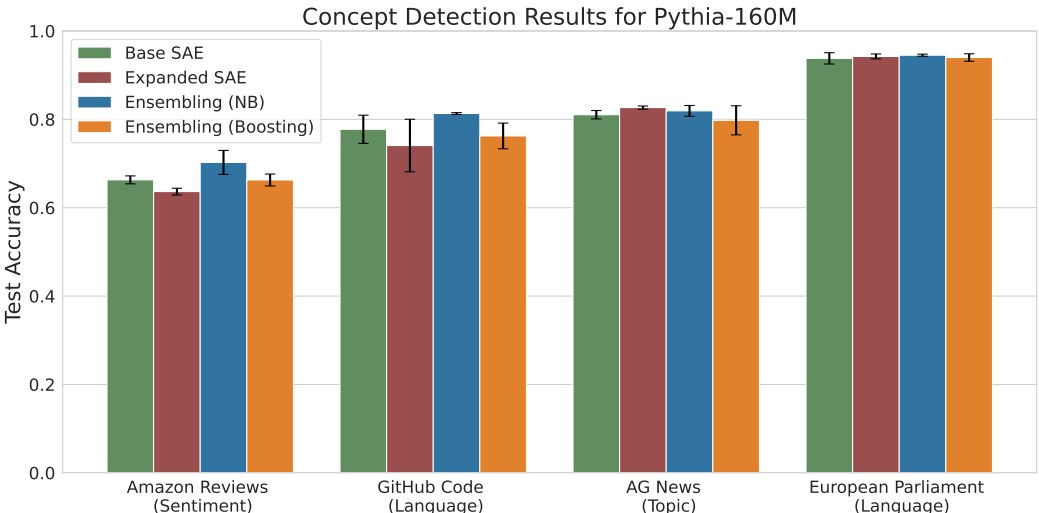

Supplementary Figure 10: Visualizing Concept Detection results for Pythia-160M. SAE Ensembles consist of 8 SAEs. Means along with 95% confidence intervals are reported across 5 experiment runs.

Supplementary Table 3: Test accuracy of the logistic regression classifier for the top concept-associated feature across four concept detection tasks for Gemma 2-2B. SAE ensembles consist of 8 SAEs. Means along with 95% confidence intervals are reported across 5 experiment runs.

| | Amazon Review (Sentiment) | GitHub Code (Language) | AG News (Topic) | European Parliament (Language) |
|---|---|---|---|---|
| Base SAE | 0.797 (0.058) | 0.666 (0.023) | 0.828 (0.027) | 0.835 (0.047) |
| Expanded SAE | 0.786 (0.091) | 0.676 (0.011) | 0.811 (0.026) | 0.738 (0.003) |
| Ensembling (NB) | **0.883 (0.029)** | 0.691 (0.005) | **0.863 (0.009)** | **0.879 (0.008)** |
| Ensembling (Boosting) | 0.816 (0.053) | **0.692 (0.004)** | 0.828 (0.039) | 0.867 (0.003) |

Supplementary Table 4: $S_{\text{SHIFT}}$ scores for the spurious correlation removal task with the top 20 gender-related features identified across four pairs of professions for Pythia-160M. SAE Ensembles consist of 8 SAEs. Means along with 95% confidence intervals are reported across 5 experiment runs.

| | Professor vs. Nurse | Architect vs. Journalist | Surgeon vs. Psychologist | Attorney vs. Teacher |
|---|---|---|---|---|
| Single SAE | 0.626 (0.134) | 0.485 (0.125) | 0.543 (0.262) | 0.318 (0.105) |
| Expanded SAE | 0.559 (0.102) | 0.703 (0.085) | 0.729 (0.080) | 0.358 (0.046) |
| Ensembling (NB) | 0.700 (0.006) | 0.770 (0.002) | 0.709 (0.006) | **0.414 (0.002)** |
| Ensembling (Boosting) | **0.778 (0.038)** | **0.802 (0.053)** | **0.746 (0.108)** | 0.341 (0.029) |

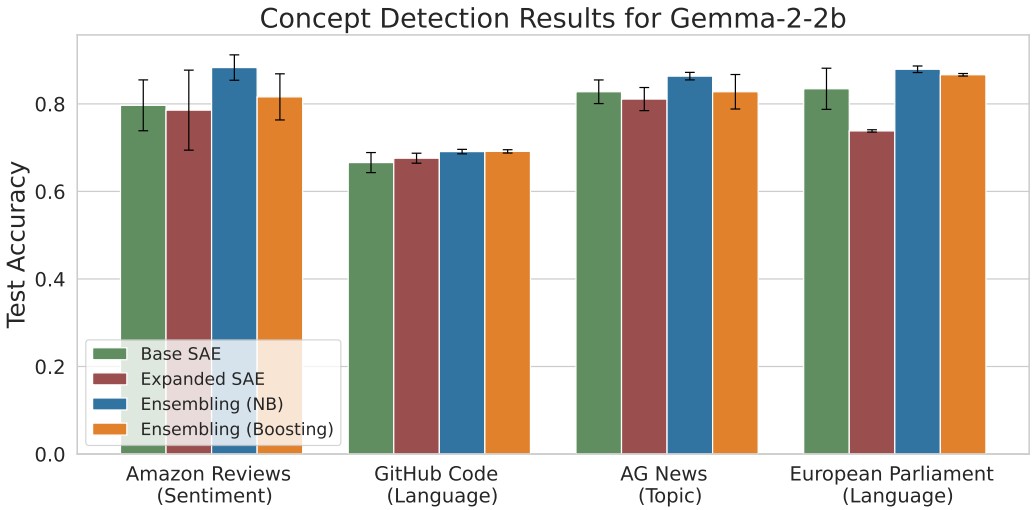

Supplementary Figure 11: Visualizing Concept Detection results for Gemma 2-2B. SAE Ensembles consist of 8 SAEs. Means along with 95% confidence intervals are reported across 5 experiment runs.

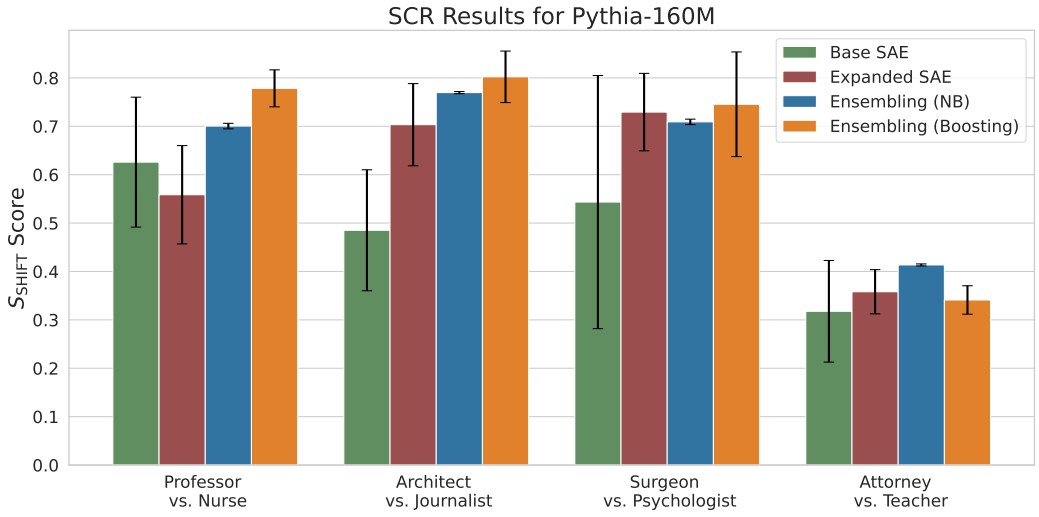

Supplementary Figure 12: Visualizing SCR results for Pythia-160M. SAE Ensembles consist of 8 SAEs. Means along with 95% confidence intervals are reported across 5 experiment runs.

Supplementary Table 5: $S_{\text{SHIFT}}$ scores for the spurious correlation removal task with the top 20 gender-related features identified across four pairs of professions for Gemma 2-2B. SAE Ensembles consist of 8 SAEs. Means along with 95% confidence intervals are reported across 5 experiment runs.

| | Professor vs. Nurse | Architect vs. Journalist | Surgeon vs. Psychologist | Attorney vs. Teacher |
|---|---|---|---|---|
| Base SAE | 0.406 (0.007) | -0.391 (0.122) | 0.206 (0.122) | -0.126 (0.141) |
| Expanded SAE | 0.154 (0.127) | -0.339 (0.177) | 0.291 (0.106) | -0.252 (0.269) |
| Ensembling (NB) | 0.415 (0.137) | **-0.259 (0.029)** | 0.121 (0.064) | -0.099 (0.061) |
| Ensembling (Boosting) | **0.594 (0.037)** | -0.528 (0.160) | **0.415 (0.177)** | **0.237 (0.076)** |

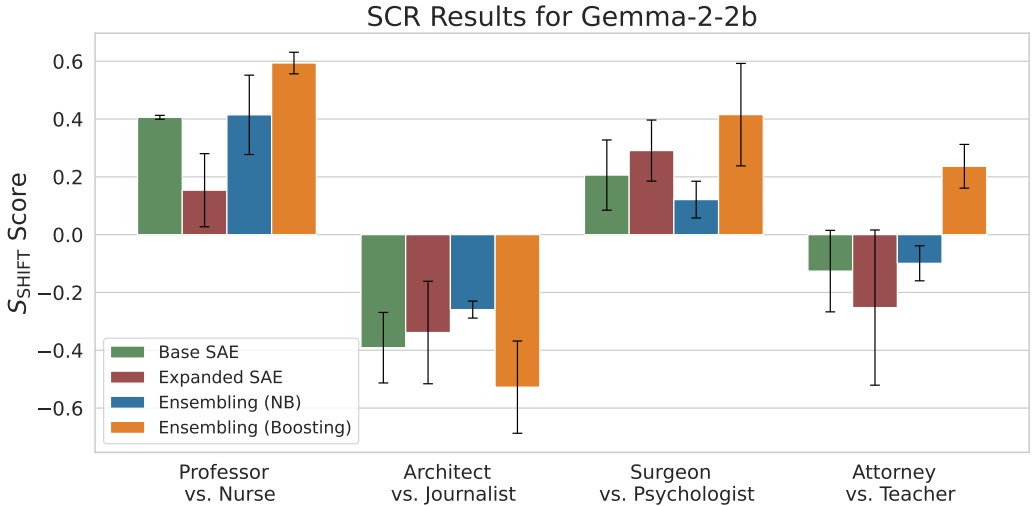

Supplementary Figure 13: Visualizing SCR results for Gemma 2-2B. SAE Ensembles consist of 8 SAEs. Means along with 95% confidence intervals are reported across 5 experiment runs.

## H    EVALUATION ON OTHER SAEBENCH METRICS

Previously, we evaluated our ensembling approaches on two of the metrics implemented in SAEBench (Karvonen et al. (2025)): concept detection and spurious correlation removal. Here we provide an extensive evaluation on the other metrics implemented in SAEBench for the multi-layer models Pythia-70M, Pythia-160M, and Gemma 2-2B. We don't include the unlearning metric since that metric is specifically for instruct models and none of our models are instruction tuned. For all the metrics, we use the default hyperparameter values provided in the code repository. For the feature absorption metric, the code repository suggests that this evaluation only makes sense if the LLM is large enough to have decent spelling knowledge and it is not recommended running this evaluation on LLMs with less than 1B parameters. Further, for the SAEs trained using Pythia-160M with our selected hyperparameters, the absorption could not be calculated due to insufficient first-letter features being detected. As a result, we run the feature absorption evaluation only on Gemma 2-2B.

Supplementary Table 6 and Supplementary Figure 14 show the results for SAEBench evaluation using SAEs trained on Pythia-70M. Ensembling approaches (specifically boosting) are able to perform slightly better than both the base SAE and the expanded SAE across all metrics in terms of the mean performance across five evaluation runs. The naive bagging approach doesn't perform well on the Targeted Probe Perturbation (TPP) task, which could be because this task is conceptually similar to Spurious Correlation Removal and the results are consistent with what was observed in that evaluation with naive bagging not performing well.

Supplementary Table 7 and Supplementary Figure 15 show the results for SAEBench evaluation using SAEs trained on Pythia-160M. Ensembling approaches (either naive bagging or boosting) perform better than the base SAE and the expanded SAE on three out of the four metrics in terms of the mean performance across five evaluation runs. The expanded SAE performs better on the TPP metric, while a similar performance drop is observed for naive bagging.

Supplementary Table 8 and Supplementary Figure 16 show the results for SAEBench evaluation using SAEs trained on Gemma 2-2B. Ensembling approaches (either naive bagging or boosting) perform slightly better than the base SAE and the expanded SAE on four out of the five metrics in terms of the mean performance across five evaluation runs. The expanded SAE performs better on the TPP metric, while a similar performance drop is observed for naive bagging.

Overall, our ensembling approaches are able to outperform or perform similarly with the baselines for most of the SAEBench metrics. This highlights that the performance improvements gained by ensembling are not limited to the concept detection and the SCR use cases, but can be generalized to multiple downstream tasks.

Supplementary Table 6: Evaluation of the ensembling approaches on different SAEBench metrics for Pythia-70M. SAE Ensembles consist of 8 SAEs. Means along with 95% confidence intervals are reported across 5 evaluation runs.

|  | RAVEL Score (↑) | AutoInterp Score (↑) | CE Loss Score (↑) | TPP (↑) |
|---|---|---|---|---|
| Base SAE | 0.3078 (0.0003) | 0.6902 (0.1281) | 0.9822 (0.0012) | 0.0164 (0.0015) |
| Expanded SAE | 0.3077 (0.0001) | 0.7140 (0.0512) | 0.9845 (0.0016) | 0.0147 (0.0008) |
| Ensembling (NB) | 0.3078 (0.0000) | 0.7382 (0.1470) | 0.9949 (0.0003) | 0.0075 (0.0004) |
| Ensembling (Boosting) | **0.3080 (0.0001)** | **0.8573 (0.1021)** | **0.9974 (0.0002)** | **0.0187 (0.0005)** |

Supplementary Table 7: Evaluation of the ensembling approaches on different SAEBench metrics for Pythia-160M. SAE Ensembles consist of 8 SAEs. Means along with 95% confidence intervals are reported across 5 evaluation runs.

|  | RAVEL Score (↑) | AutoInterp Score (↑) | CE Loss Score (↑) | TPP (↑) |
|---|---|---|---|---|
| Base SAE | 0.4969 (0.0010) | 0.8403 (0.0033) | 0.9771 (0.0004) | 0.2635 (0.0149) |
| Expanded SAE | 0.4970 (0.0010) | 0.8523 (0.0060) | 0.9783 (0.0001) | **0.2886 (0.0138)** |
| Ensembling (NB) | 0.4977 (0.0008) | **0.8567 (0.0045)** | 0.9829 (0.0000) | 0.0163 (0.0012) |
| Ensembling (Boosting) | **0.5020 (0.0008)** | 0.8524 (0.0182) | **1.0000 (0.0000)** | 0.2701 (0.0128) |

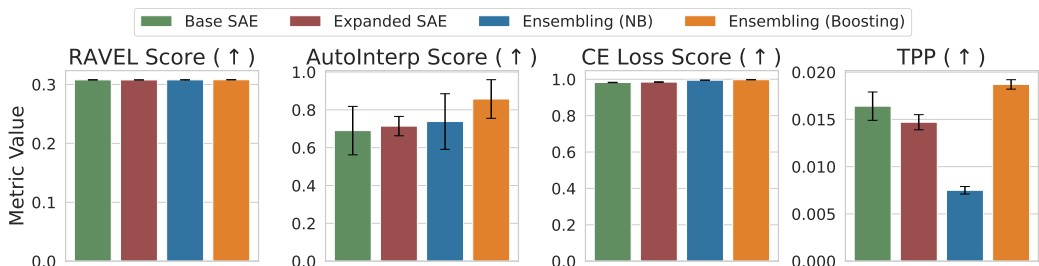

Supplementary Figure 14: Visualizing SAEBench evaluation for Pythia-70M. SAE Ensembles consist of 8 SAEs. Means along with 95% confidence intervals are reported across 5 experiment runs.

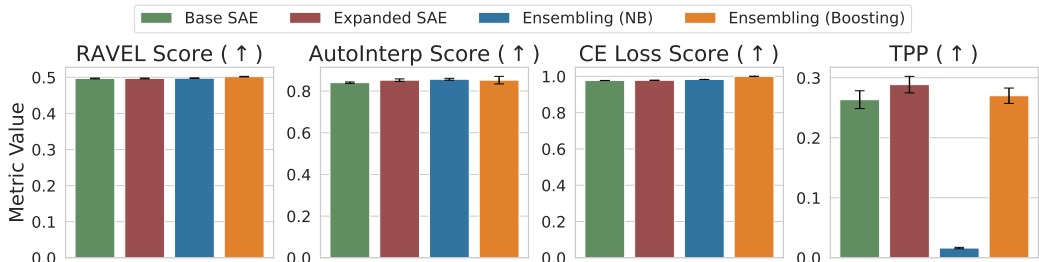

Supplementary Figure 15: Visualizing SAEBench evaluation for Pythia-160M. SAE Ensembles consist of 8 SAEs. Means along with 95% confidence intervals are reported across 5 experiment runs.

Supplementary Table 8: Evaluation of the ensembling approaches on different SAEBench metrics for Gemma 2-2B. SAE Ensembles consist of 8 SAEs. Means along with 95% confidence intervals are reported across 5 evaluation runs.

| | RAVEL Score (↑) | AutoInterp Score (↑) | CE Loss Score (↑) | TPP (↑) | Mean Full Absorption Rate (↓) |
|---|---|---|---|---|---|
| Base SAE | 0.7355 (0.0021) | 0.8030 (0.0064) | 0.9780 (0.0004) | 0.0958 (0.0162) | 0.0060 (0.0001) |
| Expanded SAE | 0.7468 (0.0042) | 0.8056 (0.0027) | 0.9780 (0.0002) | **0.1377 (0.0102)** | 0.0023 (0.0007) |
| Ensembling (NB) | **0.7625 (0.0010)** | 0.7992 (0.0082) | 0.9997 (0.0001) | 0.0162 (0.0008) | 0.0039 (0.0002) |
| Ensembling (Boosting) | 0.7520 (0.0002) | **0.8143 (0.0202)** | **1.0000 (0.0000)** | 0.1168 (0.0070) | **0.0020 (0.0010)** |

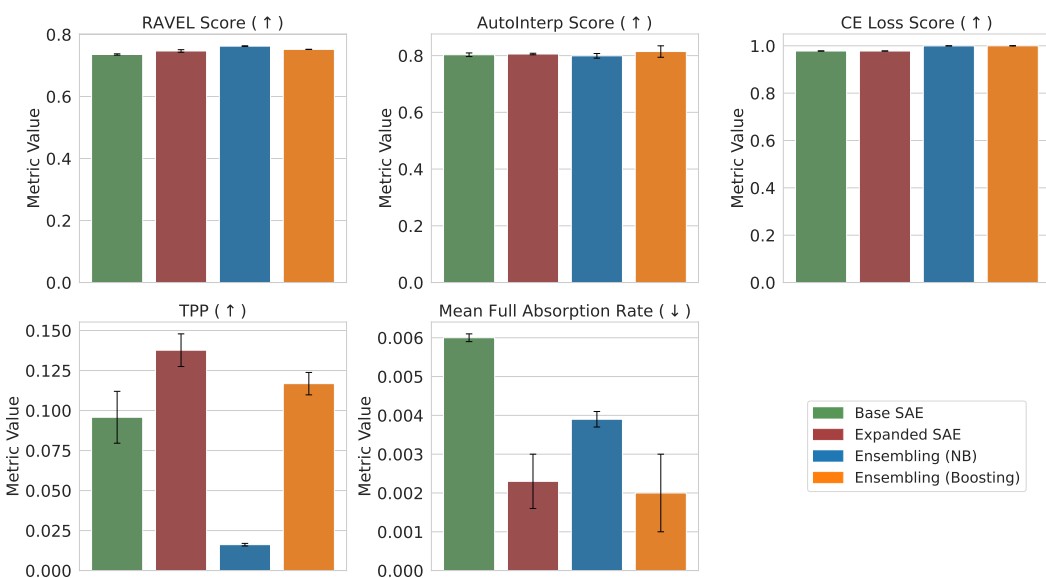

Supplementary Figure 16: Visualizing SAEBench evaluation for Gemma 2-2B. SAE Ensembles consist of 8 SAEs. Means along with 95% confidence intervals are reported across 5 experiment runs.

# I    IMPLEMENTATION DETAILS

Here we provide additional details about the data, compute, and hyperparameter selection.

## I.1    DATASET AND MODELS

The Pile dataset (Gao et al., 2020) (with copyrighted contents removed) used for training the SAEs is a large, diverse, and open-source English text dataset curated specifically for training general-purpose language models. Its diverse components include academic papers (e.g., arXiv, PubMed Central), books (e.g., Books3, BookCorpus2), code (from GitHub), web content (e.g., a filtered version of Common Crawl called Pile-CC, OpenWebText2), and other sources like Wikipedia, Stack Exchange, and subtitles. Beyond training, the Pile also serves as a benchmark for evaluating language models. More recently, the Pile has become the standard dataset for training sparse autoencoders (Bussmann et al., 2025; Cunningham et al., 2023; Lieberum et al., 2024; Marks et al., 2024; Paulo and Belrose, 2025).

All the language models we use have been previously used for training and evaluating sparse autoencoders (Bricken et al., 2023; Gao et al., 2024; Lieberum et al., 2024; Paulo and Belrose, 2025). Supplementary Table 9 provides additional details on the language models and the corresponding SAE architectures.

Supplementary Table 9: Overview of the language models and SAE architectures used for intrinsic evaluation and downstream use cases.

| Language Model | Num. Params | Num. Layers | Context Size | Activation Dimension | Layer Used | SAE Arch. |
|---|---|---|---|---|---|---|
| **Intrinsic Evaluation** | | | | | | |
| GELU-1L | 3.1M | 1 | 1024 | 512 | 1 | ReLU |
| Pythia-160M | 162.3M | 12 | 2048 | 768 | 8 | TopK |
| Gemma 2-2B | 2.1B | 26 | 8192 | 2304 | 12 | JumpReLU |
| **Downstream Use Cases** | | | | | | |
| Pythia-70M | 70.4M | 6 | 2048 | 512 | 4 | ReLU |

## I.2    TRAINING

Our ensembling algorithms are implemented in PyTorch[5] by adapting the SAELens library.[6] The pseudocode for boosting is summarized in Algorithm 1. For naive bagging, the training procedure for each SAE in the ensemble is the same as the standard SAE training. All the SAEs and the ensembles are trained on either an A100 GPU with 80GB of memory or an H100 NVL GPU with 93 GB of memory using a batch size of 10000. Supplementary Table 10 shows the time taken for a single experiment run on a single H100 GPU for ensembles with 8 SAEs. It is worth noting that naive bagging can be parallelized across multiple GPUs, bringing down the training time to that of the base SAE when the number of GPUs is equal to the number of SAEs in the ensemble. Supplementary Table 11 shows the inference times for the different SAEs, which is calculated by doing a forward pass through the trained SAE for 10 batches. The inference time for the ensembling approaches are marginally higher than the baselines, but they are essentially the same (less than 1 second) for all practical purposes.

---

[5] https://pytorch.org/
[6] https://github.com/jbloomAus/SAELens/tree/main

Supplementary Table 10: Training times for the base SAE, expanded SAE, and one experiment run for ensembles with 8 SAEs on a single H100 GPU.

|  | GELU-1L | Pythia-160M | Gemma 2-2B | Pythia-70M |
|---|---|---|---|---|
| Base SAE | 3h 2m | 5h 43m | 11h 7m | 21m |
| Expanded SAE | 19h 2m | 1d 5h 29m | 2d 11h 41m | 3h 48m |
| Ensembling (NB) | 1d 0h 16m | 1d 21h 44m | 3d 16h 56m | 3h 56m |
| Ensembling (Boosting) | 1d 8h 26m | 2d 0h 17m | 5d 5h 26m | 5h 35m |

Supplementary Table 11: Inference times (in seconds) for the base SAE, expanded SAE, and ensembling approaches with 8 SAEs on a single A100 GPU. The mean time along with 95% confidence intervals of a forward pass through the SAEs across 10 batches are reported.

|  | GELU-1L | Pythia-160M | Gemma 2-2B | Pythia-70M |
|---|---|---|---|---|
| Base SAE | 0.156 (0.209) | 0.178 (0.312) | 0.302 (0.503) | 0.105 (0.053) |
| Expanded SAE | 0.165 (0.223) | 0.184 (0.322) | 0.338 (0.551) | 0.170 (0.191) |
| Ensembling (NB) | 0.208 (0.212) | 0.549 (0.301) | 1.400 (0.515) | 0.223 (0.228) |
| Ensembling (Boosting) | 0.244 (0.422) | 0.889 (1.483) | 1.630 (0.910) | 0.249 (0.240) |

---

**Algorithm 1:** Training algorithm for the $j$th iteration of boosting. Gradient descent with a mini-batch size of 1 is shown as an illustration.

**Input:** Training activations $\{\mathbf{a}^{(n)}\}_{n=1}^{N}$, learning rate $\alpha$, sparsity coefficient $\lambda$, sparsity norm coefficient $p$, activation function $h(\cdot)$, previous SAEs $[g(\cdot; \theta^{(1)}), ..., g(\cdot; \theta^{(j-1)})]$

**Output:** Trained SAE $g(\cdot; \theta^{(j)})$

```
// Randomly initialize weights
```
initialize parameters $\theta^{(j)}$ (i.e. $\mathbf{W}_{\text{enc}}^{(j)}, \mathbf{b}_{\text{enc}}^{(j)}, \mathbf{W}_{\text{dec}}^{(j)}, \mathbf{b}_{\text{dec}}^{(j)}$)

initialize $n = 0$

**while** $n < N$ **do**

    `// Determine residual from previous SAEs`

    initialize $\boldsymbol{e} = \text{zeros\_like}(\mathbf{a}^{(n)})$

    **for** $\ell \in [j-1]$ **do**

        update $\boldsymbol{e} \leftarrow \boldsymbol{e} + g(\mathbf{a}^{(n)} - \boldsymbol{e}; \theta^{(\ell)})$

    **end**

    `// Leftover residual`

    set $\boldsymbol{r} = \mathbf{a}^{(n)} - \boldsymbol{e}$

    `// Determine predicted residual and feature coefficients`

    calculate $\hat{\boldsymbol{r}} = g(\boldsymbol{r}; \theta^{(j)})$

    calculate $\boldsymbol{c} = h(\mathbf{W}_{\text{enc}}^{(j)} \boldsymbol{r} + \mathbf{b}_{\text{enc}}^{(j)})$

    `// Calculate loss`

    set $\mathcal{L}_{\text{Boost}}\left(\mathbf{a}^{(n)}; \theta^{(j)}\right) = \|\mathbf{r} - \hat{\mathbf{r}}\|_2^2 + \lambda \|\boldsymbol{c}\|_p$

    `// Gradient step`

    update $\theta^{(j)} \leftarrow \theta^{(j)} - \alpha \nabla_{\theta^{(j)}} \mathcal{L}_{\text{Boost}}\left(\mathbf{a}^{(n)}; \theta^{(j)}\right)$

    `// update n`

    update $n \leftarrow n + 1$

**end**

---

### I.3 HYPERPARAMETER SELECTION

For the smallest model (GELU-1L), we conduct an extensive hyperparameter search across the learning rate, sparsity coefficient, and the expansion factor (Supplementary Figure 17), where the expansion factor refers to the multiplicative factor for the input activation dimensionality to get the

Supplementary Table 12: Selected hyperparameter values for the base SAE. These hyperparameters are held constant for all SAEs in the ensemble.

| Language Model | Learning Rate | Expansion Factor | TopK | Sparsity Coefficient |
|---|---|---|---|---|
| GELU-1L | 0.0003 | 32 | – | 0.75 |
| Pythia-160M | 0.0003 | 21 | 128 | – |
| Gemma 2-2B | 0.0003 | 7 | – | 0.75 |

SAE's hidden dimensionality ($k = d \times$ Expansion Factor). We select the hyperparameters that get closest to 90% explained variance while having the smallest L0 to ensure that the reconstructions are faithful to the original activations and the SAE decompositions are sparse.

For the larger Pythia-160M and Gemma 2-2B, we use the same learning rate from GELU-1L and consider expansion factors which give SAEs with a similar dimensionality ($k$) as the SAE for GELU-1L. We perform a sweep over the hyperparameter which controls the sparsity of the SAE (TopK value for Pythia-160M and the L0 coefficient for Gemma 2-2B) and select the values that give us an explained variance closest to 90% (Supplementary Figure 18). The final selected hyperparameter values are provided in Supplementary Table 12.

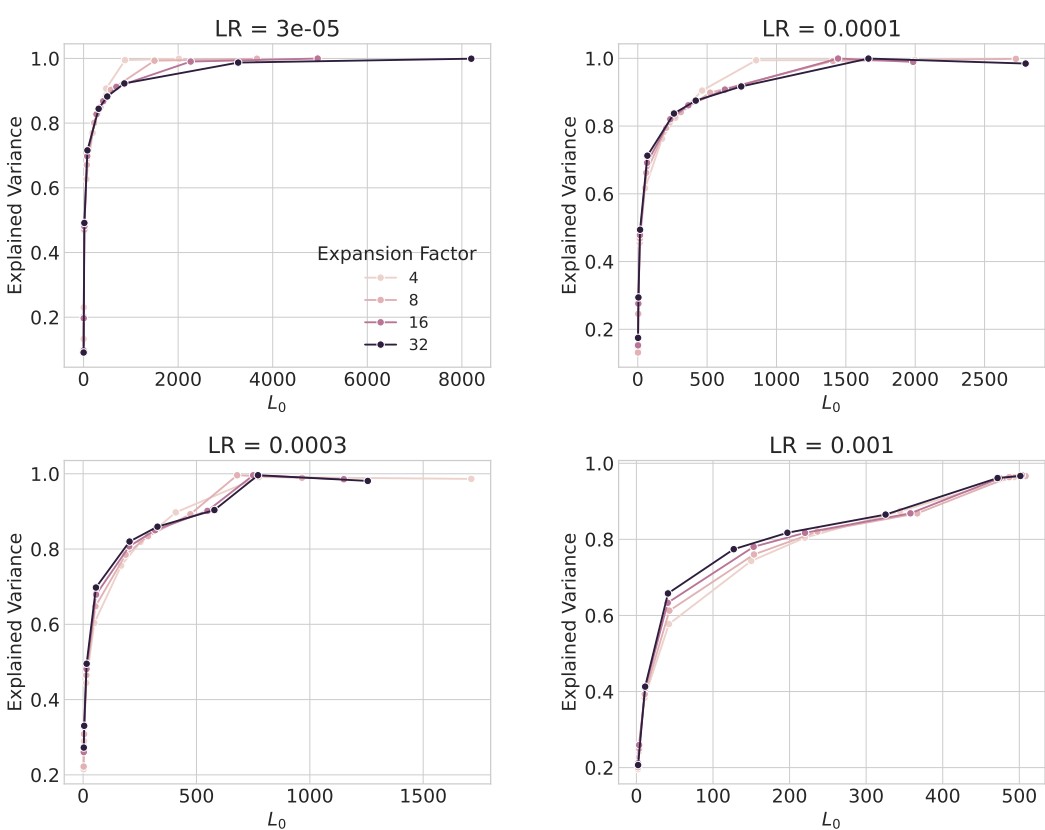

Supplementary Figure 17: Hyperparameter sweep performed for the GELU-1L activations with the ReLU SAE across different learning rates, expansion factors, and sparsity coefficients.

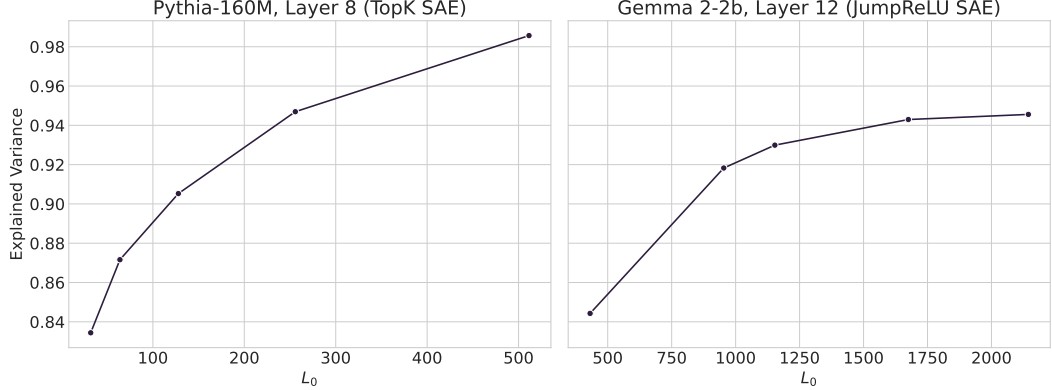

Supplementary Figure 18: Hyperparameter sweep performed for the Pythia-160M and Gemma 2-2B activations with the TopK and JumpReLU SAEs, respectively. For Pythia-160M, the sweep is across different values of $K$, and for Gemma 2-2B it is across different sparsity coefficients.

# J ADDITIONAL THRESHOLDS FOR FEATURE DIVERSITY

Here we show how the diversity metric changes for different thresholds $\tau$ (Supplementary Figures 19, 20, 21) with the number of SAEs in the ensemble across all three language models. Overall the trend remains the same as $\tau = 0.7$, with boosting learning a higher number of dissimilar features than naive bagging with each added SAE. Also, as expected, a smaller number of diverse features are learned for lower thresholds.

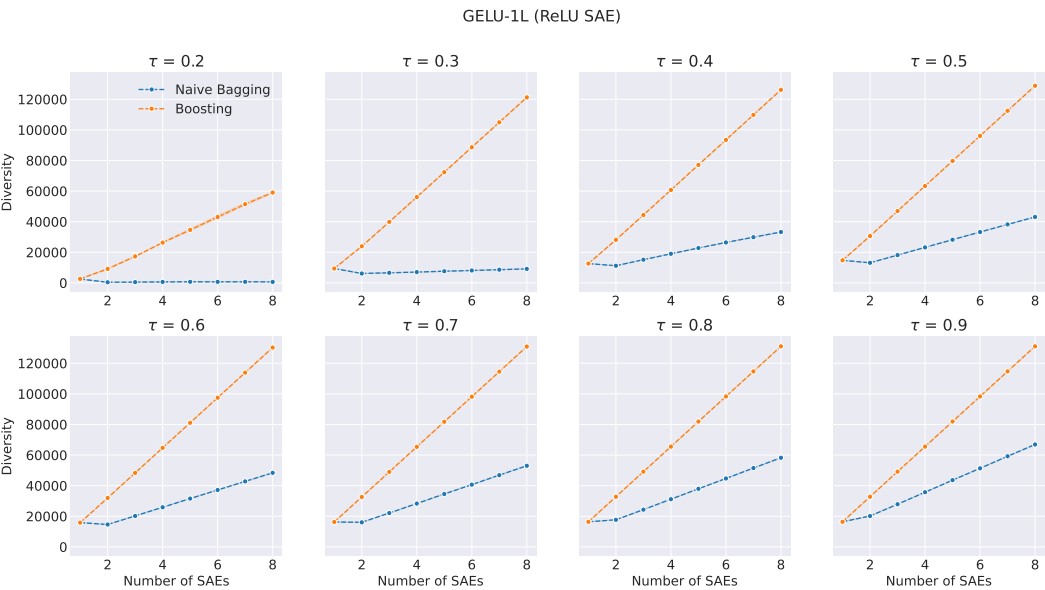

Supplementary Figure 19: Diversity metric evaluation for boosting and naive bagging across various similarity thresholds for GELU-1L. Shaded regions indicate 95% confidence intervals across 5 experiment runs.

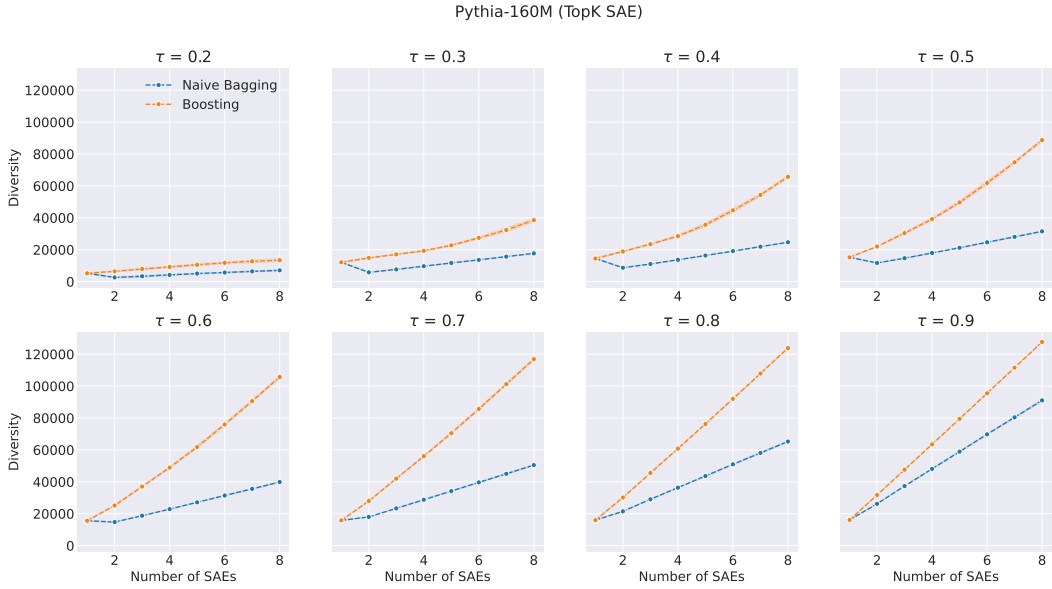

Supplementary Figure 20: Diversity metric evaluation for boosting and naive bagging across various similarity thresholds for Pythia-160M. Shaded regions indicate 95% confidence intervals across 5 experiment runs.

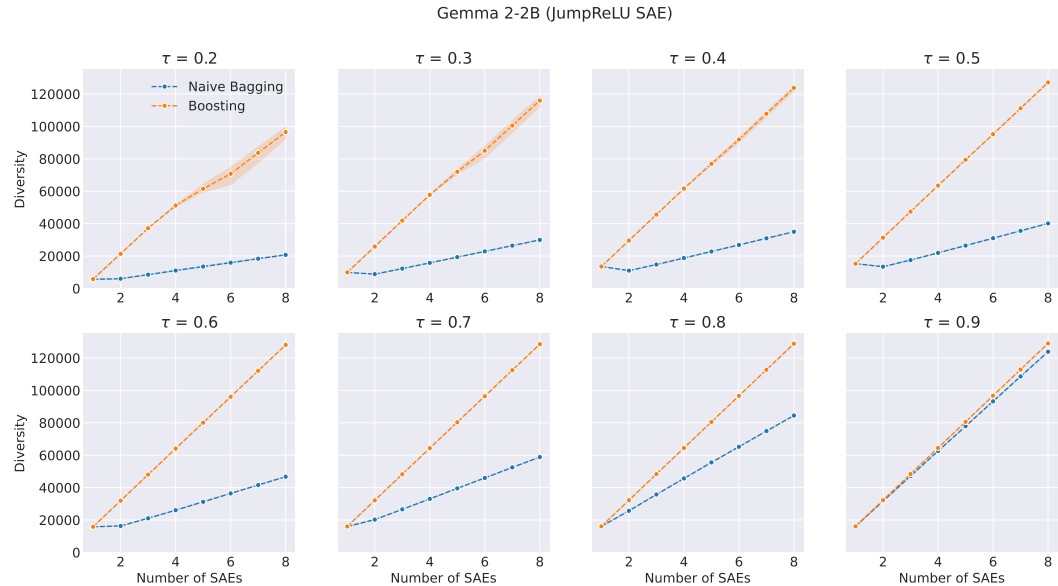

Supplementary Figure 21: Diversity metric evaluation for boosting and naive bagging across various similarity thresholds for Gemma 2-2B. Shaded regions indicate 95% confidence intervals across 5 experiment runs.

## K    ADDITIONAL RESULTS FOR THE EXPANDED SAE

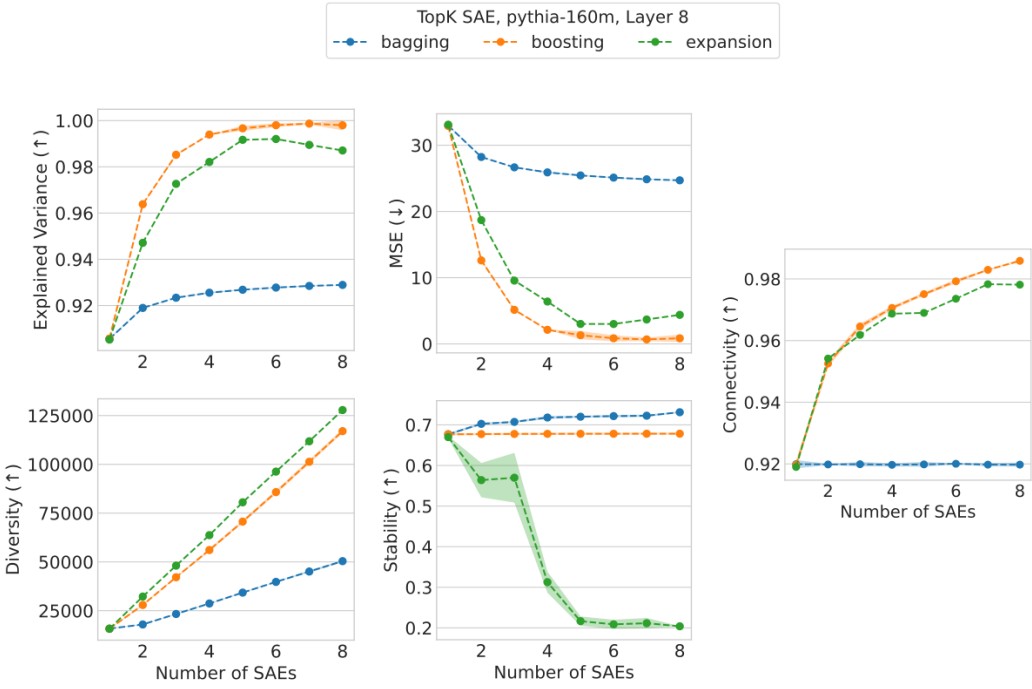

Supplementary Figure 22: Comparing SAE ensembles and the expanded SAE for Pythia-160M across different numbers of SAEs in the ensemble (corresponding to the matched dictionary sizes of the expanded SAE) for intrinsic evaluation. Shaded regions indicate 95% confidence intervals across 5 experiment runs.

