# OpenReview forum: "Ensembling Sparse Autoencoders"
_ICLR.cc/2026/Conference — Submitted to ICLR 2026_

### Official Review · Reviewer_i3SQ · 2025-10-26

**Soundness:** 2
**Presentation:** 4
**Contribution:** 3
**Rating:** 2
**Confidence:** 4

**Summary:**

This paper introduces two techniques for ensembling sparse autoencoders (SAEs): naive bagging (training many SAEs in parallel and then averaging their outputs) and boosting (sequentially training SAEs to reconstruct the error of the previous one). It then shows that ensembled SAEs score comparably to or better than appropriate baselines SAEs on various metrics adapted from prior literature.

**Strengths:**

1. The paper is very well-written.
2. The methods are well-motivated and clearly explained.
3. The authors compare against an appropriate baseline "expanded" SAE that was trained from scratch with the same width as (and comparable sparsity to) their ensembled SAEs.

**Weaknesses:**

I will discuss both weaknesses that I would like to see addressed to raise my score and various pieces of advice to the authors to improve presentation that do not affect my score.

Weaknesses that affect my score:
1. Looking at the quantitative results in table 2 (including the confidence intervals), I think those results should be summarized as ensembled SAEs performing comparably to (or perhaps slightly better than) the baselines. This would be clearer if the results were presented as a bar chart with error bars. To be clear, I don't think this weakens the results substantially; it's valuable to improve SAEs along one measure (SCR, which in my opinion is a more important measure of SAE quality) while not making them worse in other ways.
2. I find it concerning that the results in sections 5.3 and 5.4 use only one model (Pythia-70M), which is a different model from the three used in section 5.2. The authors write that they do this because Pythia-70M was used in prior work involving concept-level tasks, but this does not make sense: SAEs are an interpretability tool that is always meant to be applied to concept-level tasks. This choice also makes it difficult to compare to SAEBench; this is important because the SCR scores look very small overall (0.01-0.06 in this paper, compared to 0.2-0.4 in SAEBench). I request that the authors replicate these experiments on Gemma-2-2B and Pythia-160M; this should not be difficult because they have already trained the SAEs. This will allow greater confidence in the results (important given that the effect sizes are not very large) and allow for easy comparison with SAEBench.
3. Similarly, I request that the authors score their SAEs on the other metrics from SAEBench instead of only the SAEBench two metrics reported in section 5.3 and 5.4 and the less common intrinsic metrics in section 5.2. The point of this is ensure that the two metrics reported were not cherry-picked. To be clear, I'm not looking to see that ensembling improves on all metrics; it is okay if ensembled SAEs are comparable or even slightly worse on some of the metrics. But this provides useful context; e.g. it seems important to know if the boosted SAE's new features are less interpretable, according to autointerp score.
4. If I'm understanding correctly, the boosted SAE is not only trained sequentially, but also needs to be inferenced in multiple sequential steps. This should be emphasized, since it's a relatively fundamental change to the SAE architecture. It would also be good to see the results of training an SAE with this modified architecture end-to-end from scratch. (This is like the expanded SAE baseline, but for boosting.)
5. Since you're choosing the sparsity of the baseline SAE to be comparable with the ensembled ones, it's slightly misleading to use relative sparsity as a quality metric in table 1. That said, it's still important information to include in the table, so I think the table 1 caption should just clarify the situation.
6. Please report the number of runs that you used for computing stability.

If all of the above weaknesses are addressed and the new results don't substantially change the qualitative story then I will raise my score to an 8. I'll raise my score an intermediate amount if some are addressed. (2) and (3) are the most important.

Other points that don't affect my score:
1. It'd be good to put horizontal lines for the expanded SAE baseline in figure 2.
2. Many of the tables of numbers would be better presented as bar plots with error bars. If it's too many bars, then scores can be averaged across categories with detailed results in the appendix.
3. Typos. there is a dangling comma after the equation preceding line 180. The TopK nonlinearity is not actually an elementwise function. On line 882, "consine" should be "cosine."

**Questions:**

See weaknesses.

---

> ### Author Response · Authors · 2025-11-26
> **Response to review**
>
> We thank the reviewer for providing detailed feedback on our paper. Here we address the comments one by one. The updates in the revised manuscript are colored in olive green for easier identification.
>
> **Weaknesses**
>
> > 1. The quantitative results in Table 2 should be summarized better and presented as a bar chart with error bars
>
> We thank the reviewer for providing this helpful feedback. We agree that the results could be summarized better with a visualization. In the revised manuscript, we have updated the results section for the concept detection use case to reflect that ensembling does not substantially outperform the baselines. Additionally, we have added Supplementary Figure 8 to show the results as a bar chart with error bars. For completeness, we have also added Supplementary Figure 9 to visualize the results for the spurious correlation removal use case as a bar chart.
>
> > 2. The downstream use cases should be replicated on Pythia-160M and Gemma 2-2B
>
> We acknowledge the reviewer’s concern on evaluating the downstream use cases on additional models. Here we evaluate the downstream use cases with the SAEs on Pythia-160M and Gemma 2-2B trained in our experiments.
>
> Table R1 shows the results of the concept detection task for Pythia-160M, and Table R2 shows the results for Gemma 2-2B. The trends are consistent with those for Pythia-70M, with naive bagging outperforming the other methods on three out of the four datasets. For Pythia-160M, expanded SAE performs better on the AG News (Topic) dataset, while for Gemma 2-2B boosting performs better on the GitHub Code (Language) dataset. We have added these results in the revised manuscript along with the corresponding bar charts (Supplementary Table 2 and Supplementary Figure 10 for Pythia-160M; Supplementary Table 3 and Supplementary Figure 11 for Gemma 2-2B).
>
> Table R1. Test accuracy of the logistic regression classifier for the top concept-associated feature across four concept detection tasks for Pythia-160M. SAE ensembles consist of 8 SAEs. Means along with 95% confidence intervals are reported across 5 experiment runs.
> |             | **Amazon Review (Sentiment)** | **GitHub Code (Language)** | **AG News (Topic)** | **European Parliament (Language)** |
> |-|-|-|-|-|
> | Base SAE               | 0.663 (0.009)                 | 0.777 (0.032)               | 0.810 (0.009)        | 0.938 (0.012)                       |
> | Expanded SAE           | 0.636 (0.007)                 | 0.740 (0.059)               | **0.826 (0.003)**    | 0.942 (0.005)                       |
> | Ensembling (NB)        | **0.702 (0.027)**             | **0.813 (0.002)**           | 0.819 (0.012)        | **0.945 (0.002)**                   |
> | Ensembling (Boosting)  | 0.663 (0.013)                 | 0.762 (0.029)               | 0.798 (0.032)        | 0.939 (0.008)                       |
>
> Table R2. Test accuracy of the logistic regression classifier for the top concept-associated feature across four concept detection tasks for Gemma 2-2B. SAE ensembles consist of 8 SAEs. Means along with 95% confidence intervals are reported across 5 experiment runs.
> |            | **Amazon Review (Sentiment)** | **GitHub Code (Language)** | **AG News (Topic)** | **European Parliament (Language)** |
> |-|-|-|-|-|
> | Base SAE            | 0.797 (0.058)                 | 0.666 (0.023)               | 0.828 (0.027)        | 0.835 (0.047)                       |
> | Expanded SAE        | 0.786 (0.091)                 | 0.676 (0.011)               | 0.811 (0.026)        | 0.738 (0.003)                       |
> | Ensembling (NB) | **0.883 (0.029)**             | 0.691 (0.005)               | **0.863 (0.009)**    | **0.879 (0.008)**                   |
> | Ensembling (Boosting) | 0.816 (0.053)               | **0.692 (0.004)**           | 0.828 (0.039)        | 0.867 (0.003)
>
> Table R3 below shows the results of the SCR task for Pythia-160M, and Table R4 below shows the results for Gemma 2-2B. Overall, the trends are consistent with those obtained for Pythia-70M, with the boosted ensemble outperforming the other methods for three out of the four profession pairs. For Pythia-160M, naive bagging performs better on the Attorney vs. Teacher pair, while for Gemma 2-2B, naive bagging performs better on the Architect vs. Journalist pair. Some of the scores for the Gemma 2-2B model are negative, which could happen when the probe accuracy after ablation is less than the baseline accuracy of the spurious probe before ablation. The positive scores align with the scale in SAEBench. We have added these results in the revised manuscript along with the corresponding bar charts (Supplementary Table 4 and Supplementary Figure 12 for Pythia-160M; Supplementary Table 5 and Supplementary Figure 13 for Gemma 2-2b).

---

> ### Author Response · Authors · 2025-11-26
> **Response to review (continued)**
>
> Table R3. $S_\text{SHIFT}$ scores for the spurious correlation removal task with the top 20 gender-related features identified across four pairs of professions for Pythia-160M. SAE Ensembles consist of 8 SAEs. Means along with 95% confidence intervals are reported across 5 experiment runs.
> |                 | Professor vs Nurse | Architect vs Journalist | Surgeon vs Psychologist | Attorney vs Teacher |
> |-|-|-|-|-|
> | Base SAE            | 0.626 (0.134)      | 0.485 (0.125)            | 0.543 (0.262)            | 0.318 (0.105)        |
> | Expanded SAE          | 0.559 (0.102)      | 0.703 (0.085)            | 0.729 (0.080)            | 0.358 (0.046)        |
> | Ensembling (NB)       | 0.700 (0.006)      | 0.770 (0.002)            | 0.709 (0.006)            | **0.414 (0.002)**    |
> | Ensembling (Boosting) | **0.778 (0.038)**  | **0.802 (0.053)**        | **0.746 (0.108)**        | 0.341 (0.029)        |
>
>
> Table R4. $S_\text{SHIFT}$ scores for the spurious correlation removal task with the top 20 gender-related features identified across four pairs of professions for Gemma 2-2B. SAE Ensembles consist of 8 SAEs. Means along with 95% confidence intervals are reported across 5 experiment runs.
> |                 | Professor vs Nurse | Architect vs Journalist | Surgeon vs Psychologist | Attorney vs Teacher |
> |-|-|-|-|-|
> | Base SAE              | 0.406 (0.007)      | -0.391 (0.122)           | 0.206 (0.122)            | -0.126 (0.141)       |
> | Expanded SAE          | 0.154 (0.127)      | -0.339 (0.177)           | 0.291 (0.106)            | -0.252 (0.269)       |
> | Ensembling (NB)       | 0.415 (0.137)      | **-0.259 (0.029)**       | 0.121 (0.064)            | -0.099 (0.061)       |
> | Ensembling (Boosting) | **0.594 (0.037)**  | -0.528 (0.160)           | **0.415 (0.177)**        | **0.237 (0.076)**    |
>
> > 3. The SAEs should be evaluated on additional metrics from SAEBench
>
> We acknowledge the reviewer’s concern for limited evaluation of the SAEs on the SAEBench metrics. Here we provide an extensive evaluation on the other metrics implemented in SAEBench for the multi-layer models Pythia-70M, Pythia-160M, and Gemma 2-2B. We don't include the unlearning metric since that metric is specific to instruction-tuned models (as mentioned in the GitHub Repository: https://github.com/adamkarvonen/SAEBench/tree/main/sae_bench/evals/unlearning) and none of the models in our experiments are instruction tuned. For all the metrics, we use the default configurations provided in the code repository. For the feature absorption metric, the code repository (https://github.com/adamkarvonen/SAEBench/tree/main/sae_bench/evals/absorption) suggests that this evaluation only makes sense if the LLM is large enough to have decent spelling knowledge, and it is not recommended running this evaluation on LLMs with less than 1B parameters. Further, for the SAEs trained using Pythia-160M with our hyperparameters, the absorption could not be calculated due to insufficient first-letter features being detected. As a result, we run the feature absorption evaluation on SAEs trained for Gemma 2-2B.
>
> Table R5 below (Supplementary Table 6 and Supplementary Figure 14 in the revised manuscript) shows the results for SAEBench evaluation using SAEs trained on Pythia-70M. Ensembling approaches (specifically boosting) are able to perform slightly better than both the base SAE and the expanded SAE across all metrics in terms of the mean performance across five evaluation runs. The naive bagging approach doesn’t perform well on the targeted probe perturbation (TPP) task, which could be because this task is similar to spurious correlation removal and the results are consistent with what was observed in that evaluation with naive bagging not performing as well.
>
> Table R5. Evaluation of the ensembling approaches on different SAEBench metrics for Pythia-70M. SAE Ensembles consist of 8 SAEs. Means along with 95% confidence intervals are reported across 5 evaluation runs.
> |                 | RAVEL Score (↑)     | AutoInterp Score (↑)     | CE Loss Score (↑)        | TPP (↑)              |
> |-|-|-|-|-|
> | Base SAE              | 0.3078 (0.0003)      | 0.6902 (0.1281)           | 0.9822 (0.0012)           | 0.0164 (0.0015)       |
> | Expanded SAE          | 0.3077 (0.0001)      | 0.7140 (0.0512)           | 0.9845 (0.0016)           | 0.0147 (0.0008)       |
> | Ensembling (NB)       | 0.3078 (0.0000)      | 0.7382 (0.1470)           | 0.9949 (0.0003)           | 0.0075 (0.0004)       |
> | Ensembling (Boosting) | **0.3080 (0.0001)**  | **0.8573 (0.1021)**       | **0.9974 (0.0002)**       | **0.0187 (0.0005)**   |

---

> ### Author Response · Authors · 2025-11-26
> **Response to review (continued)**
>
> Table R6 below (Supplementary Table 7 and Supplementary Figure 15 in the revised manuscript) shows the results for SAEBench evaluation using SAEs trained on Pythia-160M. Ensembling approaches (either naive bagging or boosting) perform better than the base SAE and the expanded SAE on three out of the four metrics in terms of the mean performance across five evaluation runs. The expanded SAE performs better on the TPP metric, while a similar performance drop is observed for naive bagging.
>
> Table R6. Evaluation of the ensembling approaches on different SAEBench metrics for Pythia-160M. SAE Ensembles consist of 8 SAEs. Means along with 95% confidence intervals are reported across 5 evaluation runs.
> |                  | RAVEL Score (↑)       | AutoInterp Score (↑)        | CE Loss Score (↑)          | TPP (↑)                 |
> |-|-|-|-|-|
> | Base SAE              | 0.4969 (0.0010)        | 0.8403 (0.0033)               | 0.9771 (0.0004)             | 0.2635 (0.0149)          |
> | Expanded SAE          | 0.4970 (0.0010)        | 0.8523 (0.0060)               | 0.9783 (0.0001)             | **0.2886 (0.0138)**      |
> | Ensembling (NB)       | 0.4977 (0.0008)        | **0.8567 (0.0045)**           | 0.9829 (0.0000)             | 0.0163 (0.0012)          |
> | Ensembling (Boosting) | **0.5020 (0.0008)**    | 0.8524 (0.0182)               | **1.0000 (0.0000)**         | 0.2701 (0.0128)          |
>
>
> Table R7 below (Supplementary Table 8 and Supplementary Figure 16 in the revised manuscript) shows the results for SAEBench evaluation using SAEs trained on Gemma 2-2B. Ensembling approaches (either naive bagging or boosting) perform slightly better than the base SAE and the expanded SAE on four out of the five metrics in terms of the mean performance across five evaluation runs. The expanded SAE performs better on the TPP metric, while a similar performance drop is observed for naive bagging.
>
>
> Table R7. Evaluation of the ensembling approaches on different SAEBench metrics for Gemma 2-2B. SAE Ensembles consist of 8 SAEs. Means along with 95% confidence intervals are reported across 5 evaluation runs.
> |                  | RAVEL Score (↑)       | AutoInterp Score (↑)       | CE Loss Score (↑)          | TPP (↑)                  | Mean Full Absorption Rate (↓) |
> |-|-|-|-|-|-|
> | Base SAE              | 0.7355 (0.0021)      | 0.8030 (0.0064)            | 0.9780 (0.0004)           | 0.0958 (0.0162)          | 0.0060 (0.0001)               |
> | Expanded SAE          | 0.7468 (0.0042)      | 0.8056 (0.0027)            | 0.9780 (0.0002)           | **0.1377 (0.0102)**      | 0.0023 (0.0007)               |
> | Ensembling (NB)       | **0.7625 (0.0010)**  | 0.7992 (0.0082)            | 0.9997 (0.0001)           | 0.0162 (0.0008)          | 0.0039 (0.0002)               |
> | Ensembling (Boosting) | 0.7520 (0.0002)      | **0.8143 (0.0202)**        | **1.0000 (0.0000)**       | 0.1168 (0.0070)          | **0.0020 (0.0010)**           |
>
> Overall, our ensembling approaches are able to outperform or perform similarly with the baselines for most of the SAEBench metrics (including AutoInterp). This highlights that the performance improvements gained by ensembling are not limited to the concept detection and the SCR use cases, but can be generalized to multiple other downstream tasks.

---

> ### Author Response · Authors · 2025-11-26
> **Response to review (continued)**
>
> > 4. If I'm understanding correctly, the boosted SAE is not only trained sequentially, but also needs to be inferenced in multiple sequential steps. This should be emphasized, since it's a relatively fundamental change to the SAE architecture. It would also be good to see the results of training an SAE with this modified architecture end-to-end from scratch. (This is like the expanded SAE baseline, but for boosting.)
>
> Yes, the reviewer is correct in their understanding that the boosted SAE is inferenced in multiple sequential steps. We have updated the manuscript to emphasize this at the end of Section 4.2, which describes boosting.
>
> The reviewer also proposes a change to the training procedure of boosting, where all the boosted SAEs are trained end-to-end instead of sequentially. This is an interesting change, and we try this approach for training the boosted SAEs on two models: Gemma 2-2B (the largest model in our experiments) for the intrinsic evaluation and Pythia-70M for downstream evaluation, which is consistent with the experimental settings for our boosting approach.
>
> Table R8 below shows the evaluation results for end-to-end boosting with Gemma 2-2b on the intrinsic metrics. The end-to-end boosted SAEs are trained with the same hyperparameters as our sequential boosting, using JumpReLU SAEs with 8 SAEs in the ensemble. Our sequential boosting approach substantially outperforms end-to-end boosting across all the intrinsic metrics. The reconstruction error of the end-to-end boosted SAE is very high and the explained variance is negative, which indicates that the training is highly unstable and does not optimize well. One potential reason for this could be due to the vanishing gradient problem, since the gradients from the final loss must propagate through all SAEs in the end-to-end ensemble. Early SAEs can see their gradients multiplied through multiple layers which can make the gradients small.
>
> Table R8. Intrinsic evaluation metrics for the end-to-end boosting and sequential boosting approaches for Gemma-2-B (ensembling 8 SAEs). Means along with 95% confidence intervals are reported across 5 runs.
> |                 | Explained Variance   (↑)   | Relative Sparsity  (↓) | MSE   (↓)  | Diversity (↑) | Connectivity (↑)     | Stability (↑)  |
> |-|-|-|-|-|-|-|
> | Boosting (End-To-End)| -23.830 (15.67)         | 0.441 (0.0005)        | 215639.8 (15489.9)| 124013.0 (3.39)     | 0.218 (0.0056)     | 0.195 (0.0001)   |
> | Boosting (Ours)      | **0.995 (0.0003)**          | **0.021 (0.0002)**        | **46.538 (2.923)**    | **128415.6 (114.89)** | **0.989 (0.0003)**     | **0.583 (0.0009)**   |
>
> Table R9 below shows the results of the concept detection task using the end-to-end boosting approach for Pythia-70M. Our sequential boosting approach outperforms the end-to-end approach across all datasets.
>
> Table R9. Test accuracy of the logistic regression classifier for the top concept-associated feature across four concept detection tasks for Pythia-70M. SAE ensembles consist of 8 SAEs. Means along with 95% confidence intervals are reported across 5 experiment runs.
> |                | Amazon Reviews (Sentiment) | GitHub Code (Language) | AG News (Topic) | European Parliament (Language) |
> |-|-|-|-|-|
> | Boosting (End-To-End)| 0.603 (0.012) | 0.662 (0.004)            | 0.697 (0.018)     | 0.891 (0.002)                    |
> | Boosting (Ours)      | **0.624 (0.037)**               | **0.682 (0.021)**            | **0.759 (0.021)**     | **0.920 (0.015)**                    |
>
> Table R10 below shows the results of the SCR task using the end-to-end boosting approach for the Pythia-70M model. Our sequential boosting approach outperforms the end-to-end approach for all pairs of professions. The end-to-end approach gets negative scores for three out of the four profession pairs.
>
> Table R10. $S_\text{SHIFT}$ scores for the spurious correlation removal task with the top 20 gender-related features identified across four pairs of professions for Pythia-70M. SAE Ensembles consist of 8 SAEs. Means along with 95% confidence intervals are reported across 5 experiment runs.
> | | Professor vs. Nurse | Architect vs. Journalist | Surgeon vs. Psychologist | Attorney vs. Teacher |
> |-|-|-|-|-|
> | Boosting (End-To-End)| -0.241 (0.002) | 0.008 (0.012)  | -0.652 (0.003)| -0.185 (0.062) |
> | Boosting (Ours)| **0.066 (0.016)** | **0.013 (0.011)**| **0.045 (0.014)** | **0.029 (0.003)**  |
>
> Overall, across both intrinsic and downstream evaluations, we observe that the end-to-end boosting approach does not perform well. The training can be unstable, resulting in poor reconstructions of the original activations and the performance is even worse than the base SAE. It has low performance on the downstream tasks too and even though the concept detection scores are not too low, the poor reconstruction performance raises the question whether the end-to-end boosted SAE is reliable enough to use for any downstream application at all.

---

> ### Author Response · Authors · 2025-11-26
> **Response to review (continued)**
>
> > 5. The Table 1 caption should be updated to clarify the choice of relative sparsity
>
> We thank the reviewer for bringing up this important point about the use of relative sparsity as an intrinsic evaluation metric. We have updated the caption in Table 1 to clarify that the expanded SAE baseline is tuned to have a sparsity similar to that of the ensembled SAE.
>
> > 6. Please report the number of runs that you used for computing stability.
>
> Stability was computed across 5 runs, and the confidence intervals were obtained by considering each of the runs as the base run and calculating its stability with the others (similar to a cross validation setup). We have updated the metrics section (Section 5.2.2 in the revised manuscript to include details about the stability calculation.
>
> **Other points**
>
> > 1. It'd be good to put horizontal lines for the expanded SAE baseline in figure 2
>
> Here we note that Figure 2 presents an ablation study to investigate the impact of the number of SAEs on our ensemble methods, providing a sanity check for the theoretical justifications (as discussed in lines ~~299-311~~ 297-309 in the revised manuscript). We didn’t include a horizontal line for the expanded SAE since that SAE is tuned for a sparsity similar to that of 8 boosted SAEs, which is the last point on each plot.
>
> > 2. Many of the tables of numbers would be better presented as bar plots with error bars.
>
> We agree that bar plots with error bars can be added, but we choose to show the quantitative results as a table to make it easier to compare among different methods, especially when the performance between different methods is close. Nevertheless, we have updated the manuscript to include these visualizations in the Appendix (Supplementary Section F).
>
> > 3. Typos.
>
> We thank the reviewer for pointing these out. We have updated the manuscript to fix all of the mentioned typos.

---

### Official Review · Reviewer_9Yp1 · 2025-10-27

**Soundness:** 3
**Presentation:** 3
**Contribution:** 2
**Rating:** 4
**Confidence:** 3

**Summary:**

The authors take the finding from previous literature that often SAEs trained on the same data can learn different features and ask if they can leverage this diversity for better performance. The way they propose to do this is via classical ensembling methods from the supervised learning literature: boosting and bagging. They show that ensembling reconstructions is generally equivalent to ensembling features. With their methodology they show improved performance relative to the baseline of the expanded SAE they achieve better explained variance and MSE. This is an interesting and non-obvious result!
It seems like the bagging approach didn’t perform particularly well despite being intuitively the inspiration for the project. I would say that this likely means that the story given for why the approach works in terms of the variability of SAEs is probably not the best explanation for their good performance (it may slightly contribute but doesn’t seem to be the leading term). In fact, this implies that the reason that the approach works well is the ability to leverage information from the error terms of previous SAE reconstructions and use that as signal for training. It would be interesting to explore that motivation much more and how this connects to previous work in Compressed Sensing and Digital Signal Processing.

Overall a cool paper that has good empirical results, though possibly the link between the motivation and empirical results may not be as strong as it appears. The paper doesn’t necessarily solve a particular problem but nonetheless presents a possible improvement to the standard SAE training methodologies. It would be interesting to see if there are problems solved here (e.g. perhaps finding more finegrained features is solved with this method which could be evidenced by a feature identification study. I’d be open to increasing my score if some of the below comments are addressed.

**Strengths:**

- Clear written style
- Related work shows a good knowledge of the relevant literature
- Figure 1 is helpful in understanding the core technique
- The propositions and proofs are helpful in justifying the argument
- The mathematical framework is mostly clear (though the notation is non-standard; see below)
- Presents a useful way to improve the performance of SAEs.
- Shows results on two downstream tasks
- The evaluation uses multiple models, architectures and tasks/metrics (though see below for the disadvantages of not having clear ablations and where some of the metrics may be misleading)
- The framework is architecture agnostic which allows this approach to be useful for most SAE paradigms, there remains open opportunities for future work to improve on this

**Weaknesses:**

- Nits:
    - In Section 3.1 the activation functions and the citations are in different orders
    - Would suggest against using the letter k for the dimension of the SAE hidden layer as this conflicts with the k in the activation function for top k. Perhaps using f would be clearer.
- All of the notation section uses quite non-standard notation - it would be good to use similar notation to e.g. Gao et al or a similar paper for readability
- A Pareto plot with Sparsity or Description Length (see Ayonrinde et al 2024) on the x axis and the metrics on the y axis would be very valuable - currently it seems that there’s at least one set of hyperparameters for which the authors approach outperforms but its unclear if this represents an improvement in many different settings
- I find Figure 2 to be generally quite misleading: the appropriate baseline should be the expanded SAE rather than the base SAE in all cases as this is the parameter matched (and possibly inference time FLOP matched) version
- The metrics chosen seem to unfairly advantage the authors methods (especially in Figure 2)
    - For example relative sparsity is not the correct metric for this method as it gives an advantage to larger models by allowing them to have more features. In fact larger models should be able to achieve the same performance with fewer features (as shown in Bricken et al 2023 and many other papers) and so this gives large models a double advantage!
    - Similarity the diversity metric has the opposite problem, here we really should have a relative metric because of course a model with more features will have more dissimilar features (similarly for the connectivity metric).
    - This makes many of the metrics here quite misleading as vanity metrics.
- Using different SAEs on different models makes it hard to tell if the approach works differently on transformers of different architectures, of different sizes or using different SAE architectures - it would be useful to have a clear ablation on this point.
- I would suggest moving the setup of your Use Cases to the appendix and focusing on the results in this section. This would allow more space for detailed discussion and motivation and for you to add the statements of your propositions in the main paper rather than in the appendices.
- The main weakness of this work is that the motivation for the work doesn’t connect with what actually performed well - introduction motivates the bagging approach whereas in fact the boosting approach is what works in practise.
    - I think editing the framing substantially to account for this and explain why the boosting approach is a reasonable way forwards would be a large improvement for the coherence of this work.
- All of the metrics provided are purely mathematical and there are no metrics discussing the interpretability of the learned features either with AutoInterp (e.g. Paulo et al 2025) or with a human study (e.g. Bricken et al 2023). An AutoInterp study (perhaps using SAE-Bench as the authors are already using this), could be a good addition to the paper.

**Questions:**

- Remark 1 seems to say that c could be folded into either c or W_enc however the reasoning given for this is about W_dec rather than W_enc - could you clarify on this point?
- Would the naive bagging approach perform better if the ensemble weight $\alpha$ is allowed to be learned in finetuning rather than being restricted to 1/J?
    - More full finetuning of the bagging approach when ensembled (i.e. allowing the individual SAE weights to be updated as well as just the ensemble weighting) could also be interesting for reducing the feature redundancy problem.
    - Also note that Mudide et al gives a nice description of the feature redundancy problem which might be useful to leverage to make the argument given here
    - I'm also interested in if there are other approaches attempted to address the redundancy problem e.g. could you cluster and prune very similar feature directions?
-  Appendix C suggests that the L0 for the expanded SAE and similarly for the other approaches is 3-8k - is this correct? Normally L0 figures are <100. Is this multiplied by the batch dimension by mistake?
- What is the wall clock time difference in training and inference for the boosting and bagging approaches?
    - It seems that this is a considerable limitation and it would be good to quantify this.
    - Similarly for FLOPs.
- The explained variance metric suggested that after 4 SAEs ensemble there are significantly diminishing returns for the naive bagging case - do you agree with this intuition?
    - If so why use 8 SAEs for Table 1? I suspect there might be a better parameter/performance tradeoff using 4 SAEs
    - If you’re using comparing the 8 SAE to single SAE basline then for a proper comparison the MDL-SAE framework from Ayonrinde et al 2024 will be a metric that accounts for comparisons across SAE widths better than just sparsity which can be gamed with larger models. I’d be interested in a comparison using this metric
- I’m not totally convinced by the argument given for why NB excels in Table 2 - it seems like there should be some features which are useful in the NB case and which are learned by the Boosting model because they’re useful regardless of hierarchy?
- How does this approach differ from Matryoshka SAEs (Bussman et al 2025)? This seems like the closest work and is not compared against. Similarly, Switch SAEs (Mudide et al 2025) is also not compared against which similarly seeks to expand SAEs (in their case in a MoE style)
- It would be useful to have a study showing feature identification here - a core question would be “in which cases do the approaches learn redundant or hyper-specific features that make the approach worse than the baselines?”
- How many dead features are there in the expanded SAE baseline and what techniques were used to overcome dead features? Was the auxiliary loss from Gao et al 2024 used here? It seems possible that one of the reasons that the expanded SAE has poor performance is due to dead features.

---

> ### Author Response · Authors · 2025-11-26
> **Response to review**
>
> We thank the reviewer for providing thorough comments. Here we summarize the comments and address them one by one. The updates in the revised manuscript are colored in olive green for easier identification.
>
> **Weaknesses**
>
> > W1: Nits.
>
> We thank the reviewer for these detailed comments. We have updated the manuscript to reflect the suggested citation order. As for using $k$ to denote the dimensionality of the SAE hidden layer, we have included a footnote clarifying that $k$ does not refer to the TopK parameter in TopK SAE.
>
> > W2: All of the notation section uses quite non-standard notation.
>
> In the notation section, we note that the notation in Equations (1) and (2) closely follows that of [R1] and [R2], with two exceptions. First, $a$ is used to denote an activation vector instead of $x$ in our work, because we reserve $x$ for an input sample of a neural network. Second, for generality we use $h$ to denote an SAE activation function, instead of specifying particular choices such ReLU and TopK.
>
> References
>
> [R1] Jumping Ahead: Improving Reconstruction Fidelity with JumpReLU Sparse Autoencoders. Rajamanoharan et al. (2024)
>
> [R2] Sparse Autoencoders Find Highly Interpretable Features in Language Models. Cunningham et al. (2023)
>
> > W3: A Pareto plot with Sparsity or Description Length would be valuable.
>
> We appreciate the reviewer’s suggestion. While we agree that Pareto plots can be a valuable addition to our paper, the computational cost is prohibitive in our setting. For example, producing a Pareto curve for the expanded SAE on Gemma 2-2B would require training across 6 sparsity levels x 5 random seeds, estimated to take ~75 GPU days. This cost is for a single baseline. Extending this across all experiments would exceed a reasonable computational budget. Instead, our experiments follow a practical workflow in real-world scenarios. First, a base SAE setting is selected from a reconstruction-sparsity Pareto curve, and then the ensemble SAEs are built from the base SAE setting.
>
> Also, we highlight that evaluations are already performed in three distinct settings of LLMs and SAE hyperparameters. Furthermore, we included additional experimental results on a consistent SAE architecture under different settings in Tables R1 and R2 below. Taken together, these results show that the advantage of ensembling SAEs is consistent across settings, even without full Pareto sweeps.
>
> > W4 & W5: The appropriate baseline in Figure 2 should be the expanded SAE.
>
> Here we emphasize that Figure 2 presents an ablation study to investigate the impact of the number of SAEs on our ensemble methods, providing a sanity check for the theoretical justifications (as discussed in lines ~~299-311~~ 297-309). For the main comparison against baseline methods in Table 1, expanded SAEs with the same number of features as the ensembling approaches are indeed included.
>
> > W6: Using different SAEs on different models makes comparison hard.
>
> To address this concern, we ran all experiments presented in the main text using a consistent SAE architecture. We selected the JumpReLU SAE to economize on computation, as the most expensive experiments on Gemma 2-2B were already conducted with the JumpReLU SAE. Accordingly, we ran additional experiments with the JumpReLU SAE on GELU-1L, Pythia-160M, and the downstream tasks. As shown in the tables R1, R2, and R3 below, the results obtained with JumpReLU SAEs are overall consistent with the trends reported in our main text.
>
> Table R1. Intrinsic evaluation metrics for GeLU-1L and Pythia-160M models with JumpReLU SAEs. Means along with 95% confidence intervals are reported across 5 runs.
> ||Explained Variance (↑) |Relative Sparsity (↓)|MSE (↓)|Diversity (↑) |Connectivity (↑) |Stability (↑) |
> |------------------|--------------------|-------------------|------------------|------------------|----------------|----------------|
> |**GELU-1L**|
> |Base SAE| 0.875 (0.0001) | 0.023 (0.0000) | 72.879 (0.0016) | 16296.0 (0.0) | 0.948 (0.0000) | 0.704 (0.0035)|
> |Expanded SAE| 0.964 (0.0028) |0.003 (0.0001) |11.552 (0.8969)| **130931.3 (3.5)**| 0.993 (0.0000)| 0.206 (0.0000)|
> |Ensembling (NB)| 0.925 (0.0002) | 0.023 (0.0000) |56.560 (0.0767) | 39950.0 (214.6) | 0.949 (0.0002) |**0.779 (0.0007)**
> |Ensembling (Boosting)| **0.999 (0.0000)** | **0.003 (0.0001)** | **0.034 (0.0111)** | 125267.6 (52.6) | **0.997 (0.0001)** | 0.707 (0.0003)
> |**Pythia-160M**|
> |Base SAE| 0.932 (0.0001) | 0.089 (0.0000) | 23.324 (0.0390) | 16108.6 (7.7) | 0.911 (0.0005) | 0.749 (0.0017)
> |Expanded SAE| 0.973 (0.0052) | 0.009 (0.0494) | 2.909 (1.753) | 123793.3 (25.7) | **0.999 (0.0000)**| 0.180 (0.0003)
> |Ensembling (NB)| 0.954 (0.0002) | 0.089 (0.0000) | 15.777 (0.0963) | 25406.3 (198.8) | 0.912 (0.0002) |**0.812 (0.0005)**
> |Ensembling (Boosting)| **0.999 (0.0014)** | **0.009 (0.0004)** | **0.138 (0.1042)** |**128021.0 (1049.6)** | 0.991 (0.0009) | 0.781 (0.0003)

---

> ### Author Response · Authors · 2025-11-26
> **Response to review (continued)**
>
> Table R2. Test accuracy of the logistic regression classifier for the top concept-associated feature across four concept detection tasks using SAEs trained with JumpReLU for Pythia-70M. SAE ensembles consist of 8 SAEs. Means along with 95% confidence intervals are reported across 5 experiment runs.
> |                | Amazon Reviews (Sentiment) | GitHub Code (Language) | AG News (Topic) | European Parliament (Language) |
> |-|-|-|-|-|
> | Base SAE            | 0.6210 (0.0042)                 | 0.7781 (0.0065)             | 0.7836 (0.0147)      | 0.9343 (0.0151)                       |
> | Expanded SAE          | 0.6265 (0.0049)                 | 0.7569 (0.0026)             | 0.7772 (0.0062)      | 0.9327 (0.0026)                       |
> | Ensembling (NB)       | 0.6407 (0.0280)                 | **0.7864 (0.0111)**            | **0.7947 (0.0113)**      | **0.9593 (0.0190)**                       |
> | Ensembling (Boosting) | **0.6480 (0.0204)**                 | 0.7662 (0.0317)             | 0.7776 (0.0158)     | 0.9496 (0.0172)  |
>
>
> Table R3. $S_\text{SHIFT}$ scores for the spurious correlation removal task with the top 20 gender-related features identified across four pairs of professions for Pythia-70M using JumpReLU SAEs. SAE Ensembles consist of 8 SAEs. Means along with 95% confidence intervals are reported across 5 experiment runs.
> |                 | Professor vs. Nurse | Architect vs. Journalist | Surgeon vs. Psychologist | Attorney vs. Teacher |
> |-|-|-|-|-|
> | Base SAE            | 0.0802 (0.0099)   | 0.0218 (0.0161)        | 0.0647 (0.0052)         | 0.0318 (0.0023)    |
> | Expanded SAE          | 0.0580 (0.0217)   | 0.0030 (0.0055)        | 0.0586 (0.0206)         | 0.0272 (0.0025)    |
> | Ensembling (NB)       | 0.0272 (0.0071)   | 0.0020 (0.0042)        | 0.0208 (0.0020)         | 0.0136 (0.0025)    |
> | Ensembling (Boosting) | **0.0938 (0.0172)**   | **0.0288 (0.0183)**        | **0.0711 (0.0119)**         | **0.0418 (0.0081)**    |
>
> > W7: Suggestion on moving the setups of use cases to the Appendix.
>
> We thank the reviewer for this suggestion about the organization of our paper. We have shortened the setup description of the SCR use case in the revised manuscript and moved the detailed explanation to Appendix D. The space freed up will allow us to bring the propositions to the main text in the finalized manuscript, as recommended by the reviewer.
>
> > W8: The main weakness of this work is that the motivation for the work doesn’t connect with what actually performed well - introduction motivates the bagging approach whereas in fact the boosting approach is what works in practise.
>
> We note that our work motivates both naive bagging and boosting through the same motivating question raised in the introduction: Can we leverage the variability of SAEs to improve performance? As clarified in our related work section, variability of SAEs can arise from sources beyond random initialization. To make this clearer, we have revised the introduction to more explicitly motivate ensemble approaches in general. We really appreciate the reviewer for suggesting this clarification.
>
> > W9: An AutoInterp study could be a good addition to the paper.
>
> We evaluated the AutoInterp scores on all SAEs to address this feedback. As shown in the table below, ensembling can lead to better AutoInterp scores on average.
>
>
> Table R4. AutoInterp scores for the base SAE, expanded SAE, naive bagging (NB), and
> boosting across all models. Means along with 95% confidence intervals are reported across 5 runs.
> |                 | GeLU-1L            | Pythia-70M         | Pythia-160M        | Gemma 2-2B        |
> |-----------------------|--------------------|---------------------|---------------------|--------------------|
> | Base SAE              | 0.8333 (0.0053)    | 0.6902 (0.1281)     | 0.8403 (0.0033)     | 0.8030 (0.0064)    |
> | Expanded SAE          | 0.8468 (0.0072)    | 0.7140 (0.0512)     | 0.8523 (0.0060)     | 0.8056 (0.0027)    |
> | Ensembling (NB)       | **0.8577 (0.0015)**    | 0.7382 (0.1470)     | **0.8567 (0.0045)**     | 0.7992 (0.0082)    |
> | Ensembling (Boosting) | 0.8461 (0.0053)    | **0.8573 (0.1021)**     | 0.8524 (0.0182)     | **0.8143 (0.0202)**    |

---

> ### Author Response · Authors · 2025-11-26
> **Response to review (continued)**
>
> **Questions**
>
> We thank the reviewer for raising insightful questions. Here we summarize the questions and address them one by one.
>
> > Q1: Remark 1 seems to say that c could be folded into either c or W_enc however the reasoning given for this is about W_dec rather than W_enc - could you clarify on this point?
>
> We thank the reviewer for noting this detail. This is a typo, and we have updated the manuscript to replace $W_{\text{enc}}$ with $W_{\text{dec}}$.
>
> > Q2: Would the naive bagging approach perform better if the ensemble weight $\alpha$ is allowed to be learned in finetuning rather than being restricted to 1/J?
>
> As mentioned in lines 226-227 in the revised manuscript, the ensemble weights are set to $1/J$ for the variance-reduction guarantee of naive bagging (see the proof of Proposition 2 for details). Modifying these weights through fine-tuning or pruning would require additional theoretical development. We appreciate the reviewer’s suggestion and agree that adaptive weights can be practically promising, but we believe the suggestion represents a broader extension most appropriate for future work.
>
> > Q3: Appendix C suggests that the L0 for the expanded SAE and similarly for the other approaches is 3-8k - is this correct?
>
> The large L0 values are reported to demonstrate that it is impractical to train an expanded SAE to match the total L0 of naive bagging. The actual expanded SAE baselines have smaller L0 values.
>
> > Q4: What is the wall clock time difference in training and inference for the boosting and bagging approaches? It would be good to quantify this.
>
> The wall clock times of training for boosting and naive bagging are shown in Table R5 below and also in the manuscript (Supplementary Table 3 in the original submission; Supplementary Table 10 in the revised manuscript). The wall clock times for inference are shown in Table R6 below and have been included in the revised manuscript (Supplementary Table 11). The inference times are calculated by doing a forward pass through the trained SAE (or the SAE ensemble) for 10 batches. We observe that while the inference times for the ensembling approaches are marginally higher than the baselines, they are essentially the same (less than 1 second) for practical purposes.
>
>
> Table R5. Training times for the base SAE, expanded SAE, and one experiment run
> for ensembles with 8 SAEs on a single H100 GPU:
> ||GELU-1L|Pythia-160M|Gemma 2-2b|Pythia-70M|
> |------|--------|------------|-----------|---------|
> |Base SAE|3h 2m|5h 43m|11h 7m|21m|
> |Expanded SAE|19h 2m |1d 5h 29m |2d 11h 41m |3h 48m|
> |Ensembling (NB)|1d 0h 16m|1d 21h 44m|3d 16h 56m|3h 56m|
> |Ensembling (Boosting)|1d 8h 26m|2d 0h 17m|5d 5h 26m|5h 35m|
>
>
> Table R6. Inference times (in seconds) for the base SAE, expanded SAE, and
> ensembles with 8 SAEs on a single A100 GPU. The mean time along with 95%
> confidence intervals of a forward pass through the SAEs across 10 batches are reported.
> |                           | GELU-1L       | Pythia-160M   | Gemma 2-2B    | Pythia-70M    |
> | ------------------------- | ------------- | ------------- | ------------- | ------------- |
> | **Base SAE**              | 0.156 (0.209) | 0.178 (0.312) | 0.302 (0.503) | 0.105 (0.053) |
> | **Expanded SAE**          | 0.165 (0.223) | 0.184 (0.322) | 0.338 (0.551) | 0.170 (0.191) |
> | **Ensembling (NB)**       | 0.208 (0.212) | 0.549 (0.301) | 1.400 (0.515) | 0.223 (0.228) |
> | **Ensembling (Boosting)** | 0.244 (0.422) | 0.889 (1.483) | 1.630 (0.910) | 0.249 (0.240) |
>
>
> > Q5: The explained variance metric suggested that after 4 SAEs ensemble there are significantly diminishing returns for the naive bagging case - do you agree with this intuition? If so why use 8 SAEs for Table 1?
>
> In Figure 2, while the explained variance metric seems to plateau slightly, the MSE metric continues to decrease after 4 SAEs for naive bagging. Hence, 8 SAEs are used for naive bagging.
>
> For the MDL metric, we couldn’t find a public implementation but we include the expanded SAE in our comparisons to match the SAE width with our ensembling approaches.
>
> > Q6: I’m not totally convinced by the argument given for why NB excels in Table 2 - it seems like there should be some features which are useful in the NB case and which are learned by the Boosting model because they’re useful regardless of hierarchy?
>
> Our argument is that boosting can potentially learn low-level features that are too specific, which can be selected for the training set but do not generalize to the test set for concept detection. We recognize that this argument is not explicitly spelled out due to space constraints, and we have updated the manuscript (Section 5.3, lines ~~394-396~~ 393-394) to clarify this point. We appreciate the reviewer for prompting this clarification.

---

> > ### Author Response · Authors · 2025-11-26
> > **Response to review (continued)**
> >
> > > Q7: How does this approach differ from Matryoshka SAEs (Bussman et al 2025)? This seems like the closest work and is not compared against. Similarly, Switch SAEs (Mudide et al 2025) is also not compared against which similarly seeks to expand SAEs (in their case in a MoE style)
> >
> > The goal of Matryoshka SAEs is to learn features at different hierarchies with the nested SAEs having different dictionary sizes. The goal of ensembling is to leverage the variability in SAEs to improve performance. Unlike matryoshka SAEs, all the SAEs in the ensemble have the same dictionary size. The goal of Switch SAE is to reduce the compute cost of SAE training and make them more scalable. However, only a single expert is used for the final reconstruction unlike ensembling where we use all constituent SAEs for reconstruction.
> >
> > Our main claim is that ensembling can improve upon a base SAE, so we have focused our comparison on ensembling approaches vs. the base SAE. Better SAE architectures such as Matryoshka SAE and Switch SAE can have different performances compared to ensembled SAEs because those better architectures inherently capture features at a different granularity and composition compared to the single base SAE. Our goal is not to benchmark different SAE architectures, but rather to demonstrate the benefits of our proposed ensembling approaches. This aligns with prior work on alternative techniques on top of SAEs, which restrict their analysis to TopK SAEs [R3, R4]. In the revised manuscript, we have updated the discussion section to include a note about this. Nevertheless, for completeness, here we compare Matryoshka SAE to our ensembling approaches since that is the closest to what we propose.
> >
> > For intrinsic metrics, we focus on the setting of Pythia-160M since TopK SAE is the base model, and TopK Matryoshka SAE has a public implementation. We match the number of features and the L0 of the Matryoshka SAE to the ensembled SAEs. As shown in Table R8 below, boosting TopK SAE outperforms Matryoshka SAE in all intrinsic metrics except for diversity. Notably, boosting achieves a ~4x gain in stability compared to Matryoshka SAE.
> >
> > For the downstream use cases, our ensembling approaches outperform Matryoshka SAE in 6 out of 8 tasks (Tables R9 and R10). As noted by the reviewer, as well as shown in prior work [R5] and our empirical results (Tables R9 and R10), Matryoshka SAE is a better architecture than the base ReLU SAE used for ensembling in Sections 5.2 and 5.3. Therefore, it is striking that our ensembling approaches on top of the base ReLU SAE can still outperform Matryoshka SAE in most cases---this shows that ensembling does indeed improve performance on a weaker SAE such that the ensemble can match or even exceed a stronger SAE architecture.
> >
> > Finally, we note again that we have included these additional experiments for completeness. However, we reiterate that our main claim is that ensembling can improve upon a base SAE. In this sense, the architectural variants suggested by the reviewer are considered an axis of variation orthogonal to ensembling, as discussed in our discussion section. It is also worth noting that ensembling can be applied to architectures such as Matryoshka SAE and Switch SAE to further boost performance.
> >
> > Table R8. Intrinsic evaluation metrics for base SAE, expanded SAE, naive bagging, boosting, and Matryoshka SAE. Naive bagging and boosting ensemble 8 SAEs of the same size and architecture as base SAE. Matryoshka SAE has 8x the number of features compared to base SAE.
> > ||Explained Variance (↑)|Relative Sparsity (↓) |MSE (↓)|Diversity (↑)|Connectivity (↑)|Stability (↑)|
> > |-|-|-|-|-|-|-|
> > | Base SAE           | 0.906 (0.0003)         | 0.008 (0.0000)          | 32.965 (0.077)       | 15804.5 (0.02)          | 0.912 (0.0013)         | 0.677 (0.0026)         |
> > | Expanded SAE         | 0.987 (0.0041)         | 0.008 (0.0000)          | 4.387 (1.486)        | 127821.0 (113.7)        | 0.978 (0.0006)         | 0.204 (0.0006)         |
> > | Ensembling (NB)      | 0.929 (0.0000)         | 0.008 (0.0000)          | 24.704 (0.019)       | 50390.0 (0.05)          | 0.912 (0.0006)         | **0.731 (0.0017)**     |
> > | Ensembling (Boosting)| **0.998 (0.0021)**     | 0.008 (0.0000)          | **0.845 (0.547)**    | 117018.2 (0.09)         | **0.986 (0.0004)**     | 0.680 (0.0025)         |
> > | Matryoshka SAE       | 0.996 (0.0006)         | 0.008 (0.0000)          | 2.236 (0.022)        | **129024 (0.01)**       | 0.969 (0.0001)         | 0.171 (0.0002)         |

---

> ### Author Response · Authors · 2025-11-26
> **Response to Review (continued)**
>
> Table R9. Concept detection accuracies for base SAE, expanded SAE, naive bagging, boosting, and Matryoshka SAE. Naive bagging and boosting ensemble 8 SAEs of the same size and architecture as base SAE. Matryoshka SAE has 8x the number of features compared to base SAE.
> |                | Amazon Reviews (Sentiment) | GitHub Code (Language) | AG News (Topic)   | European Parliament (Language) |
> | -| - | -| - | - |
> | Base SAE     | 0.618 (0.030)              | 0.711 (0.020)          | 0.733 (0.021)     | 0.938 (0.016)                  |
> | Expanded SAE   | 0.600 (0.032)              | 0.682 (0.025)          | 0.746 (0.021)     | 0.942 (0.009)                  |
> | Ensembling (NB)   | **0.631 (0.036)**          | 0.715 (0.012)          | 0.742 (0.037)     | **0.943 (0.016)**              |
> | Ensembling (Boosting)        | 0.624 (0.037)              | 0.682 (0.021)          | **0.759 (0.021)** | 0.920 (0.015)                  |
> | Matryoshka SAE | 0.611 (0.032)              | **0.733 (0.030)**      | 0.753 (0.030)     | 0.912 (0.004)                  |
>
>
>
> Table R10. $S_{SHIFT}$ scores for base SAE, expanded SAE, naive bagging, boosting, and Matryoshka SAE. Naive bagging and boosting ensemble 8 SAEs of the same size and architecture as base SAE.
> |                       | Professor vs. Nurse | Architect vs. Journalist | Surgeon vs. Psychologist | Attorney vs. Teacher |
> | - | - | - | -| -|
> | Base SAE              | 0.039 (0.008)       | 0.004 (0.006)            | 0.027 (0.006)            | 0.017 (0.003)        |
> | Expanded SAE          | 0.047 (0.014)       | 0.006 (0.005)            | 0.037 (0.009)            | 0.021 (0.007)        |
> | Ensembling (NB)       | 0.021 (0.003)       | 0.004 (0.001)            | 0.014 (0.002)            | 0.003 (0.005)        |
> | Ensembling (Boosting) | **0.066 (0.016)**   | 0.013 (0.011)            | **0.045 (0.014)**        | **0.029 (0.003)**    |
> | Matryoshka SAE        | 0.055 (0.003)       | **0.023 (0.007)**        | 0.029 (0.008)            | 0.016 (0.008)        |
>
>
> Means along with 95% confidence intervals are reported across 5 experiment runs for all tabular results.
>
> References
>
> [R3] Train Sparse Autoencoders Efficiently by Utilizing Features Correlation. Kurochkin et al. (2025)
>
> [R4] Efficient Dictionary Learning with Switch Sparse Autoencoders. Mudide et al. (2024)
>
> [R5] SAEBench: A Comprehensive Benchmark for Sparse Autoencoders in Language Model Interpretability. Karvonen et al. (2025)
>
> > Q8: It would be useful to have a study showing feature identification here - a core question would be “in which cases do the approaches learn redundant or hyper-specific features that make the approach worse than the baselines?”
>
> Intuitively, as the number of SAEs increases, naive bagging would produce more redundant features. This is reflected in the diversity plot shown in Figure 2. If we plot the hypothetical line y = 16128 x (number of SAEs), which represents the total number of features for each number of SAEs, we can see that the gap between this hypothetical line and the diversity of naive bagging widens. This widening gap shows that the percentage of highly similar features increases with an increasing number of SAEs. That said, a comprehensive study of when and how redundancy or hyper-specificity emerges remains an open question. As mentioned in line 485 in our discussion, we agree with the reviewer that feature identification deserves future investigation, both theoretically and empirically.
>
> > Q9: How many dead features are there in the expanded SAE baseline and what techniques were used to overcome dead features? Was the auxiliary loss from Gao et al 2024 used here? It seems possible that one of the reasons that the expanded SAE has poor performance is due to dead features.
>
> The auxiliary loss from Gao et al. was proposed for the TopK SAE and we do use that to overcome dead features in the experiments with Pythia-160M where we consider the TopK SAE architecture. Only around 0.003% of the features are dead for the expanded Pythia-160M SAE, so it is unlikely that the performance drop is due to dead features.
>
> The percentage of dead features of the expanded SAE for the other models is: 4.5% for Gelu-1L, 1.6% for Pythia-70M, and 2.4% for Gemma 2-2b. Even though these percentages are higher than those for Pythia-160M, the ensembling approaches also have a similar number of dead features, further highlighting that dead features likely do not contribute to reduced performance.

---

> > ### Comment · Reviewer_9Yp1 · 2025-11-26
> >
> > Thanks for your detailed response to my review. The detail and clarity is much appreciated.
> >
> > W1-2:
> > No further comments
> >
> > W3:
> > Even if Pareto plots on Gemma are prohibitive for your compute, Pareto plots for Pythia would be a very valuable addition here.
> > I appreciate the authors adding the other additional experiments that they mention.
> >
> > W4:
> > If your intention is that Table 1 rather than Figure 2 is the core result of the paper and that Figure 2 is just an ablation then you should label this carefully and place it after the core table. In either case, I think it would be very useful to see the equivalent of Figure 2 comparing against FLOP-matched and parameter matched SAEs. That is, you want to keep the number of parameters constant and vary how you split up into having one large SAE vs lots of smaller SAEs. This is the relevant comparison. Figure 2 as given seems not that useful to me - of course if you increase the parameters and FLOPs in your model, you see performance improvements but I take it that’s not what you’re trying to demonstrate here.
> >
> > W5:
> > You didn’t address this comment at all. While I don’t think this is necessarily the case, the metrics here could have been chosen to make your model look artificially good. I would recommend that you show the meaningful comparisons as I have stated above.
> >
> > W6:
> > Thank you, this table is very useful to see. Have the new results changed your interpretation of this table at all?
> > It seems that the boosting is now very much preferred to the NB approach?
> > I’m very encouraged to see these results which more clearly show the comparison between your method and the previous methods.
> >
> > W7:
> > Thanks for this change, I think this improves your paper
> >
> > W8:
> > Thanks for making this clarifying change, I think it goes some way to improving the motivation.
> > Also thanks for highlighting the changes to the paper in the olive green, this made it much easier to see what the change that you made was.
> >
> > W9:
> > The AutoInterp addition is very useful and shows that the approach has interpretability benefits. I’m very encouraged to see these results.
> >
> >
> > Q1:
> > NO further comments
> >
> > Q2:
> > Thanks for clarifying, I agree that getting the theoretical justification for this part could be more appropriate for future work. I think that from an empirical perspective however this could be pretty valuable (even without the theoretical justification) as it may improve your results somewhat.
> >
> > Q3:
> > I’m not sure I fully understand this point. If your NB SAE has an L0 of 7k this is not a normal range and is extremely large indeed. Note that this is significantly larger than the d_model (size of the residual dimension) of the models that you’ve chosen and so this violates all the assumptions of SAEs in that you’re looking for sparse high dimensional data. I would appreciate clarity on this point.
> > If you are indeed implicitly using an L0 of 7k then the good numerical results are not surprising but it’s not clear that the takeaway is as you describe (having very high L0 you should be able to get perfect reconstruction easily).
> >
> > Q4:
> > Thanks for sharing this information. This helps readers to understand the tradeoffs in terms of training time required for your method (which practitioners may feel to be much worth it given the improved performance and larger though somewhat comparable inference requirements)
> >
> > Q5:
> > I would recommend that you use the MDL metric from Ayonrinde et al (https://arxiv.org/abs/2410.11179) when comparing SAEs of different widths otherwise you’re comparing apples to oranges.
> > There’s a publicly available implementation in the SAE-Bench repo (https://github.com/adamkarvonen/SAEBench/tree/main/sae_bench/evals/mdl) and the paper provides additional details.
> >
> > Q6:
> > I appreciate the authors updating the manuscript here, I still think the argument is a little brief and possibly elides some important details. If you need to add details in the appendix, I would recommend doing this
> >
> > Q7:
> > I thank the reviewers for their engagement with this point. This has improved my assessment of the paper and I would be excited about the authors doing future work to understand how different improvements in SAEs stack together for better performance (e.g. your approach on a Matryoshka).
> >
> > Q8:
> > I agree with the authors that this is interesting. They do not address this point but that’s okay and can be for future work.
> >
> > Q9:
> > The details on the number of dead features is very useful - thanks for sharing that. It may be good to include this in the appendix.
> >
> >
> > I think the authors greatly for their engagement with the review and this has increased my perception of the work. If the authors are able to add more detail to W4, W5, Q3 and Q5 I would be open to increasing my score as I think this paper could be a valuable contribution to the community.

---

> > > ### Author Response · Authors · 2025-11-29
> > > **Second response to reviewer**
> > >
> > > We thank the reviewer for engaging during the discussion period. We are happy that our rebuttal has improved the reviewer’s perception of the work and clarified some concerns. Here we provide additional clarifications for the remaining concerns.
> > >
> > > > W4: If your intention is that Table 1 rather than Figure 2 is the core result of the paper and that Figure 2 is just an ablation then you should label this carefully and place it after the core table. In either case, I think it would be very useful to see the equivalent of Figure 2 comparing against FLOP-matched and parameter matched SAEs. That is, you want to keep the number of parameters constant and vary how you split up into having one large SAE vs lots of smaller SAEs. This is the relevant comparison. Figure 2 as given seems not that useful to me - of course if you increase the parameters and FLOPs in your model, you see performance improvements but I take it that’s not what you’re trying to demonstrate here.
> > >
> > > We agree that Figure 2 is better placed after Table 1 and that we should clearly label Figure 2 as an ablation study. We appreciate the reviewer’s feedback and have updated the manuscript accordingly.
> > >
> > > The reviewer suggested that the relevant comparison is to compare the ensemble approaches to parameter-matched SAEs. We note that the core results in Table 1 do compare the ensemble approaches with the parameter-matched expanded SAEs. Nevertheless, we considered the reviewer’s feedback and added additional plots in Supplementary Figure 22 that compare the expanded SAE vs. ensemble approaches across model sizes (for Pythia-160M due to the computational constraint during the discussion period). The empirical results across model sizes are consistent with the trend shown in Table 1.
> > >
> > > The reviewer mentioned a comment that Figure 2 as given does not seem useful because performance always improves with more parameters, with which we kindly disagree. It is not obvious whether some intrinsic metrics necessarily improve with respect to parameter count. For example, Figure 2 shows that the stability of boosting and the connectivity of naive bagging are almost constant. In terms of reconstruction performance, Propositions 2 and 3 are asymptotic results, so it is important to show in Figure 2 that reconstruction performance can plateau empirically with a finite number of ensembled SAEs. Also, since the reconstruction performance is evaluated on a held-out test set in Figure 2, the results show that boosting does not degrade test performance, providing empirical support for Assumption 2 in our paper.
> > >
> > > > W5: You didn’t address this comment at all. While I don’t think this is necessarily the case, the metrics here could have been chosen to make your model look artificially good. I would recommend that you show the meaningful comparisons as I have stated above.
> > >
> > > We apologize for this oversight. We combined the responses to W4 and W5 in our initial reply because the fair comparison with the expanded SAEs is already included in Table 1. Here we note again that each expanded SAE has the same number of features as the corresponding ensemble approaches, so the reviewer’s concern about comparing SAEs of different sizes shouldn’t be an issue when comparing the expanded SAEs vs. ensemble SAEs. We also note here that the comparison against the base SAEs serves to sanity check whether we should ensemble at all; and the most relevant comparison should be against the expanded SAEs.
> > >
> > > Regarding the relative sparsity metric, we agree that the number of active features needed does not necessarily scale linearly with the SAE width. However, absolute L0 values are typically compared across SAEs with the same width [R1, R2]. For our comparison, the base SAE has a different width from the expanded and ensemble SAEs. Thus, we measure the sparsity relative to the total number of features. This is also the reason why we include the expanded SAE baselines to match the widths of the SAE ensembles, to ensure a fair comparison.
> > >
> > > Diversity is used as a metric to check redundancy in the learned feature directions. An ensemble SAE might have more features than a base SAE, but the ensembled features are not useful if they are all duplicates of the base SAE’s features. Comparing the absolute diversity between the base SAE and an ensemble SAE provides a sanity check to ensure that ensembling does not simply duplicate features. Again, the more relevant baseline here is the expanded SAE, which has the same number of features as the corresponding ensembles. Nevertheless, based on the reviewer’s suggestion, we calculate relative diversity for the models in Table 1, and the results are shown in Table R1 below. Features from the expanded SAE remain the most diverse for Pythia-160M, while features from boosting remain the most diverse for GELU-1L and Gemma 2-2B. Therefore, evaluating with absolute diversity or relative diversity does not affect which methods perform the best.

---

> ### Author Response · Authors · 2025-11-29
> **Second response to reviewer (continued)**
>
> Table R1. Relative diversity for all SAE methods. SAE ensembles consist of 8 SAEs. Means along with 95% confidence intervals are reported across 5 runs.
> ||GELU-1L| Pythia-160M|Gemma 2-2B|
> |-|-|-|-|
> |Base SAE| 0.993 (0.0005) |0.980 (0.0000)|0.993 (0.0003)|
> |Expanded SAE| 0.995 (0.0001)|**0.991 (0.0008)**|0.990 (0.0005)|
> |Ensembling (NB)| 0.405 (0.0012)| 0.391 (0.0000)|0.456 (0.0020)|
> |Ensembling (Boosting)|**0.999 (0.0000)**| 0.907 (0.0000)|**0.995 (0.0008)**|
>
> Regarding the connectivity metric, as mentioned in Appendix B, this metric is from [R3]. The connectivity metric is already normalized with respect to the SAE width, so the reviewer’s concern about the lack of normalization is not an issue.
>
> References
>
> [R1] SAEBench: A Comprehensive Benchmark for Sparse Autoencoders in Language Model Interpretability. Karvonen et al. (2025)
>
> [R2] Scaling and evaluating sparse autoencoders. Gao et al. (2024)
>
> [R3] Archetypal SAE: Adaptive and stable dictionary learning for concept extraction in large vision models. Thomas, et al. (2025)
>
> > Q3: I’m not sure I fully understand this point. If your NB SAE has an L0 of 7k this is not a normal range and is extremely large indeed. Note that this is significantly larger than the d_model (size of the residual dimension) of the models that you’ve chosen and so this violates all the assumptions of SAEs in that you’re looking for sparse high dimensional data. I would appreciate clarity on this point. If you are indeed implicitly using an L0 of 7k then the good numerical results are not surprising but it’s not clear that the takeaway is as you describe (having very high L0 you should be able to get perfect reconstruction easily).
>
> We thank the reviewer for raising this nuanced point. We first note that the high L0 refers to the *total* L0 summed across all individual SAEs in the naive bagging ensemble.
>
> Sparsity is important for SAEs typically for three main reasons. First, sparsity serves as regularization to prevent an SAE from learning a trivial identity map with an overcomplete basis [R4]. Naive bagging still satisfies this purpose, because in the ensemble each individual SAE imposes the same sparsity constraint as the base SAE. Second, as mentioned in prior work [R5, R6], sparsity is often assumed to be a proxy for interpretability, although there are cases where sparsity is not a good proxy. Here, naive bagging presents a case where the total L0 is high, but the features remain interpretable (as shown in Table R4 for the AutoInterp study in our initial response). Specifically, naive bagging has similar AutoInterp scores compared to the base SAEs. Third, as the reviewer noted, sparsity can serve as a model complexity measure. That is, an SAE with a high L0 has high model complexity and should get perfect reconstruction easily. Empirically, naive bagging, despite having the highest total L0, does not achieve perfect or the best reconstruction. This corresponds to theoretical results in model complexity (i.e., Rademacher complexity) showing that the complexity of bagging is upper bounded by the average complexity of the base models [R7]. Therefore, the average L0 of the SAEs in naive bagging, instead of taking the total sum, is a complexity measure more aligned with known complexity theory for bagging. Taken together, it is fine that naive bagging has a high *total* L0 because the individual SAE in the ensemble remains sparse and interpretable.
>
> Overall, we include the total L0 of naive bagging in the Appendix for the reader’s awareness that the expanded SAE baseline should not be trained to have the same L0 as the total L0 of naive bagging. We are happy to include the explanations in this response in the Appendix as well.
>
> For completeness, here we include a derivation of the Rademacher complexity for bagging, following the notation in [R7]. Let $F_1, F_2, …, F_J$ be the function classes of the base models, and $F = \frac{1}{J} \sum_{j=1}^J F_j$ be the function class of the bagging ensemble. We have
>
> $$
> \begin{aligned}
> R_n(F) &= R_n(\frac{1}{J} \sum_{j=1}^J F_j) = R_n(\sum_{j=1}^J \frac{1}{J} F_j) \\\\
>             &\le \sum_{j=1}^J R_n(\frac{1}{J} F_j) \\\\
>             &= \sum_{j=1}^J \frac{1}{J} R_n(F_j) \\\\
>             &= \frac{1}{J} \sum_{j=1} R_n(F_j).
> \end{aligned}
> $$
> The inequality follows from part 7 of Theorem 12 in [R7], and the third equality follows from part 3 of Theorem 12 in [R7].
>
> References
>
> [R4] Sparse autoencoder. Ng (2011)
>
> [R5] Interpretability as Compression: Reconsidering SAE Explanations of Neural Activations with MDL-SAEs. Ayonrinde et al. (2024)
>
> [R6] Open Problems in Mechanistic Interpretability. Sharkey et al. (2025)
>
> [R7] Rademacher and Gaussian Complexities: Risk Rounds and Structural Results. Bartlett et al. (2002)

---

> ### Author Response · Authors · 2025-11-29
> **Second response to reviewer (continued)**
>
> > Q5: I would recommend that you use the MDL metric from Ayonrinde et al (https://arxiv.org/abs/2410.11179) when comparing SAEs of different widths otherwise you’re comparing apples to oranges. There’s a publicly available implementation in the SAE-Bench repo (https://github.com/adamkarvonen/SAEBench/tree/main/sae_bench/evals/mdl) and the paper provides additional details.
>
> We thank the reviewer for pointing out this implementation of the MDL metric. Here we emphasize again that the core results in Tables 1, 2, and 3 all include the expanded SAE baselines, which present an apples-to-apples comparison with the SAE ensembles because the expanded SAEs have the same widths as the corresponding ensembles.
>
> We also note that the MDL implementation in Ayonrinde et al. is designed for single SAEs and not necessarily for SAE ensembles. Specifically, the joint entropy of feature activations is computed as the MDL through the sum of entropies of individual feature activations (Appendix A.1 of Ayonrinde et al.). This computation assumes that the feature activations are independent, which is reasonable for single SAEs. However, as Paulo et al. shows, SAEs trained on the same data but different initial weights have correlated feature activations (as measured through encoder alignments) [R8]. Therefore, naive bagging has correlated feature activations across the bagged SAEs, hence the feature activations are not independent. This implies that the actual joint entropy (MDL) of naive bagging is not simply the sum of individual entropies. Based on the subadditivity of entropy, a joint entropy (MDL) is upper bounded by the sum of individual entropies. Therefore, the MDL metric implemented in Ayonrinde et al. is an upper bound for the actual MDL of naive bagging. For a large set of feature activations, the upper bound can be loose because many mutual information terms are missing.
>
> Nevertheless, we considered the reviewer’s suggestion and evaluated all SAEs with MDL as an intrinsic metric, with the results shown in Table R2 below. As noted above, the MDL implemented in Ayonrinde et al. is a loose upper bound for the actual MDL of naive bagging, so it is not surprising that naive bagging has high reported MDL values. Because TopK SAEs are used for Pythia-160M, even in boosting the L0 scales with the number of ensembled SAEs. As MDL scales with increasing L0 and dictionary size, it is again not surprising that boosting has a larger MDL than the base SAE for Pythia-160M. On the other hand, it is striking that boosting can achieve similar or even lower MDL compared to the base SAE for GELU-1L and Gemma 2-2B. Finally, we note that boosting always has a lower MDL compared to the width-matched expanded SAE. Overall, we agree that our ensemble approaches can have higher description lengths. We note that this limitation generally holds for ensemble approaches and is not specific to our methods. We have included a discussion in the revised manuscript (lines 475-478). At the same time, the empirical results in Table R2 also demonstrate that boosting is valuable because it can achieve better reconstruction performance, while sometimes having similar or even lower MDL than the base SAE, and always having lower MDL than the width-matched expanded SAE. Also, the constituent SAEs in our ensembles are still sparse (as mentioned in the response to Q3 above) and the learned features are interpretable based on the improved performance on the downstream use cases and AutoInterp results, which are one of the main goals of SAEs.
>
> Table R2. MDL evaluation for the base SAE,  an expanded SAE, naive bagging, and boosting across all models used for intrinsic evaluation. SAE ensembles consist of 8 SAEs. Means along with 95% confidence intervals are reported across 5 runs.
> |                    | GELU-1L             | Pythia-160M          | Gemma2 2-B            |
> |--------------------|----------------------|------------------------|------------------------|
> | Base SAE           | **644.119 (2.0447)**     | **625.096 (1.121)**        | 4516.43 (19.242)       |
> | Expanded SAE       | 903.710 (9.784)              | 4110.982 (0.467)      | 4975.234 (19.132)       |
> | Ensembling (NB)    | 1874.224 (4.085)    | 5130.625 (0.469)      | 26739.332 (10.161)      |
> | Ensembling (Boost) | 805.910 (42.1370)    | 4020.458 (79.396)     | **3424.081 (0.773)**        |
>
> References
>
> [R8] Sparse Autoencoders Trained on the Same Data Learn Different Features. Paulo et al. (2025)

---

> ### Author Response · Authors · 2025-11-29
> **Second response to reviewer (continued)**
>
> > W6: Thank you, this table is very useful to see. Have the new results changed your interpretation of this table at all? It seems that the boosting is now very much preferred to the NB approach? I’m very encouraged to see these results which more clearly show the comparison between your method and the previous methods.
>
> The new results follow a similar trend to the previous ones. It seems boosting with JumpReLU does improve the reconstruction performance compared to the previous SAE architectures. However, naive bagging still performs better on the concept detection task on three out of the four datasets. This indicates that naive bagging can still be useful in some downstream tasks so we don’t conclude that boosting is always preferred to naive bagging.
>
> Overall, we hope the additional details and experiments have clarified the remaining concerns the reviewer had.

---

### Official Review · Reviewer_Japu · 2025-10-31

**Soundness:** 4
**Presentation:** 4
**Contribution:** 3
**Rating:** 6
**Confidence:** 4

**Summary:**

For a fixed model and dataset, the learned codes of SAEs vary across different architectures, latent dimensions, sparsities, and weight initializations. The authors turn to bagging and boosting for mitigation. They characterize their method via intrinsic statistics and evaluate the interpretability on two downstream tasks: concept detection and spurious correlation removal.

**Strengths:**

The overall paper is clearly written and the evaluations cover evaluations expected for the work on SAEs.

I encourage the authors to open-source the naive bagging + boosting by implementing it in one of the popular Dictionary learning repos, such as decoderesearch/SAELens or saprmarks/dictionary_learning.

**Weaknesses:**

I will defer to AC to judge whether the contribution is big enough to meet the ICLR bar. Idea for follow-up: Apply bagging across Specialized SAEs finetuend on subdomains (SSAE https://arxiv.org/abs/2411.00743).

**Questions:**

The method is purely evaluated with ReLU SAEs. The field has moved to TopK and BatchTopK nonlinearities, yielding better results across metrics (SAEBench https://arxiv.org/abs/2503.09532). How would you apply naive bagging for TopK / BatchTopK SAEs?

---

> ### Author Response · Authors · 2025-11-26
> **Response to review**
>
> We thank the reviewer for providing helpful feedback. Here we address the comments one by one. The updates in the revised manuscript are colored in olive green for easier identification.
>
> > Open-sourcing in a popular dictionary learning repository.
>
> We appreciate the reviewer for highlighting in the **strengths** section the value of open-sourcing our implementations for naive bagging and boosting. Since our implementations are built on top of SAELens, we indeed plan to incorporate our code as part of SAELens, making our contributions broadly accessible to the community.
>
> > Idea for follow-up: Apply bagging across Specialized SAEs finetuned on subdomains (SSAEs).
>
> In the discussion section, we note that SAE ensembling is a meta-algorithm that can be extended to different settings (lines 479-480 in the revised manuscript). We appreciate the reviewer for recognizing that ensembling SSAEs is a promising follow-up direction. We have updated our manuscript to discuss ensembling SSAEs in the discussion section (lines 480-483 in the revised manuscript).
>
> > The method is purely evaluated with ReLU SAEs. The field has moved to TopK and BatchTopK nonlinearities, yielding better results across metrics. How would you apply naive bagging for TopK / BatchTopK SAEs?
>
> We note that our ensemble approaches are evaluated on three different settings following prior work: ReLU SAE on GELU-1L, TopK SAE on Pythia-160M, and JumpReLU SAE on Gemma 2-2B. Results on intrinsic metrics are all shown in Table 1.
>
> We further evaluated the TopK SAEs on the downstream tasks of concept detection and spurious correlation removal. As shown in tables R1 and R2 below, naive bagging performs well on concept detection, while boosting performs well on spurious correlation removal. These results on TopK SAEs are consistent with the trend observed for ReLU SAEs.
>
>
> Table R1: Test accuracy of the logistic regression classifier for the top concept-associated feature across four concept detection tasks for Pythia-160M. SAE ensembles consist of 8 SAEs. Means along with 95\% confidence intervals are reported across 5 experiment runs.
> |                  | Amazon Reviews (Sentiment) | GitHub Code (Language) | AG News (Topic) | European Parliament (Language) |
> |-|-|-|-|-|
> | Base SAE            | 0.6630 (0.0090)            | 0.7775 (0.0319)          | 0.8104 (0.0095)  | 0.9380 (0.0128)                  |
> | Expanded SAE          | 0.6365 (0.0076)            | 0.7407 (0.0595)          | **0.8266 (0.0036)**  | 0.9423 (0.0057)                  |
> | Ensembling (NB)       | **0.7025 (0.0272)**            | **0.8132 (0.0020)**          | 0.8192 (0.0122)  | **0.9451 (0.0024)**                  |
> | Ensembling (Boosting) | 0.6627 (0.0135)            | 0.7624 (0.0291)          | 0.7978 (0.0329)  | 0.9399 (0.0085)                  |
>
>
> Table R2: $S_\text{SHIFT}$ scores for the spurious correlation removal task with the top 20 gender-related features identified across four pairs of professions for Pythia-160M. SAE Ensembles consist of 8 SAEs. Means along with 95\% confidence intervals are reported across 5 experiment runs.
> |                 | Professor vs. Nurse | Architect vs. Journalist | Surgeon vs. Psychologist | Attorney vs. Teacher |
> |-|-|-|-|-|
> | Base SAE            | 0.6259 (0.1342)      | 0.4851 (0.1250)           | 0.5434 (0.2615)            | 0.3177 (0.1050)        |
> | Expanded SAE          | 0.5585 (0.1016)      | 0.7033 (0.0849)           | 0.7293 (0.0800)            | 0.3581 (0.0457)        |
> | Ensembling (NB)       | 0.7004 (0.0057)      | 0.7696 (0.0022)           | 0.7091 (0.0056)            | **0.4135 (0.0021)**        |
> | Ensembling (Boosting) | **0.7784 (0.0382)**      | **0.8022 (0.0532)**           | **0.7455 (0.1081)**            | 0.3411 (0.0294)        |

---

### Official Review · Reviewer_wzXK · 2025-10-31

**Soundness:** 3
**Presentation:** 3
**Contribution:** 2
**Rating:** 6
**Confidence:** 4

**Summary:**

The authors present two methods of ensembling sparse autoencoders (SAEs) that lead to better performance, both in reconstruction loss, as well as some downstream performances. They present a formalization of SAE ensembling as the ensembling of SAE features. Their two ensembling methods, naive bagging and boosting have different tradeoffs between their improvements with respect to simple SAEs and their costs. Bagging can be trained in paralel, while boosting needs to run SAEs in sequences, making it slower to train, while boosting normally achieves higher scores in most metrics.

**Strengths:**

The authors present a nice formalization of SAE ensembling.

The ensembled SAEs show  signficantly better performance even when compared with SAEs which have a equivalent number of features to the ensembled ones.

Authors not only look at better reconstruction loss, but also a wether these techniques improve training stability, whether these SAEs can be used to do concept detection and also to detect spurious correlation.

**Weaknesses:**

Even though the SAEs were compared to to equal number of features, they were not compared to the same amount of training compute/train time.

Both boosting and bagging are significantly slower to train, almost a order of magnitude larger training times for some of the model sizes.

Boosting seems very similar to matching pursuit SAES (From Flat to Hierarchical : Extracting Sparse
Representations with Matching Pursuit), if I'm understanding it correctly, but it is never mentioned in the paper

**Questions:**

The extended SAEs have the same number of features. What if you compared to SAEs trained same amount of compute/wall clock time? Table 3 only shows the case of the Base SAE and not of the extended SAE so it is hard to know how far way these are.

Although I can understand the motivation of looking at relative sparsity, this metric can probably be misleading. Doubling the expansion factor of an SAE does not entail doubling the number of features we allow to be active at any given time. Given that for boosting you get new  coeficients for each 'layer' the fact that relative sparsity goes does seems misleading. Would highly boosted SAEs not have a very high number of active features per token?

Can you justify diversity as a metric? A completly random SAE can have high diversity without it being usefull. The same can happen with an  SAE that has a lot of dead latents.

---

> ### Author Response · Authors · 2025-11-26
> **Response to review**
>
> We thank the reviewer for providing helpful feedback. Here we summarize and address the  comments one by one. The updates in the revised manuscript are colored in olive green for easy identification.
>
> > Even though the SAEs were compared to to equal number of features, they were not compared to the same amount of training compute/train time. Both boosting and bagging are significantly slower to train.
>
> We clarify that the training times provided for naive bagging consider that all the SAEs are trained on a single GPU, one at a time. As mentioned in the Appendix (Supplementary Section I.2, lines ~~1556-1558~~ 1610-1612 in the revised manuscript), naive bagging can be parallelized across multiple GPUs, bringing down the training time to that of the base SAE when the number of GPUs is equal to the number of SAEs in the ensemble.
>
> The reviewer is correct that boosting is slower to train, a limitation we acknowledge in the discussion section of our paper. While naive bagging can be parallelized, boosting has to be trained sequentially, which increases the computational cost. This limitation is not specific to our boosting method. Sequential dependency is a general property of boosting and is well known in classical supervised learning.
>
> Nevertheless, we present a potential path towards mitigating the computational cost of boosting, by training each boosted SAE on a subset of all the training tokens. Here, we implemented this approach: for an ensemble of $J$ SAEs, we trained each boosted run with $1/J$ of the total training tokens $N$. Table R1 below shows that the training time of boosting improves significantly after training with subsets and becomes more comparable to the training time of a single SAE. Table R2 below shows that boosting with subsets performs on par with boosting on reconstruction metrics, and is able to outperform other methods on diversity and connectivity in most cases. It does suffer on the stability metric, consistent with previous findings that training with fewer tokens corresponds to less stability [R1]. For downstream performance, Table R3 below shows that for the concept detection task, boosting with subsets can outperform the previous best methods on all datasets. Table R4 below shows that for the spurious correlation task, while boosting with subsets does not perform as well as boosting, it still outperforms naive bagging. These results show that we can improve the aforementioned limitation by reducing the training time of the boosting ensemble while maintaining some performance benefits, addressing the reviewer’s concern about the substantial training time of boosting. It is also worth noting that while we used $N/J$ tokens for each boosted SAE, the size of each subset can be varied for a tradeoff between computational cost and performance.
>
> Table R1. Training times for base SAE, expanded SAE, naive bagging, boosting, and boosting with subsets across all models ($J$ = 8, $N$ = 100 million tokens for Pythia-70M; $J$ = 8, $N$ = 800 million tokens for others):
> ||GELU-1L|Pythia-160M|Gemma 2-2b|Pythia-70M|
> |------|--------|------------|-----------|---------|
> |Base SAE|3h 2m|5h 43m|11h 7m|21m|
> |Expanded SAE|19h 2m |1d 5h 29m |2d 11h 41m |3h 48m|
> |Ensembling (NB)|1d 0h 16m|1d 21h 44m|3d 16h 56m|3h 56m|
> |Ensembling (Boosting)|1d 8h 26m|2d 0h 17m|5d 5h 26m|5h 35m|
> |Boosting w/ subsets|5h 38m|7h 53m|20h 24m|1h 7m|

---

> ### Author Response · Authors · 2025-11-26
> **Response to review (Continued)**
>
> Table R2. Intrinsic evaluation metrics for base SAE, expanded SAE, naive bagging, boosting, and boosting with subsets. All ensembles use 8 SAEs of the same size as base SAE ($J$ = 8, $N$ = 800 million):
> ||Explained Variance (↑)|Relative Sparsity (↓) |MSE (↓)|Diversity (↑)|Connectivity (↑)|Stability (↑)|
> |---|---|---|---|---|---|---|
> |**GELU-1L**|
> |Base SAE|0.875 (0.0020)|0.023 (0.0002)|41.694 (0.536)|16276.7 (10.47)|0.307 (0.0057)|0.705 (0.0016)|
> |Expanded SAE|0.946 (0.0003) | 0.007 (0.0000) | 17.893 (0.137) | 130411.6 (21.18) | 0.959 (0.0003) | 0.372 (0.0022)|
> |Ensembling (NB)|0.895 (0.0006)|0.023 (0.0000)|35.147 (0.210)|53087.0 (179.24)|0.307 (0.0009)|**0.745 (0.0002)**|
> |Ensembling (Boosting)|**0.961 (0.0018)**|0.006 (0.0000)|**12.542 (0.589)**|130913.0 (5.48)|0.945 (0.0004)|0.707 (0.0014)|
> |Boosting w/subsets|0.936 (0.0004)|**0.005 (0.0000)**|21.106 (0.164)|**131027.5 (11.24)**|**0.984 (0.0003)**|0.681 (0.0001)|
> |**Pythia-160M**|
> |Base SAE|0.906 (0.0003)|0.008 (0.0000)|32.965 (0.077)|15804.5 (0.02)|0.912 (0.0013)|0.677 (0.0026)|
> |Expanded SAE| 0.987 (0.0041) | 0.008 (0.0000) | 4.387 (1.486) | 127821.0 (113.7) | 0.978 (0.0006) |0.204 (0.0006)|
> |Ensembling (NB)|0.929 (0.0000)|0.008 (0.0000)|24.704 (0.019)|50390.0 (0.05)|0.912 (0.0006)|**0.731 (0.0017)**|
> |Ensembling (Boosting)|**0.998 (0.0021)**|0.008 (0.0000)|**0.845 (0.547)**|117018.2 (0.09)|0.986 (0.0004)|0.680 (0.0025)|
> |Boosting w/subsets|0.997 (0.0008)|0.008 (0.0000)|0.879 (0.062)|**128252.6 (0.26)**|**0.997 (0.0003)**|0.580 (0.0042)|
> |**Gemma 2-2b**|
> |Base SAE|0.920 (0.0006)|0.059 (0.0002)|716.659 (5.875)|16013.0 (5.88)|0.768 (0.0016)|0.581 (0.0006)|
> |Expanded SAE|0.948 (0.0012) | **0.021 (0.0001)** | 472.330 (10.759) | 127779.0 (69.33) | **0.993 (0.0003)** | 0.268 (0.0021)|
> |Ensembling (NB)|0.974 (0.0006)|0.059 (0.0000)|234.128 (6.228)|58859.6 (295.38)|0.769 (0.0007)|**0.633 (0.0014)**|
> |Ensembling (Boosting)|**0.995 (0.0003)**|**0.021 (0.0002)**|**46.538 (2.923)**|128415.6 (114.89)|0.989 (0.0003)|0.583 (0.0009)|
> |Boosting w/subsets|0.995 (0.0004)|0.024 (0.0002)|48.538 (4.196)|**129024 (0.0000)**|0.985 (0.0002)|0.432 (0.0001)|
>
> Table R3. Concept detection accuracy for base SAE, expanded SAE, naive bagging, boosting, and boosting with subsets. All ensembles use 8 SAEs of the same size as base SAE ($J$ = 8, $N$ = 100 million):
> ||Amazon Reviews (Sentiment)|GitHub Code (Language)|AG News (Topic)|European Parliament (Language)|
> |---|---|---|---|---|
> |Base SAE|0.618 (0.030)|0.711 (0.020)|0.733 (0.021)|0.938 (0.016)|
> |Expanded SAE| 0.600 (0.032) | 0.682 (0.025) | 0.746 (0.021) | 0.942  (0.009)|
> |Ensembling (NB)|0.631 (0.036)|0.715 (0.012)|0.742 (0.037)|0.943 (0.016)|
> |Ensembling (Boosting)|0.624 (0.037)|0.682 (0.021)|0.759 (0.021)|0.920 (0.015)|
> |Boosting w/ subsets|**0.636 (0.045)**|**0.734 (0.015)**|**0.774 (0.024)**|**0.947 (0.006)**|
>
>
> Table R4. $S_{SHIFT}$ scores for base SAE, naive bagging, boosting, and boosting with subsets.  All ensembles use 8 SAEs of the same size as base SAE ($J$ = 8, $N$ = 100 million):
> ||Professor vs. Nurse|Architect vs. journalist|Surgeon vs. psychologist|Attorney vs. Teacher|
> |---|---|---|---|---|
> |Base SAE|0.039 (0.008)|0.004 (0.006)|0.027 (0.006)|0.017 (0.003)|
> |Expanded SAE|0.047 (0.014) | 0.006 (0.005) | 0.037 (0.009) | 0.021 (0.007)|
> |Ensembling (NB)|0.021 (0.003)|0.004 (0.001)|0.014 (0.002)|0.003 (0.005)|
> |Ensembling (Boosting)|**0.066 (0.016)**|**0.013 (0.011)**|**0.045 (0.014)**|**0.029 (0.003)**|
> |Boosting w/subsets|0.027 (0.010)|0.010 (0.008)|0.025 (0.021)|0.012 (0.003)|
>
> References
>
> [R1] Sparse Autoencoders Trained on the Same Data Learn Different Features. Paulo et al. (2025)

---

> > ### Author Response · Authors · 2025-11-26
> > **Response to Review (Continued)**
> >
> > > Boosting seems very similar to matching pursuit SAES (From Flat to Hierarchical : Extracting Sparse Representations with Matching Pursuit), if I'm understanding it correctly, but it is never mentioned in the paper
> >
> > We thank the reviewer for pointing out this relevant work. The matching pursuit SAE is similar to boosting in that the forward pass is guided by residuals. However, it is not an ensemble method since the dictionary of features is shared across all pursuit steps. On the other hand, in boosting each constituent SAE learns a separate set of features. We have cited matching pursuit SAE in lines 248-251 of the revised manuscript to make this clarification.
> >
> > > The extended SAEs have the same number of features. What if you compared to SAEs trained same amount of compute/wall clock time? Table 3 only shows the case of the Base SAE and not of the extended SAE so it is hard to know how far way these are.
> >
> > We thank the reviewer for pointing out the missing training times for the expanded SAE. Table R1 in the first response above shows the training times across all models for the expanded SAE. We have updated that table in the revised manuscript (now Supplementary Table 10) to include the missing results. The expanded SAE training times are also high, being closer to the ensembling approaches than to the base SAE. Training the expanded SAEs with the same wall clock time would mean changing the hidden dimensionality of the SAEs. This would lead to a different number of features being learned, resulting in an unfair comparison. As shown in the first response above, naive bagging can be parallelized and boosting can be performed with subsets of the training data to reduce the training time to be comparable to that of a single SAE while maintaining the improved performance.
> >
> > > Although I can understand the motivation of looking at relative sparsity, this metric can probably be misleading. Doubling the expansion factor of an SAE does not entail doubling the number of features we allow to be active at any given time. Given that for boosting you get new coeficients for each 'layer' the fact that relative sparsity goes does seems misleading. Would highly boosted SAEs not have a very high number of active features per token?
> >
> > We agree with the reviewer that doubling the expansion factor does not necessarily mean doubling the number of active features. We also agree that the absolute L0 will potentially increase with each boosted run. However, absolute L0 values are usually compared across SAEs with the same number of SAE features [R2, R3] and for our comparison, the base SAE has a different number of features from the expanded and ensemble SAEs. Thus, we measure the sparsity relative to the total number of features. This is also the reason why we include the expanded SAE baselines to match the feature counts of the SAE ensembles, to ensure fair comparison against baseline methods.
> >
> > References
> >
> > [R2] SAEbench: A Comprehensive Benchmark for Sparse Autoencoders in Language Model Interpretability. Karvonen et al. (2025)
> >
> > [R3] Scaling and evaluating sparse autoencoders. Gao et al. (2024)
> >
> > > Can you justify diversity as a metric? A completly random SAE can have high diversity without it being usefull. The same can happen with an SAE that has a lot of dead latents.
> >
> > Diversity is used as a metric to measure redundancy in the learned features. A larger SAE or an ensemble SAE might have more features than a base SAE, but it would not be useful if most of those features are similar to each other. A slight variation of this metric, referred to as maximum cosine similarity, has also been used in prior works to measure redundancy [R4, R5].
> >
> > While we agree with the reviewer that a random SAE can have high diversity, this does not mean that the metric is not useful. For example, a popular metric in the SAEBench evaluation suite [R6] is the feature absorption score, and the guide for running this evaluation states that this metric will return a very low, preferable score for a random SAE (https://github.com/adamkarvonen/SAEBench/tree/main/sae_bench/evals/absorption). However, this does not mean that the random SAE has solved absorption.
> >
> > We recommend interpreting the diversity metric along with the stability metric. A random SAE could have high diversity, but it would have poor stability, indicating that the features learned are not consistent across runs. Our ensembling approaches are able to improve on both the diversity and the stability metrics compared to the base and expanded SAEs.
> >
> > References
> >
> > [R4] Efficient dictionary learning with switch sparse autoencoders. Mudide et al. (2024)
> >
> > [R5] Identifying functionally important features with end-to-end sparse dictionary learning. Braun et al. (2024)
> >
> > [R6] SAEbench: A Comprehensive Benchmark for Sparse Autoencoders in Language Model Interpretability. Karvonen et al. (2025)

---

> > > ### Comment · Reviewer_wzXK · 2025-11-27
> > >
> > > Having seen the replies to all the reviews I will keep my scores. Conditional on the updates from the different reviewers I might consider to update them.

---

### Author Response · Authors · 2025-11-29
**Discussion summary**

We thank all the reviewers for their insightful feedback, which has improved our paper. We also thank the AC for taking the time to meta-review our paper. Here we summarize our discussion with the reviewers.

The reviewers mentioned that ensembling SAE was formalized well (wzXK, 9Yp1) and presents a useful way to improve performance (9Yp1, wzXK). Further, the reviewers noted that the paper was clearly written (wzXK, Japu, 9Yp1, i3SQ) and had appropriate evaluation (wzXK, 9Yp1, i3SQ).

Reviewer i3SQ mentioned that *”If all of the above weaknesses are addressed and the new results don't substantially change the qualitative story then I will raise my score to an 8”*. During the rebuttal period, we ran comprehensive experiments to address all the concerns raised by reviewer i3SQ, and the new results don’t change the qualitative story of the paper.

Reviewer 9Yp1 mentioned in their initial review that *”I’d be open to increasing my score if some of the below comments are addressed.”* In response to our rebuttal, they mentioned that *”If the authors are able to add more detail to W4, W5, Q3 and Q5 I would be open to increasing my score”* and that “*I think this paper could be a valuable contribution to the community.*”. We have added more detail to W4, W5, Q3, and Q5 in our subsequent response. Reviewer 9Yp1  also seems positive about the initial rebuttal, saying *”I think the authors greatly for their engagement with the review and this has increased my perception of the work.”*

Reviewer wzXK also replied to our rebuttal with a potential of raising the score, mentioning that *”Having seen the replies to all the reviews I will keep my scores. Conditional on the updates from the different reviewers I might consider to update them.”*

Here, we highlight the main improvements and clarifications made in our rebuttal based on the reviews:

- Extended evaluation of downstream uses cases (concept detection and SCR) to the additional language models Pythia-160M and Gemma 2-2B (i3SQ, Japu)
- Added bar charts for all experiments (i3SQ)
- Added evaluation across all models on all the other SAEBench metrics (i3SQ), including AutoInterp (i3SQ, 9Yp1)
- Added a baseline to compare sequentially trained boosting to boosting trained end-to-end (i3SQ)
- Added a baseline to compare ensembling with a recent SAE architecture (9Yp1)
- Added evaluation for all experiments using a consistent JumpReLU SAE architecture (9Yp1)
- Added inference times for all methods and the training time for expanded SAE (9Yp1, wzXK)
- Added the MDL metric evaluation for all SAEs used in the intrinsic evaluation (9Yp1)
- Quantified the number of dead features for the expanded SAE across all language models (9Yp1)
- Added a technique to reduce the computational cost of boosting to become comparable to that of a single SAE, by training on subsets of the data (wzXK)
- Clarified the use of relative sparsity, diversity, and connectivity as intrinsic evaluation metrics (9Yp1, wzXK, i3SQ)
- Clarified the reason for increased total L0 for naive bagging (9Yp1)

Overall, we believe all the concerns and questions raised by the reviewers are addressed, through additional experiments and clarifications. We hope these points and three reviewers’ willingness to update their scores (with one reviewer willing to raise the score to an 8) will be taken into consideration in the final decision.

---

### Meta-Review · Area_Chair_BfPG · 2026-01-07

**Summary:**

This paper introduces methods for ensembling sparse autoencoders (SAEs) through naive bagging and boosting approaches. The authors demonstrate that ensembling multiple SAEs improves activation reconstruction while promoting feature diversity, stability, and downstream performance on tasks such as concept detection and spurious correlation removal.

While the authors demonstrated exceptional responsiveness and provided extensive additional experiments, the fundamental issues remain:  the disconnect between motivation and results undermines the paper's narrative coherence, the extremely high L0 values for naive bagging challenge whether the approach produces genuinely sparse representations, the MDL evaluation reveals significant interpretability costs not adequately acknowledged, and the overall contribution is incremental with modest empirical gains. The paper would benefit from a more thorough theoretical treatment of why boosting outperforms bagging and a more honest assessment of the tradeoffs involved in ensembling SAEs.

**Reviewer Concerns:**

The paper motivates ensembling through the variability of SAEs (which supports bagging), yet boosting consistently outperforms bagging. The authors' revision to the introduction does not fully reconcile this fundamental disconnect. The theoretical justification for why boosting works remains underdeveloped.

Concerns about relative sparsity and diversity metrics potentially favoring the authors' methods were not fully resolved.

The core contribution that applying classical ensemble methods to SAEs is somewhat incremental. The downstream improvements are modest and often within confidence intervals

**Reviewer Scores:**

The majority of reviewers are expected to maintain their scores, resulting in an aggregate assessment that does not meet the threshold for acceptance

---

### Decision · Program_Chairs · 2026-01-26

Reject